# Channelised, distributed, and disconnected: subglacial drainage under a valley glacier in the Yukon

**Camilo Rada**[1] **and Christian Schoof**[1]

[1]Department of Earth, Ocean and Atmospheric Sciences, University of British Columbia. 2207 Main Mall, Vancouver, BC Canada.

**Correspondence:** Camilo Rada (camilo@rada.cl)

**Abstract.** The subglacial drainage system is one of the main controls on basal sliding, but remains only partially understood. Here we use an eight-year dataset of borehole observations on a small, alpine polythermal valley glacier in the Yukon Territory to assess qualitatively how well the established understanding of drainage physics explains the observed temporal evolution and spatial configuration of the drainage system. We find that the standard picture of a channelising drainage system that evolves towards higher effective pressure explains many features of the dataset. However, our dataset underlines the importance of hydraulic isolation of parts of the bed. We observe how disconnected portions of the bed systematically grow towards the end of the summer season, causing the drainage system to fragment into progressively more distinct subsystems. We conclude with an adaptation of existing drainage models that aims to capture the ability of parts of the bed to become hydraulically disconnected due to basal cavities of finite size becoming disconnected from each other as they shrink.

## 1 Introduction

Basal sliding often accounts for about half of the observed surface speed of glaciers (e.g. Gerrard et al., 1952; Vivian, 1980; Boulton and Hindmarsh, 1987; Blake et al., 1994; Harper et al., 1998). The sliding rate typically shows a marked seasonal variation, with summer sliding speeds sometimes two or three times faster than winter averages (Nienow et al., 1998a; Sole et al., 2011; Ryser et al., 2014b). These variations are controlled by the subglacial drainage system (Iken and Bindschadler, 1986; Gordon et al., 1998; Nienow et al., 1998b; Mair et al., 2001; Harper et al., 2005). However, the physical processes controlling the magnitude

and timing of sliding rate variations are still incompletely understood.

The main variable linking subglacial drainage processes to basal sliding is effective pressure $N$, defined as the difference between normal stress at the bed (averaged over the scale of any basal heterogeneities) and water pressure, where normal stress is usually taken to be equal to the overburden pressure. Increased basal water pressure reduces $N$ and provides partial support for the weight of the glacier, reducing the contact surface with the underlying bedrock, and therefore enhancing basal sliding (Lliboutry, 1958; Hodge, 1979; Iken and Bindschadler, 1986; Fowler, 1987; Schoof, 2005; Gagliardini et al., 2007). A similar effect is observed on glaciers resting on a till layer, where a lower $N$ reduces the yield stress of the till, and therefore also enhances basal sliding (sliding is here intended to include motion at shallow depths within the till layer as well as at the ice-till interface) (Engelhardt et al., 1978; Iverson et al., 1999; Tulaczyk et al., 2000; Truffer et al., 2001). Conversely, large effective pressures enhance the mechanical coupling at the bed interface and therefore reduce sliding.

The magnitude of $N$ is controlled by the combined effect of the rate of meltwater supply and the configuration of the englacial and subglacial conduits that drain the water out of the glacier (Iken et al., 1983; Kamb et al., 1985; Iken and Bindschadler, 1986). For a given conduit configuration, an increase in water supply is likely to decrease effective pressure. Specifically, water pressure gradients must increase in order to evacuate the additional water input to the system, requiring larger water pressures near locations of water supply to the bed.

The extent to which water pressure is raised by increased water supply depends on the following three factors: the permeability of till underlying the glacier, the configuration of

conduits, both at the bed and in the ice, and the storage capacity of the drainage system, which can act to buffer the effect of additional water supply. In turn, the conduits that make up the drainage system can change in response to changes in water input, as the associated changes in effective pressure affect the rate at which viscous creep closes subglacial or englacial conduits. Changes in sliding (themselves due to changes in effective pressure) will also affect the opening of basal cavities (Hoffman and Price, 2014), and changes in discharge affect the rate of conduit enlargement by wall melting. Therefore, over time, the response of the drainage system to the same water input pattern can change (Schoof, 2010).

Current drainage models have succeeded in reproducing observed variations of glacier velocities at a seasonal scale, and several features of the drainage system. These models typically consider a system composed of two main types of conduits, R-(Röthlisberger) channels (Röthlisberger, 1972) and linked cavities (Lliboutry, 1968; Walder, 1986). Other types of conduits and modes of water transport have been hypothesised (Alley et al., 1986; Walder, 1982; Walder and Fowler, 1994; Ng, 2000; Boulton et al., 2007; van der Wel et al., 2013), but their relevance to alpine glaciers remain unclear.

R-channels grow by turbulent dissipation of heat and close due to ice creep. The creep closure of a channel is driven by the effective pressure, and balanced by the melting of its walls by heat dissipated from turbulent water flow (Röthlisberger, 1972). Multiple channels in close proximity are unstable. In such configuration, one channel that is slightly larger than its neighbours will also carry a larger discharge resulting in higher dissipation and a faster opening rate. The creep closure rate will also be faster in the larger channel than the smaller one, but is less sensitive to size than the opening rate (Schoof, 2010). Therefore, the larger channel will grow larger at the expense of the smaller ones. This process tends to focus water flow into a few large channels, leading to the formation of an arterial drainage system covering a small fraction of the glacier bed (Fountain and Walder, 1998; Schoof, 2010; Hewitt, 2011, 2013; Werder et al., 2013).

In an R-channel, steady state is reached at a higher effective pressure when the channel discharge ($Q$) increases, as a faster melt rate has to be offset by a faster closure rate. By implication, when water drains through channels, an increase in water supply should increase effective pressure around the channels, and slow the glacier down (Nye, 1976; Spring and Hutter, 1982; Schoof, 2010).

On the other hand, linked cavity systems are thought to provide a less efficient transport mechanism, where slow water flow provides negligible heat dissipation. Cavities are kept open by the sliding of ice over bed roughness elements, which causes an ice-bed gap to open in their lee, while they also close by viscous creep (Lliboutry, 1968; Kamb et al., 1985; Fowler, 1987).

Unlike channels, multiple cavities can co-exist in close proximity, because a larger cavity size facilitates faster creep closure rates, while the opening rate is generally assumed not to depend significantly on size. Therefore, larger cavities will tend to close faster and converge to equilibrium with small ones (Kamb et al., 1985; Fowler, 1987; Creyts and Schoof, 2009).

In contrast to channels, equilibrium in a linked cavity system is reached at lower effective pressure when $Q$ increases: cavities have to grow to accommodate additional discharge, and this requires creep closure to be suppressed by a reduced effective pressure. Therefore, an increase in $Q$ should decrease effective pressure, and speed the glacier up (Kamb, 1987; Schoof, 2010).

If a cavity becomes disconnected, its fixed volume will result in a water pressure drop if sliding accelerates. Conversely, decelerating basal sliding will lead to relatively high water pressure in order to prevent creep closure, reducing basal drag. In other words, isolated cavities can act either as sticky spots when basal sliding speeds up or as slippery spots when it slows down, working as a buffer for basal sliding variations (Iken and Truffer, 1997; Bartholomaus et al., 2011).

The formation of channels can be understood as an instability in drainage through a distributed network of conduits, and can be expected to occur when water supply rates to the bed are sufficiently large (Schoof, 2010; Hewitt, 2011, 2013; Werder et al., 2013). However, even then the formation of a well-developed arterial channel network requires time and may not be fully complete in a single summer melt season.

Drainage models that include the above physics (e.g., Werder et al., 2013), still fail to reproduce direct borehole observations of subglacial conditions (Flowers, 2015). These include the existence of disconnected areas that show no signs of flow-related changes in water pressure (Hodge, 1979; Engelhardt et al., 1978; Murray and Clarke, 1995; Hoffman et al., 2016), the development of widespread areas of high water pressure during winter (Fudge et al., 2005; Harper et al., 2005; Ryser et al., 2014a; Wright et al., 2016), large pressure gradients over short distances (Murray and Clarke, 1995; Iken and Truffer, 1997; Fudge et al., 2008; Andrews et al., 2014), sudden reorganisations of the drainage system (Gordon et al., 1998; Kavanaugh and Clarke, 2000), high spatial heterogeneity, boreholes exhibiting anti-correlated temporal pressure variations (Murray and Clarke, 1995; Gordon et al., 1998; Andrews et al., 2014; Lefeuvre et al., 2015; Ryser et al., 2014a), and englacial conduits (Fountain and Walder, 1998; Nienow et al., 1998b; Gordon et al., 1998; Fountain et al., 2005; Harper et al., 2010).

The relative scarcity of subglacial observations make it difficult to assess how common these phenomena are, and in some cases, the physical processes involved. In this paper, we take a holistic view of an eight-year dataset of borehole water pressure records and surface conditions obtained from a small polythermal valley glacier in the Yukon Territory, Canada. This dataset includes 311 boreholes with up to 150 being recorded simultaneously. We attempt to present a com-

prehensive picture of the evolution of the drainage system, incorporating all the main features of the borehole record.

Our aim in this paper is to assess qualitatively the extent to which established understanding of drainage physics is compatible with our observations, and where existing models are in conflict with those observations. We will then present a modification of a class of existing models intended to account for what appears to be the most significant missing physics: the development of hydraulically isolated patches of the bed.

The paper is laid out as follows: in section 2, we describe the field site and observational methodology. An overview of our observations is given in section 3, with a physical interpretation presented in section 4. Motivated by our observations, we present the model modification in section 5, focusing on the dynamic organisation of the drainage system into active and hydraulically isolated components. This will not provide a full account of the presented dataset, which also includes surface speed and other variables. An in-depth study of the evolution of the subglacial drainage system structure and its relationships with measured surface speeds is ongoing and will be presented in upcoming papers.

To help the reader to navigate through the numerous observations presented in this paper, we provide below an extended overview of its contents, highlighting the most important points to be considered:

– The observed drainage system consists of three main components (sections 3.1)

  1. Channelised: efficient, turbulent drainage at low water pressure

  2. Distributed: slow water velocities, damped response to diurnal meltwater input, high water pressure

  3. Disconnected: near-overburden mean water pressure with no diurnal variations

– The "disconnected" areas display a small but statistically significant and sustained drop in mean pressure during the melt season, suggesting weak connections potentially through porewater diffusion in till (sections 3.1 & 4.2).

– The connected drainage system consists of spatially distinct parts (subsystems) that appear to act independently. Each is characterised by a common diurnal pressure variation pattern that differs markedly from other subsystems (sections 3.2 & 4).

– Pressure variations in boreholes in disconnected areas can also occur due to bridging effects and potentially due to ice motion, the latter giving rise to low-amplitude, high frequency pressure variations shared by distant boreholes (sections 3.2, 4.2 & 4.3).

– Observations suggest the existence of a dense network of englacial conduits, but it is unclear if these can transport water over extended distances horizontally (sections 3.3 & 4.2).

– During a spring event, a large distributed drainage system quickly develops over a large fraction of the bed. This splits into an increasing number of subsystems over the summer season, each potentially focusing around a channelised drainage axis. The extent of disconnected areas of the bed grows as a result (sections 3.4 & 4).

– The transition from connected to disconnected is abrupt, with the connected parts of the bed having a high hydraulic diffusivity (sections 3.5 & 4.2). Disconnection and reconnection "events" typically occur as water pressure is falling and rising, respectively. These observations motivate the modification of existing drainage models presented in section 5.

– The timing and degree of channelisation reached by the subglacial drainage system varies widely depending on weather and surface conditions during summer, and the spatial pattern of drainage can change from year to year (sections 3.6 & 4.1).

– Abrupt growth of the distributed drainage system, analogous to that observed during the spring event, can be observed during the summer in response to a sudden, abundant meltwater input following an extended hiatus, the latter usually caused by a mid-summer snowfall event (section 4)."

## 2 Field site and methods

All observation presented were made on a small (4.28 km$^2$), unnamed surge-type alpine glacier in the St. Elias Mountains, Yukon Territory, Canada, located at 60° 49' N, 139° 8' W (Fig. 1). We will refer to the site as "South Glacier" for consistency with prior work (Paoli and Flowers, 2009; Flowers et al., 2011, 2014; Schoof et al., 2014). Surface elevation ranges from 1,960 to 2,930 m above sea level (asl), with an average slope of 12.6°. The equilibrium line altitude (ELA) lies at about 2,550 m (Wheler, 2009). Bedrock topography at the site has been reconstructed from extensive ground-penetrating radar (GPR) surveys (Wilson et al., 2013), reporting an average and maximum thickness of 76 m and 204 m respectively. Direct instrumentation and radar scattering (Wheler and Flowers, 2011; Wilson et al., 2013) reveal a polythermal structure with a basal layer of temperate ice. Exposed bedrock in the valley consists mainly of highly fractured Shield Pluton granodiorite (Dodds and Campbell, 1988; Crompton et al., 2015). Borehole videos have also shown the presence of granodiorite cobbles in the basal ice, and highly turbid water near the bottom of freshly drilled

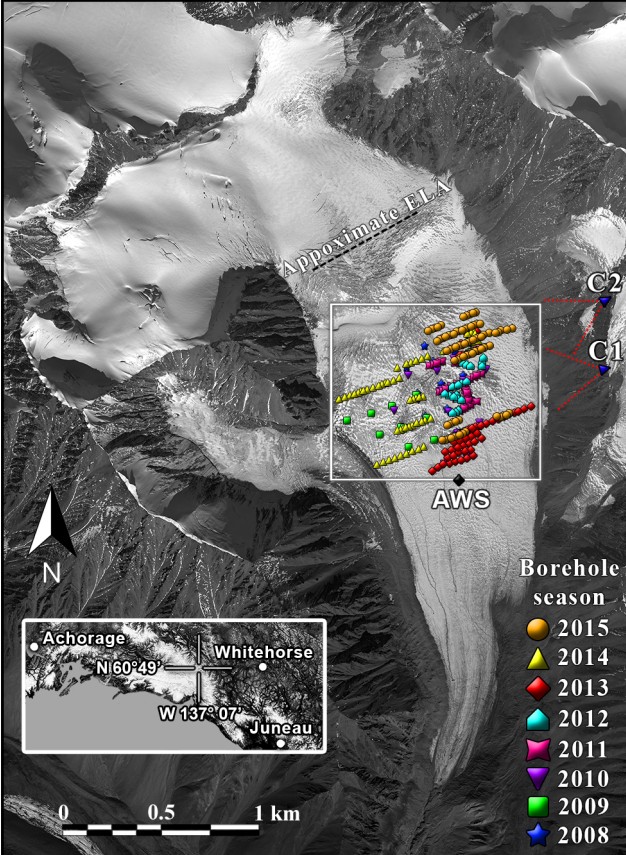

**Figure 1.** WorldView-1 satellite image of South Glacier taken on September 2nd, 2009. Borehole positions are marked according to the year of drilling, showing the most recent year in repeatedly drilled locations. Time-lapse camera positions (C1 & C2), Automatic Weather Station (AWS) approximate equilibrium line (ELA) are also indicated. The inset map shows the general location in the Yukon. The white box corresponds to the area shown in Fig. 2. Note that the different symbols indicating years of borehole drilling are used systematically through the text.

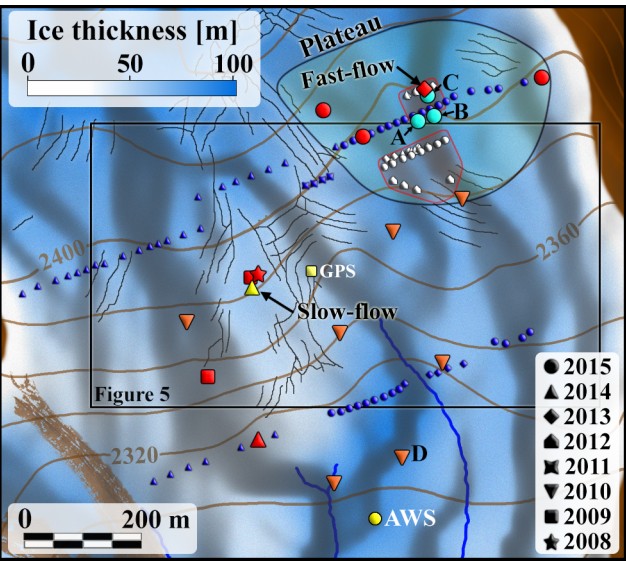

**Figure 2.** Detailed map of the study area. The following symbols indicate specific boreholes: those used for non spatially-biased statistics (blue symbols), displaying behaviour similar to the fast-flow hole in Fig. 5 (red symbols), re-drilled ones (light blue symbols), and those used in Fig. 10 (orange symbols), the 2014 slow-flow borehole in Fig. 6 (yellow triangle), the location of the Automatic Weather Station (yellow circle), and the central GPS tower shown in Flowers et al. (2014) (yellow square). The red outlines encompass all the boreholes displayed in figure 14, shown here using coloured and white markers. Black lines indicate major crevasses, blue lines surface streams. Contours show surface elevation, blue shading ice thickness. Grey shading indicates the upstream area, calculated assuming an hydraulic gradient given by an effective pressure equal to half of the ice overburden pressure, and computed using the $D\infty$ method described by Tarboton (1997).

boreholes. Frozen-on sediments and a basal layer of till of unknown depth are visible in some borehole imagery, and till thicknesses in excess of two metres are exposed near the snout.

An automatic weather station (AWS) operated at 2,290 m next to the lower end of the study area between July 2006 and August 2015 (MacDougall and Flowers, 2011) as part of a simultaneous energy balance study (Wheler and Flowers, 2011). The average net mass balance over the whole glacier during the period 2008-2012 was estimated to be between -0.33 and -0.45 m/year water equivalent (Wheler et al., 2014), corresponding to 37-51 cm/year of average glacier thinning. Elevation changes in the study area derived from differential Global Positioning System (GPS) measurements of borehole locations (taken after drilling) suggest a thinning of 59

cm/year over the same period, and 37 cm/year in the period 2008-2015.

We use air temperatures (specifically positive air temperatures, meaning the maximum of measured temperature and $0°$ C) and Positive Degree Days (PDD, defined in the usual way as the integral with respect to time over positive air temperatures) as the main proxy of the water input into the subglacial drainage system. Temperature estimates after the August 2015 removal of the on-glacier AWS were calculated by a calibrated linear regression of data from a second AWS operated since 2006 by the Geological Survey of Canada and the University of Ottawa 8.8 km to the Southwest, at an elevation of 1845 m.

Surface velocities were measured with a GPS array (Flowers et al., 2014), and display a strong seasonal contrast. The velocity at the GPS tower at the centre of the array (see Fig. 2) varied from 30.6 to 17.9 m/year between summer 2010 and early spring 2011. Modelled basal motion in our study area accounts for 75–100% of the total surface motion (see

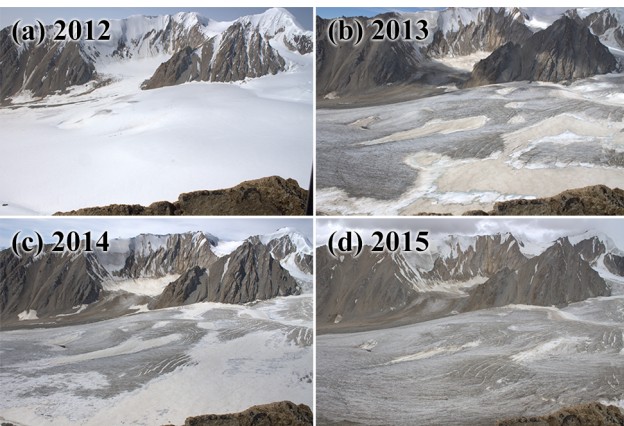

**Figure 3.** Photographs of the study area taken from camera C1 (see Fig. 1) on July 19th, 2012–2015, as indicated in each panel. The interannual variability evident in the photo will be discussed in section 4.

Fig. 6b in Flowers et al. (2011), where our study area is located between 1600 and 2500 metres).

Between 2008 and 2015, 311 boreholes were drilled to the bed (Schoof et al., 2014) in the upper ablation area of the glacier between 2,270 and 2,430 m asl (Figs. 1 and 3), covering an area of approximately 0.6 km$^2$, with an average ice thickness of 63.4 m and a maximum of 100 m. No moulins are visible in or above this area. Instead, the surface meltwater is routed into the glacier through abundant crevasses (Fig. 2). The basal layer of temperate ice in the study area extends up to 30–60 m above the bed.

Boreholes were instrumented with pressure transducers providing continuous subglacial water pressure records, with up to 150 boreholes being recorded simultaneously. The inclination of the boreholes was not measured, but the drilling technique used aims to ensure minimal deviations from vertical. A comparison with GPR data shows that borehole lengths were generally in agreement with ice thickness within a 6% margin (Wilson et al., 2013). Contact with the bed was deemed to have been established if water samples taken from the bottom of the holes showed significant turbidity. Otherwise, a borehole camera was used to assess bed contact visually; a significant number of additional, unsuccessful drilling attempts terminated at englacial cobbles near the bed. With only a few exceptions, sensors were installed only in holes that we were confident had reached the bed, and placed 10–20 cm from the bottom. Boreholes typically froze shut within one to two days, becoming isolated from the surface. The spatial distribution of new boreholes varied each year, not following a regular pattern. However, they were generally 15–60 m apart along cross-glacier lines, with lines 60–120 m apart. A map of all boreholes drilled is shown in Fig. 1. The region labelled as the "plateau" in Fig. 2 was re-drilled every year between 2011 and 2015.

Pressure data were acquired using Barksdale model 422-H2-06 and 422-H2-06-A and Honeywell model 19C200PG5K and SPTMV1000PA5W02 transducers. Each sensor was embedded in clear epoxy to provide mechanical strength and waterproofing. Most transducers installed from summer 2013 onwards were equipped with a Ray 010B ¼" brass piston snubber as protection against transient high-pressure spikes, without altering the signal at the sensor sampling frequencies as verified by doubly-instrumented boreholes (see supplementary material section 1). Data were recorded by Campbell Scientific CR10, CR10X and CR1000 data loggers, set to log at intervals of 2 minutes during summer for CR10(X) loggers, switching to 20 minutes for the rest of the year, and at intervals of 1 minute for CR1000 loggers year-round. In the present paper, water pressure values will be reported in metres of water (the height of the water column that would produce that pressure).

During the summers of 2014 and 2015, a total of 10 custom-made digital sensor pods was installed in boreholes. These pods were built around an ATMega328P microprocessor and communicated via the RS-485 protocol with custom-made data loggers constructed using the Arduino Mega open-hardware platform. The sensor pods recorded pressure, conductivity, turbidity, reflectivity in five spectral bands, tilt, orientation, movement, temperature, and confinement. The latter is a measure of the magnitude of the acceleration produced by an internal vibrating motor, used to assess whether the sensor was hanging freely in water, or tightly confined within solid walls. Seven of the digital sensors were installed in the same boreholes as the standard analogue transducers to assess data quality (see section 1 of the supplementary material).

We have not used data from a stream gauge at the outlet of the glacier, maintained for part of the observation period by the Simon Fraser University glaciology group, for two reasons: first, several surface melt streams and at least one major lateral stream enter the glacier below the study site. Second, the instrumentation at the stream site was destroyed on multiple occasions by flood waters, and a continuous record is not available.

The limited available stream gauging data suggests typical summer flow around 1-2 m$^3$/s, with maximum values around 5 m$^3$/s and minima below the measuring capacity of the gauging station (Crompton et al., 2015). However, the outlet stream was never observed to run dry (J. Crompton, personal communication).

## 3   Results

### 3.1   Modes of water flow: fast, slow and unconnected

Despite a large diversity of borehole pressure records, a few general patterns are easy to identify. The most common is the contrast between an inactive winter regime and an ac-

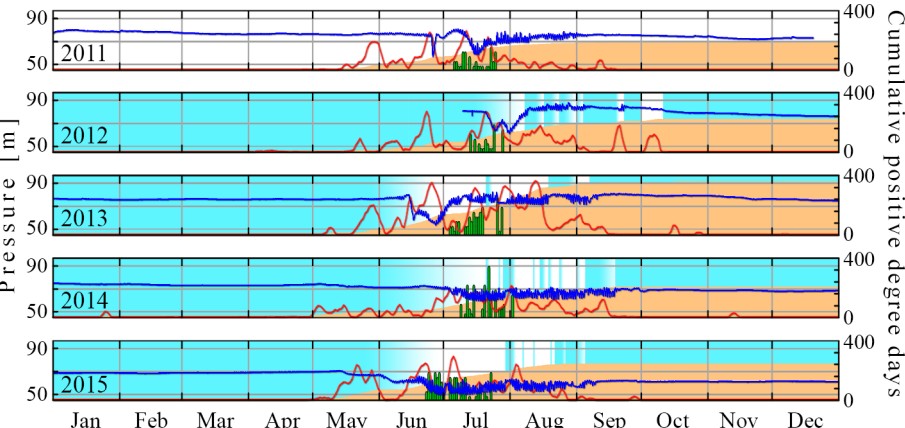

**Figure 4.** Pressure time series recorded in borehole D of Fig. 2 from 2011 to 2015 (blue line). Daily PDD values are shown as a red line, annual cumulative positive degree days as orange shading, and fresh snow cover determined from time lapse imagery as light blue shading. The fading blue at the end of the winter indicates the appearance of larger snow-free patches and the filling of a perennial supraglacial pond in the study area, rather than the complete disappearance of the winter snowpack. Green bars indicate the count of boreholes drilled each day on a scale from zero to 13.

tive summer period. During winter, most sensors show stable, high (near overburden) water pressures, interrupted only during a 2-4 months period of summer activity starting in June-July (Fig. 4). The onset of the active summer period (or "spring event") occurs during rapid thinning of the snowpack under high summer temperatures. After the spring event, 20% of sensors show a drop in diurnal running mean pressure, and most start displaying diurnal oscillations.

Pressure records alone do not allow us to determine the characteristics of water flow at the bed, and visual observations at the bottom of boreholes often fail due to the high turbidity of the water after drilling. However, in a few exceptional cases, we were able to observe water flow at the bed directly. We will describe the two most clear-cut cases.

On July 28th, 2013, while installing a sensor at the bottom of a borehole, strong periodic pulls were felt through the sensor cable, revealing a conduit with turbulent, fast water flow in the bottom 50 cm of the borehole. This borehole was also the only one in which there was an audible sound of flowing water. The location of the hole is marked as "Fast-Flow" in Fig. 2, it was drilled at the very end of the field operations, and no further detailed on-site investigation was conducted.

The fast-flow borehole was 93 m deep and drained at a depth of 87 m during drilling. On the first recorded diurnal pressure peak, the water reached a pressure of about 5.2 m (6% of ice overburden). A water sample retrieved from the bottom showed moderate turbidity. Two pressure sensors were installed in this borehole, 10 and 80 cm above the bed, the upper one with a snubber and the lower one without.

Panel c of Fig. 5 shows the pressure recorded in the fast-flow borehole for the first 33 days after installation, and panel 5d shows the pressure records in three boreholes along the same line across the glacier at 15 m spacing. Note the lack of similarity between the fast-flow hole pressure record and those from other nearby boreholes. This lack of similarity contrasts with the typical behaviour of boreholes exhibiting diurnal pressure oscillations. Such boreholes usually share a similar pattern of pressure oscillations with one or more neighbouring boreholes, forming a cluster that extends some distance laterally across the glacier (see section 3.2).

However, in the case of the fast-flow borehole, somewhat similar temporal pressure patterns were observed down-glacier and at much larger distances than the 15 m lateral borehole spacing, as shown in panels 5e and 5g, and less so in panel 5h. By contrast, a set of boreholes exhibiting very different variations close to those in panel 5d is shown in panels 5f. For reference, panel 5i shows the remaining pressure time series recorded in the same area, highlighting the diversity of pressure patterns observed. No systematic time lags were found between peaks in the fast-flow borehole and pressure peaks of boreholes displayed in panels 5e and 5g.

The grouping of boreholes into panels in Fig. 5 was done on the basis of spatial proximity in panel c, and on the basis of a commonality of diurnal pressure variations in the remaining panels. In particular, we have clustered the records on the basis of commonality in how the amplitude of diurnal pressure variations changes in time. For instance, the similarity between the records in panel 5g should be obvious. However, note that there can be subtler similarities: panels 5c, 5e and 5g at least partially share a period of larger diurnal amplitudes leading up to August 3rd, a hiatus lasting until August 10th punctuated by a diurnal pressure peak late on August 6th, and a period of renewed diurnal oscillations lasting until August 17th; this differs from the pattern of diurnal oscillations seen in panel 5h. Grouping boreholes in this way is partially a subjective measure, and we will present a more systematic clustering method (which has helped to guide the groupings here) in a separate paper (see also

mined from time lapse imagery (light blue shading, fading colour indicates partial cover). (c) Pressure in the fast-flow borehole (red) and its correlation with temperature in grey, computed for any given time over a 3-day running window. Note that two sensors were installed in the fast-flow borehole, offset vertically from each other by 70 cm, making the two lines indistinguishable most of the time at the presented scale. Later in section 3.5, the complete record will be displayed, where the two curves are more distinguishable. (d-i) Pressure records from other boreholes marked on the map. The colour of plots corresponds to borehole marker colours on the map; the same convention is used in all subsequent figures. Symbol shapes represent drilling years as in Fig. 2, uncoloured (grey) markers correspond to boreholes with active sensors during the displayed period whose pressure time series are not shown in the figure.

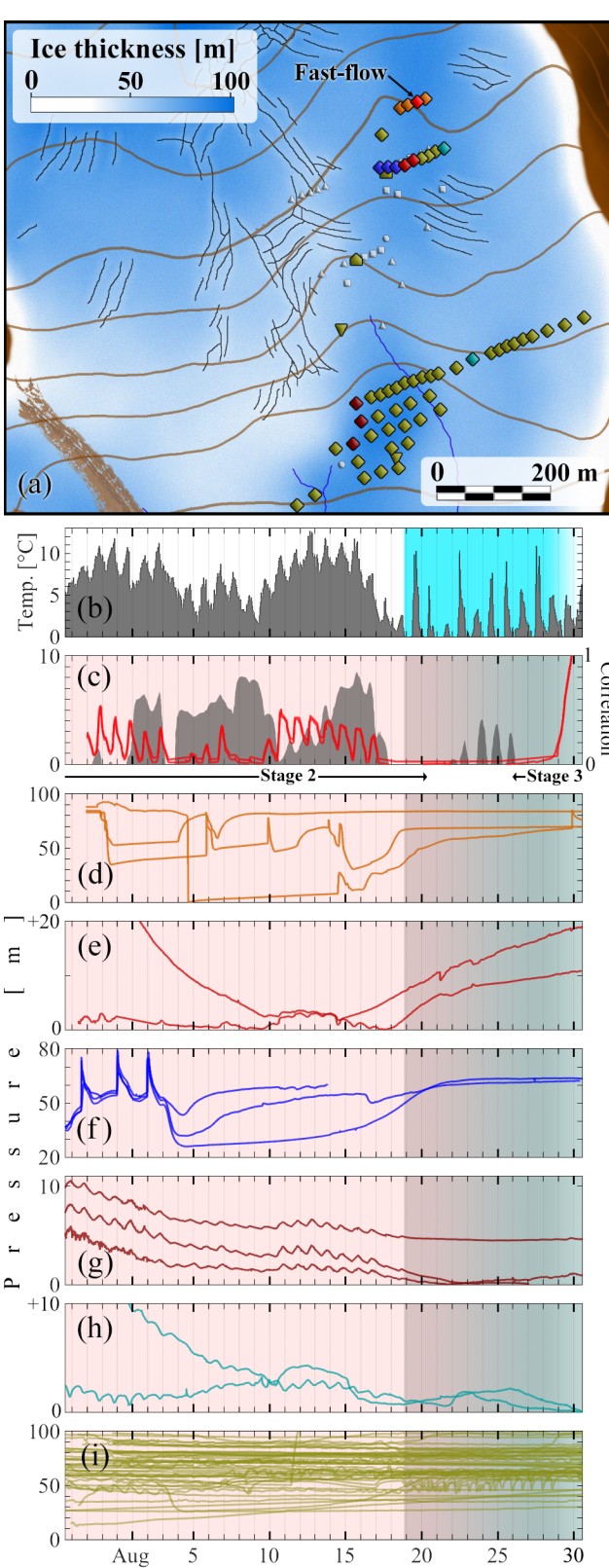

**Figure 5.** Locations and pressure time series for the boreholes associated with the fast-flow borehole during the summer of 2013. (a) The map uses the same scheme as Fig. 2, but omits the upstream area shading. (b) Temperatures (grey) and fresh snow cover deter-

Gordon et al. (1998); Huzurbazar and Humphrey (2008)). All borehole groupings presented in the following figures were manually selected using the same criteria as described for Fig. 5.

Several features stand out in the pressure record from the fast-flow borehole: sharp diurnal pressure peaks and a small time lag between peak surface temperatures and diurnal water pressure maxima (1-3 hours), as well as the general similarity between the temporal variations in pressure and temperature. The correlation between the two, computed over a moving window, stayed above 0.8 for several days (Fig. 5c, grey shading). This high correlation was more pronounced late in the season, also coinciding with the water pressure dropping to atmospheric values at night.

A contrasting observation of water flow was made on July 23rd, 2014, when a clear water sample was retrieved from the bottom of a borehole ("Slow-flow" in Fig. 2) and the borehole camera was deployed. The resulting borehole video (see supplementary material) reveals a slowly-flowing, thin layer of turbid water at the borehole bottom overlain by clear water, an unusual condition that allowed the observation, as the water in a bed-terminating hole is usually highly turbid due to the basal sediments disturbed by the drill jet.

The slow-flow borehole was 62 m deep, and the first recorded diurnal pressure peak reached 48 m (85% of ice overburden). One pressure transducer with snubber was installed 6 cm above the bed. Figure 6c shows the pressure recorded in the slow-flow borehole (black line). Pressure records from three other boreholes in the same across-glacier line and one sensor downstream are shown in red in the same panel, while the record of a fifth borehole in the same line is shown in panel 6d. Note that there are four virtually indistinguishable records in panel 6c during July 23rd-25th (see also figure 7). After a data gap caused by a corrupted compact flash card, the records have become more dissimilar by August 2nd, but continue to exhibit common pressure variations. The pressure time series from the borehole that is part of the line immediately below the slow-flow hole by contrast has significantly higher mean water pressure and the diurnal

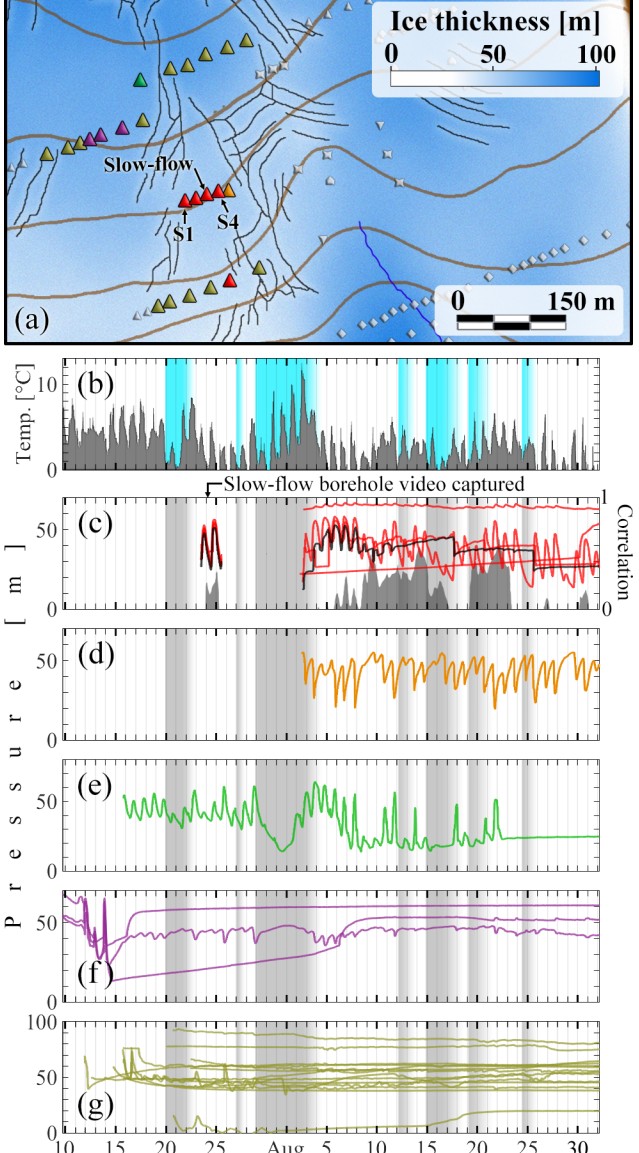

**Figure 6.** Locations and pressure time series for the boreholes associated with the slow-flow borehole on July and August 2014, with the same plotting scheme as Fig. 5 (see corresponding caption). Panel c shows pressure in the slow-flow borehole (black) and three other boreholes in the same line. The correlation with temperature has been calculated using the only borehole that remains connected over the whole interval. The remaining panels show pressure time series from other nearby holes as indicated by the line and borehole marker colours. The time series from boreholes S1–S4 are shown in more detail in Fig. 7.

.

pressure variations have a much smaller amplitude. We have included it in panel 6c because it is the only one in that lower line that matches one of the other pressure records in panel 6c well, if we remove their means and scale them to have unit variance.

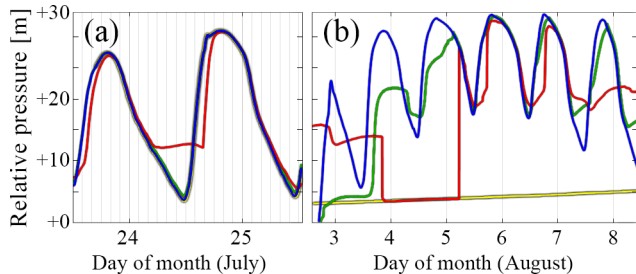

**Figure 7.** Pressure records from the four sensors marked S1–S4 in 6b in July (a) and August (b) 2014. Colour coding is red (S1), yellow (S2), green (S3) and blue (S4). We have applied a constant value offset in pressure to each time series (meaning, added a constant to the directly measured pressure) to make the agreement between the records clearer. The offset values are, in order, 27, 26, 24, and 29 metres in (a), and 27, 20, 22, and 27 in (b). Note that the S2–S4 time series in panel (a) agree so well with each other that they are barely distinguishable.

Most boreholes showing diurnal pressure oscillations share the general features displayed by the slow-flow borehole, specifically 1) smooth pressure peaks and troughs, 2) pressure patterns well differentiated from the atmospheric temperature pattern, 3) mean pressures during periods with diurnal oscillations that lie between 55-120% of the overburden ice pressure (much higher than in the fast-flow hole), 4) peak pressures that typically lag peak temperatures by 2-8 hours and 5) patterns of temporal pressure variations that are often similar to neighbouring boreholes both in the along- and across-glacier direction.

On average, during summer, 71% of sensors showed the behaviour observed in the slow-flow borehole at some point, as assessed visually from the presence of smooth diurnal pressure oscillations. Only 8 boreholes (3% of the total, shown as red markers in Fig. 2) exhibited water pressures dropping to atmospheric pressure, one of the key characteristics of the fast-flow borehole. Six of them were found during the three years with the highest cumulative positive degree day count in the dataset (2013: 437 °C days, 2009: 386 °C days, 2015: 297 °C days).

These figures may however not be representative as drilling was concentrated in some areas. For this reason, we have selected 70 boreholes in two across-glacier profiles (blue markers in Fig. 2). Among those, 81% show a behaviour qualitatively similar to the slow-flow borehole, and 4% that of the fast-flow one. Note however that even these statistics remain biased, as borehole spacing along these lines is concentrated in areas that were of interest due to likely drainage activity, and crevassed areas are under-represented as sensor signal cables typically have a short life span there.

We emphasise that the borehole in which fast flow was observed initially displayed a relatively smooth diurnal cycle, and the statistics above are based on the identification of diurnal pressure oscillations reaching atmospheric pressure at

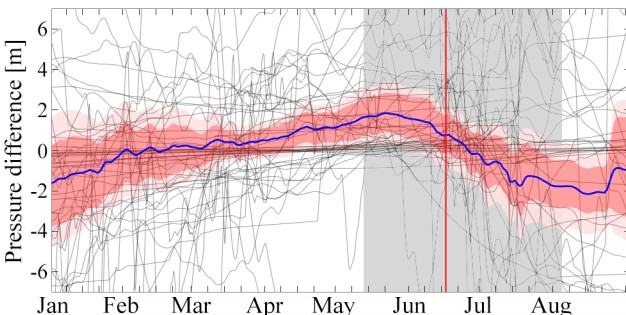

**Figure 8.** Mean water pressure computed over a 1-day running window for each of the 55 sensors that did not display diurnal oscillations during 2016, shown in black. For legibility, we have subtracted the mean over all sensors and the time window shown. The blue line shows the mean over all the black lines at a given time (i.e. over all the sensors) and the bootstrap confidence intervals (Efron and Tibshirani, 1993) of 90 and 99% (dark and light pink shading, respectively). Gray shading represent the period over which the initiation of diurnal oscillations was observed in connected boreholes, and the red vertical line is the median time at which diurnal oscillations first appeared in the 70 boreholes that did experience such oscillations during 2016.

night: it is therefore possible that more boreholes intersect conduits with fast-flowing water, without the observed pressure records indicating as much.

The remaining 26% (or 15% in the two cross-glacier lines in Fig. 2) of boreholes do not show any significant diurnal pressure oscillations at any point during the year. These "disconnected" boreholes usually show year-round mean pressures between 90-120% of ice overburden. Disconnected boreholes frequently show a near-constant pressure signal, but not always, with some exhibiting difficult-to-interpret temporal variability. In 2016 there were 55 disconnected boreholes, allowing us to treat their behaviour statistically. Despite only slight differences in mean pressures between winter and summer there is, however, a slow but statistically significant decrease in water pressure during summer, starting around the spring event and amounting to about 6% of the overburden pressure in total (Fig. 8).

## 3.2 Spatial patterns in water pressure variations

When the whole dataset is viewed over a given time window during summer, it is often possible to identify multiple clusters of boreholes, each exhibiting a specific pattern of temporal pressure variations. Often, these patterns are defined by the way in which the amplitude of diurnal oscillations changes over time. While boreholes in a given clusters will share the pattern of temporal variability, this will differ significantly from the pattern of temporal variability in the other clusters.

One example of this phenomenon comes from the boreholes in Fig. 5f, where we can see a group of boreholes that

display a very coherent signal but with a distinctive two-day period. However, those boreholes in figure 5f are directly adjacent to those in 5e. The latter by contrast show a very different pattern of diurnal pressure variations (that we have associated with the fast-flow borehole, along with panels 5c and 5g).

Less clear-cut, though indicative of the same phenomenon is Fig. 6, where we see boreholes in panels d-f that exhibit quite different diurnal pressure variations from those observed in panel c (the group associated with the slow-flow borehole). Figure 3 of Schoof et al. (2014) also shows an example of the same phenomenon during July and August 2011: borehole B in that figure is, in fact, one of a group of 5 that exhibit almost identical diurnal water pressure oscillations that are quite distinct from those in boreholes A1–A6 in the same figure.

The spatial patterning of the drainage system into distinct clusters becomes much clearer when a dense borehole array with good spatial coverage is available. During the summer of 2015, there were 88 boreholes with active sensors on the plateau indicated in Fig. 2, and 66 further downstream. The corresponding pressure records are presented in Fig. 9. Between June 26th and August 27th, 42 boreholes on the plateau (panel 9c) and 11 boreholes downstream (panel 9d) showed a highly coherent pressure signal that was qualitatively different from the atmospheric temperature signal (panel 9b) and the majority of other boreholes in the two areas (panel 9f). There was no consistent time lag between sensors in the plateau and downstream. However, there was a clear drop in amplitude of diurnal oscillations (panels 9c and 9d), where the latter showed amplitudes around 15-30% of those seen in the plateau.

Five of the sensors on the plateau were capable of conductivity measurements (panel 9g). We emphasise that, in general, the spatial patterning was recognisable only in the pressure records, and pressure oscillations were not associated with conductivity changes. Although all five sensors showed very similar temporal variations in pressure (panel 9c), the conductivity time series bear far less similarity to each other, with only a handful of abrupt conductivity changes common to three of the sensors (S2 and S4 marked in panel 9g).

The group of 14 boreholes in panel 9e also shares common diurnal pressure variation patterns, though this is not immediately clear as the mean pressures and amplitude of pressure variations varies significantly. For that reason we have highlighted in black one line that shows these variations clearly. Notably, these variations are "inverted" versions of the pressure variations seen in panel 9b, with peaks becoming troughs and vice versa. These anti-correlated boreholes, in contrast to those in panels 9c and 9d, have smaller diurnal oscillations amplitudes, and the oscillations are superimposed on a signal with near-constant running mean and high mean pressure, usually close to overburden. Therefore, if diurnal variations were filtered out, these pressure records would resemble the winter regime. At 15 metres spacing be-

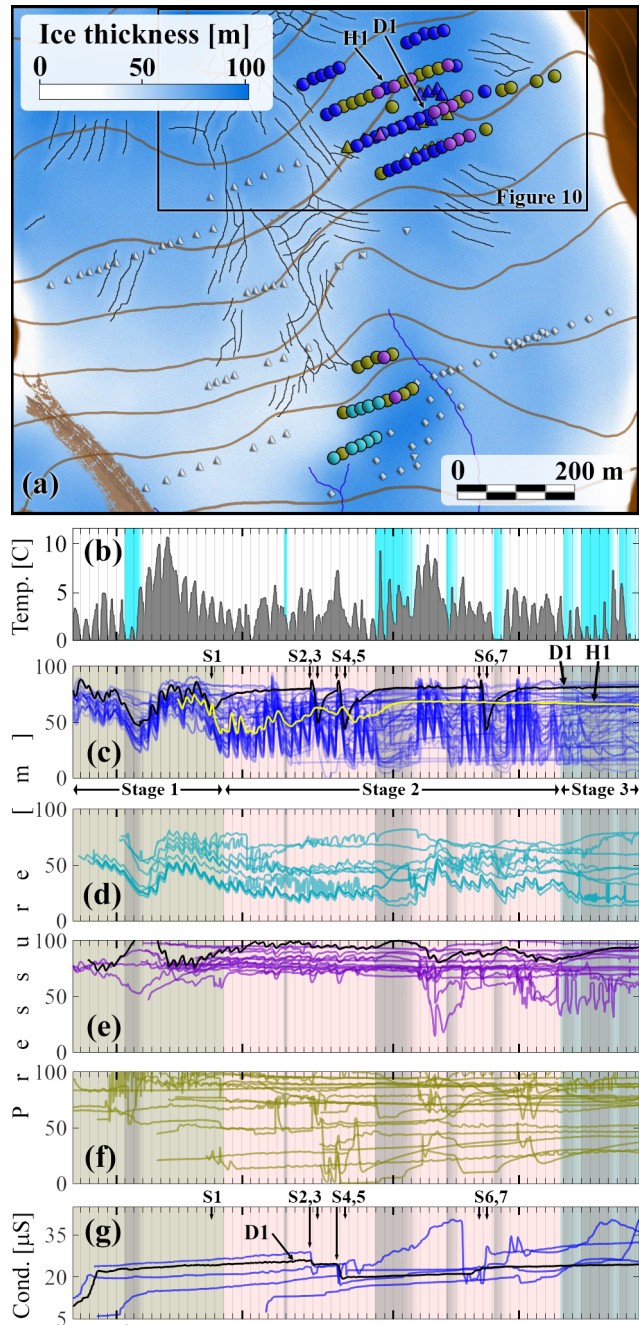

**Figure 9.** Locations and pressure time series for all 82 boreholes on the plateau area and 20 boreholes down-glacier during June–August 2015, plotted using the same scheme as Fig. 5. (b) Temperatures (grey) and fresh snow cover (light blue), (c) Pressure in 42 boreholes on the plateau that shared similar water pressure variations. The highlighted time series are from boreholes D1 (black) and H1 (yellow). Both boreholes are indicated on the map. S1–S7 indicate 'switching events' in the D1 record (see section 3.4). (d) Boreholes downstream of the plateau showing similar pressure variation to those in (b). (e) Pressure records that are anti-correlated to those on panels (c)–(d). (f) Pressure records from the remaining boreholes on the map, (g) Conductivity records from the six digital sensors included in panel c.

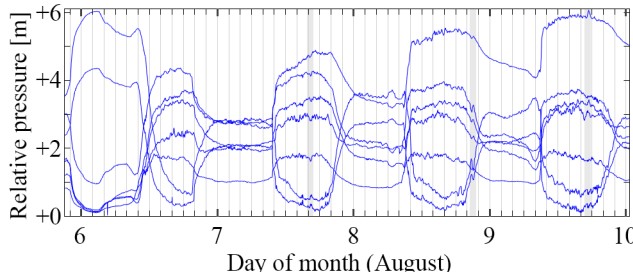

**Figure 10.** Relative pressure variations in 7 boreholes (orange symbols in Fig. 2) displaying common small amplitude diurnal oscillations with high-frequency content during August 2011. To make these visible in the same plot, we have applied offsets of 93, 71, 64, 61, 63, 66 and 27 metres to the measured pressures. Three common high-frequency features are highlighted by grey vertical bands.

tween boreholes, we do not observe sequences of boreholes smoothly transitioning from correlated to anti-correlated, in the sense that there appears to be no continuous change in phase and amplitude from borehole to borehole: we observe a sharp boundary between correlated and anti-correlated boreholes, or correlated boreholes and boreholes exhibiting no diurnal oscillations. Note that one of the records in panel f of Fig. 6 also anti-correlates strongly with the record in panel 6e during the later part of the time window shown, and the record in panel 6d anti-correlates strongly with the record from the adjacent borehole S4 during August; anti-correlation of this kind is a common feature of the dataset, often but not always involving boreholes in close proximity.

There is typically another set of boreholes that show very similar diurnal variations in water pressure super-imposed on near-constant or slowly changing diurnal running mean values. The diurnal variations for this set have very small amplitude (typically 0.2–0.6 m, exceptionally up to 6 m), and resemble a square-wave with super-imposed high-frequency variations. Matching oscillations can be observed in multiple boreholes spread over large distances, both along and across the glacier, and both diurnal and much higher frequency features in the pressure signal are preserved between these boreholes. An example from 2011 involving boreholes across the width of the study area, and recorded by different data loggers, is shown in Fig. 10. Clearly, the oscillations can be both correlated or anti-correlated with each other. Not shown in Fig. 10 is the longer-term evolution of water pressure in the same boreholes. While they share short-time-scale temporal variability, their long-term pressure variations are generally not well correlated.

## 3.3   Three-dimensional drainage structures

The pressure observations primarily give us a two-dimensional picture of the drainage system. The drilling process itself as well as borehole camera investigation provides additional information on englacial connections (Fountain

et al., 2005; Harper et al., 2010). 37% of all the boreholes drained completely or partially during the drilling process, as did 39% of those in the cross-glacier lines marked as blue symbols in Fig. 2. For simplicity, we will give statistics for the entire dataset in running text below, and the corresponding figure for the cross-glacier lines in parentheses. Of the boreholes that drained during drilling, only 14% (0%) drained when reaching the bed, and the remaining 86% (100%) drained at some point during the drilling process, suggesting connections to englacial conduits or voids. Such connections were also observed on multiple occasions using the borehole camera. Drainage events occurred at all depths during drilling, but with a slight preference for greater depths, with 60% (59%) happening in the lower half of the boreholes. This remains true for the 2012 drilling campaign, where the first sensors were installed before the spring event and observations are likely to reflect winter conditions. Unfortunately, water level change and duration of drainage events were not recorded.

During the borehole re-freezing process, 29% (11%) of the boreholes showed a pressure spike typically about 1.3 times overburden pressure, suggesting that freezing happened in a confined space. In total, 62% (73%) of these initially confined boreholes showed diurnal oscillations during the first week, suggesting that some degree of connection was developed with a drainage system experiencing diurnally varying water input.

In 2014 and 2015, three one-year-old boreholes were re-drilled, and the sensors were recovered (boreholes A, B and C in Fig. 2). During this process, we found that holes A and C had sections about 8–12 m long near the bed that had remained unfrozen for the entire year, suggesting that boreholes, as well as natural englacial conduits close to the bed, could remain open through the winter. In borehole A, contact with the bed had erroneously been assumed after the initial drilling based on highly turbid water. However, borehole video footage taken after re-drilling showed that the original borehole had terminated at an isolated rock. From the depth of nearby boreholes, we estimate that the sensor was installed approximately 4 m above the bed. Nonetheless, the diurnal water pressure oscillations recorded in borehole A continued to mimic other nearby bed-terminating boreholes that were drilled in 2014 and 2015, indicating a persistent connection. Fig. 11i shows the pressure record in borehole A and the point at which the re-drilling took place.

### 3.4 Seasonal development of the subglacial drainage system

We have described the apparent spatial patterning of the drainage system above. This patterning is however not fixed but evolves over time. In Fig. 9c, it is clear that all 42 boreholes show very coherent temporal pressure variations at the start of the observation period. During late July and August, the pressure variations in some of the boreholes become more distinct until, by late August, there is no longer a common signal and all boreholes show dissimilar temporal pressure variations.

In Fig. 9c, this emerging patterning is evident only in the more disordered appearance of the plot for later times. In Fig. 11, we show the 42 boreholes of Fig. 9b separated into sub-groups. Within these groupings, it is clear that boreholes can switch from having closely correlated pressure records to behaving independently and, less frequently, to being strongly correlated again (panels d–h in particular); For simplicity, we refer to boreholes as being "connected" while they exhibit the same temporal pattern of pressure variations, and as "disconnected" otherwise. For each grouping, we have computed a mean pressure displayed in black, including only the boreholes that are connected at a given time; in some cases, no boreholes were connected to each other, and we used the last borehole still to exhibit diurnal oscillations to define the set of connected boreholes. In that case, the black mean curve obscures the corresponding, coloured borehole pressure time series. These mean curves for each panel are re-plotted in the corresponding borehole marker colour in panel 11c.

The major dichotomy in Fig. 11 is between the groupings in panels d–g and i on one hand and panel k on the other. The distinctions between panels d–g in particular are more subtle, and generally relate to the absence or subdued nature of certain diurnal peaks in them: for instance, panels d–f all show diurnal oscillations on August 1st and 2nd, while the larger group in g does not; there are other examples close to the end of the summer season. For all groupings in panels d–g, it appears that the early season records resemble each other more closely than those late in the season, as was already evident in Fig. 9b, and that fewer boreholes have disconnected early in the season.

At a much smaller scale, a similar fragmentation of the drainage system is shown in Figs. 6b and 7, where we see four boreholes that are initially very well connected during late July having become much less well connected in August, although with the diurnal pressure oscillations still showing some similarities between several of the boreholes. Interestingly, one borehole (S2, yellow) has ceased to exhibit oscillations by August, but is straddled by two that still do (S1 and S3, 15 m to either side), suggesting a relatively fine-scale structure to the drainage system locally.

In addition to spatial patterning, Figs. 9 and 11 hint at an overall evolution towards lower mean water pressures and larger diurnal oscillations. The seasonal evolution of the drainage system may be evident not only in its spatial extent, but also in the evolution of mean water pressure and its response to surface melt input. Perhaps the simplest measure of sensitivity to surface melt input is what we term the relative amplitude of pressure to temperature oscillations: we compute standard deviations of pressure time series from boreholes that exhibit diurnal oscillations at some point of the season, and also standard deviations of positive air temperatures (the maximum of air temperature and 0 °C). We com-

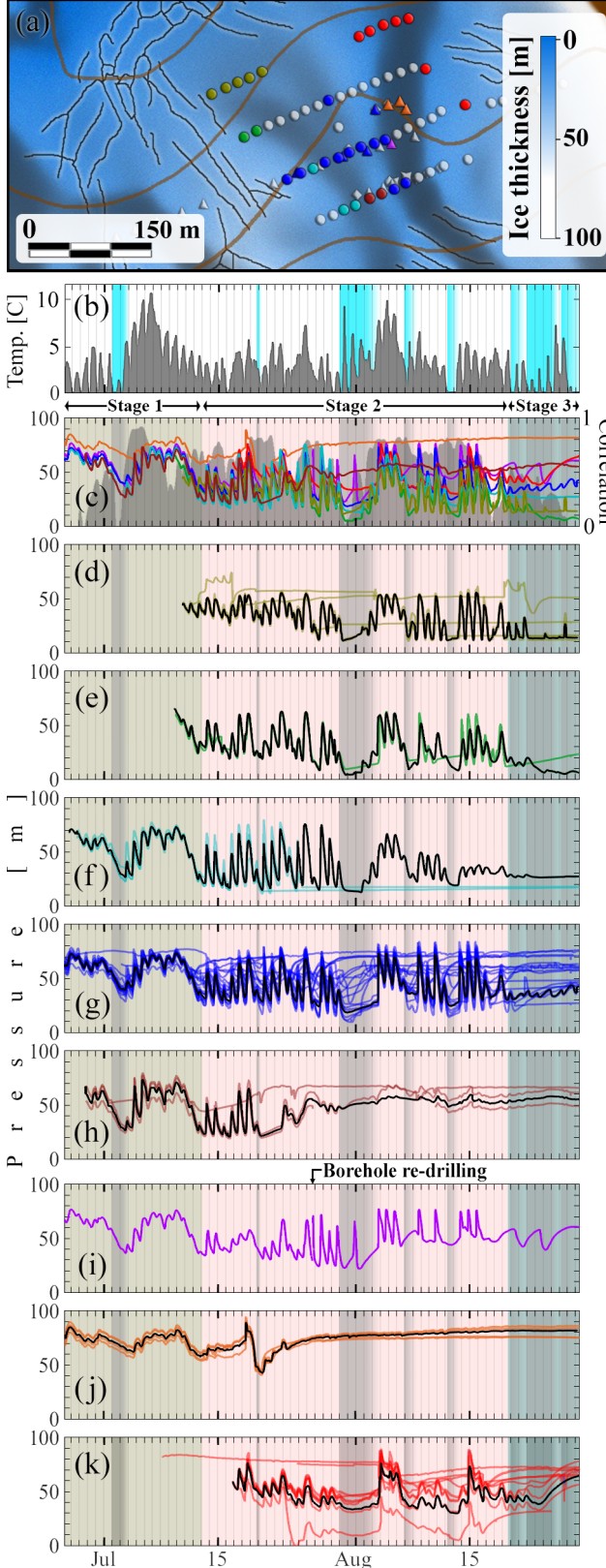

**Figure 11.** Locations and pressure time series for all 42 boreholes shown in Fig. 9b, grouped according to the similarities between their diurnal pressure variations in July–August 2015, same plot-

ting scheme as in Fig. 5. (a) The map area is indicated as a box in figure 9a. Grey shading is the upstream area shown in Fig. 2 (b) Air temperature (grey) and fresh snow cover (light blue). (d–k) borehole pressure time series, colour of plots corresponds to colour of borehole marker on the map. Black lines are the mean pressure in each panel, computed only over those boreholes that are "connected" at a given time (see main text); the black lines frequently obscure one of the borehole time series. (c) The black "mean" curves from panels (d–k), plotted in the corresponding borehole marker colour. The maximum cross-correlation coefficients, allowing for time lags of up to six hours, between all pressure records and air temperature computed over a moving three-day window is shown in dark grey.

pute these standard deviations over one-day running windows, and define the ratio of the two running standard deviations to be the relative amplitude of pressure to temperature variations. Taking the running standard deviation of air temperature as a marker of surface melt rate variability (see section 2 in the supplementary material for a discussion about this assumption), the relative amplitude defined in this way gives an indication of how sensitive the drainage system is to variations in water input.

In Fig. 12, we see that the running standard deviation in pressure only vaguely tracks its temperature counterpart. However, the relative amplitude systematically increases during much of the season (Fig. 12c), except during an interval of colder weather and surface snow around the beginning of August, while the mean water pressure also decreases (Fig. 12d).

## 3.5 Basal hydrology transitions and "Switching events"

Above, we have seen that boreholes can become disconnected from each other, going from a state in which they undergo synchronous and virtually identical pressure variations over time to a state in which borehole pressure appears to evolve independently. The reverse change also happens, though less frequently (except during the spring event). The change from connected to disconnected and its reverse can take different forms. In a few cases, disconnection is gradual, with the boreholes continuing to exhibit similar diurnal pressure oscillations that progressively become more dissimilar in amplitude, phase and mean water pressure. The record from H1 in Fig. 9c (yellow line in panel c) is one such example. In most cases, however, the transition is abrupt, and the same is true of boreholes connecting with each other: a rapid change in water pressure can occur over the course of a few hours or less as a connection is established. We term such abrupt transitions "switching events", following Kavanaugh and Clarke (2000).

Figure 7 shows multiple examples in the boreholes labelled S1–S4 in Fig. 6c, spaced 15 m apart. Perhaps unsurprisingly, the majority of switching events involving new connections seem to occur while water pressure is increasing or after a recent increase, while disconnections tend to

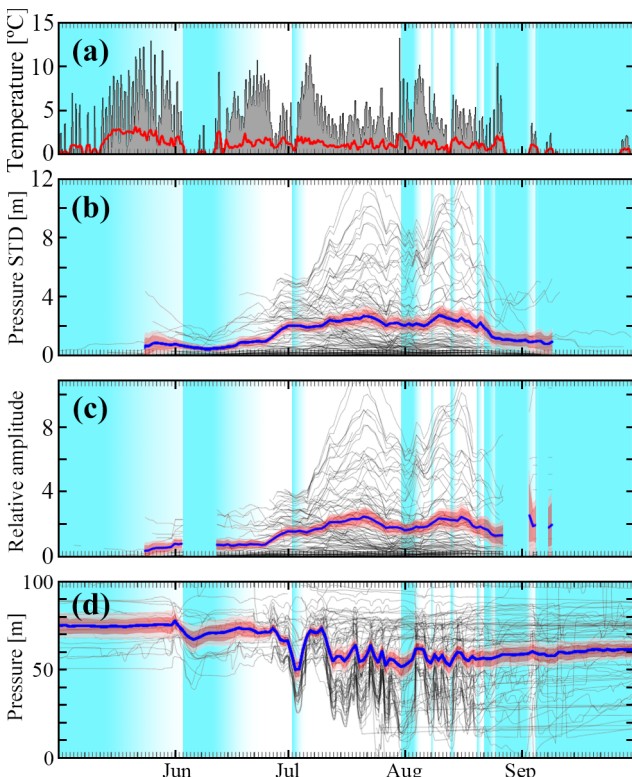

Figure 12. Relative amplitudes of pressure and temperature diurnal oscillations from May to September 2015. (a) Positive air temperature (grey), and its standard deviation over a 1-day running window (red). (b) Standard deviation of pressure over a 1-day running window (thin black lines) for each borehole in the plateau area, and the mean of these standard deviations (blue) with bootstrap confidence intervals of 90 % (dark pink) and 99% (light pink). (c) Ratio between pressure and temperature standard deviations shown in (a) and (b), computed only where standard deviation of air temperature is non-zero. (d) Mean pressure computed over a 1-day running window. Light blue shading represents fresh snow on the glacier surface.

occur as water pressure is falling (see Fig. 11c,d,f for several obvious examples), though the two are rarely symmetric, with disconnection usually occurring at a lower water pressure than the original connection. The record from sensor D1 in Fig. 9c is one such example, where arrows labelled S1–S7 mark multiple switching events. The borehole originally disconnected from the main group on July 11th, but reconnects on several occasions during periods of high water pressure in the active drainage system, disconnecting when water pressure subsequently drops. Note that for the first two reconnections, S2 on July 21st and S4 on July 24th, the switching events are clearly associated with large drops in conductivity as seen in panel g, suggesting an inflow of meltwater that has spent less time in contact with the bed (Oldenborger et al., 2002).

Towards the end of the melt season, most boreholes have become disconnected from each other, and water pressure

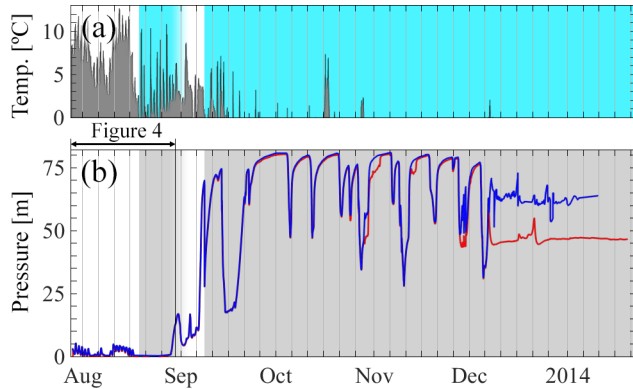

Figure 13. Extended pressure time series from the 2013 fast-flow borehole (Fig. 5). (a) air temperature (grey) and fresh snow cover (light blue). (b) Pressure recorded by two sensors installed 10 cm and 80 cm above the bed (red and blue respectively).

in them typically rises again towards overburden, remaining nearly constant through the winter. However, in some cases, we observe quasi-periodic pressure variations in winter as previously reported in Schoof et al. (2014). Figure 13 shows the winter pressure record for the two sensors installed in the fast-flow borehole, extending the summer record shown in Fig. 5c. As in other boreholes, we see water pressures rising at the end of the summer season. This is briefly interrupted during early September, when surface snow cover temporarily disappears, and a drop in water pressure occurs in the borehole, accompanied by the resumption of diurnal oscillations. This is followed once again by the termination of diurnal oscillations and a sharp rise in water pressure towards overburden once the surface becomes snow-covered again. Unlike in most other boreholes, that rise towards overburden is however interrupted by oscillations lasting from 2–12 days. During these oscillations, water pressure can drop rapidly to as little as a quarter of the overburden, followed by a slower rise in pressure back towards overburden, stabilisation, and a renewed rapid drop.

## 3.6 Inter-annual variability

As observed in Fig. 4, there is large inter-annual variability in positive air temperatures and hence, presumably, in surface melting, both in terms of onset and intensity. In addition, we expect that differences in the snow-pack can also affect water delivery to the englacial system; presumably, a thicker snow-pack can store or refreeze surface meltwater, and leads to higher average surface albedo during the summer. Figure 3 shows a view of the study area from an automated camera on July 19th in 2012–2015. These images illustrate very significant differences in surface snow cover at the height of the summer melt season: the visible snow cover in each image is part of the remaining winter snowpack.

Alongside the inter-annual variability in temperature and snow cover, there are also significant season-to-season differences in the water pressure records. Differences in drilling objectives from season to season make year-to-year comparisons difficult except in one part of the study area. Figure 14 shows a compilation of pressure records from a set of boreholes drilled in almost the same locations every year in the lower plateau area from 2012 to 2015, as indicated by two red polygons in Fig. 2. There are four boreholes (surrounding the fast-flow hole in Fig. 5) included here for 2013–2015 that were not drilled in 2012, and four that were drilled in 2012 but not in later years. Alongside air temperatures and snow cover, we also indicate the total PDD count prior to June 15th, and at the end of September. Also shown are the median of the dates on which diurnal oscillations appear and disappear over all boreholes with functioning sensors (red lines). The latter are clearly a crude measure of drainage system evolution as they are biased by borehole locations and drilling dates. Despite these differences, the borehole pressure records clearly indicate some systematic differences, with a relative absence of diurnal pressure oscillations in 2012 and 2013, though accompanied by very low water pressures associated with the fast-flow borehole in 2013 (see also Fig. 5), and a larger number of "connected" boreholes with large-amplitude diurnal oscillations in 2015. We also note that we drilled new boreholes in 2014–2015 in the location of the 2013 fast-flow hole without encountering more evidence of turbulent water flow.

## 4  Discussion

The seasonal evolution of the drainage system we observe is broadly consistent with existing ideas about drainage physics. A drainage system forms annually, triggered abruptly by the delivery of meltwater to the bed in a "spring event" (Iken and Bindschadler, 1986). The timing of the spring event varies significantly from year to year, taking place when most of the glacier surface is still snow covered, but always after the appearance of the first sizable snow-free patches (Fig. 3, see also Nienow et al. (1998b)). This suggests that the development of drainage pathways through the surface snow cover is a precursor to water delivery to the bed, with the timing most likely dictated by snow depth, temperatures, and early season melt rates (see also Harper et al., 2005). Additionally, the appearance of bare ice will significantly lower albedo, and could lead to a significant increase in melt production. After the spring event, most boreholes show strongly correlated diurnal pressure variations, suggesting extensive hydraulic connections, and at least a slight drop in water pressure. However, when compared with late-season diurnal pressure fluctuations, these early season pressure oscillations have smaller amplitudes and lower correlation with the inferred surface melt rates, suggesting a relatively inefficient drainage system. We will refer to this initial state of the

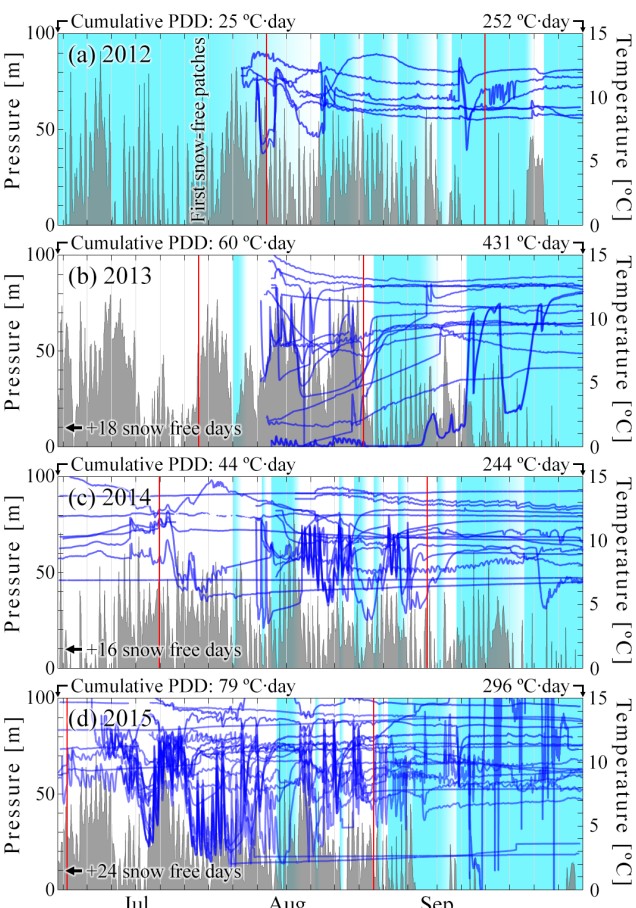

**Figure 14.** Overview of pressure variations on the lower portion of the plateau area from 2012 to 2015. Each panel includes air temperature (grey), coverage of fresh snow (light blue), and vertical red lines displaying the median date of initiation and termination of diurnal oscillations on all active sensors each year. Cumulative positive degree days are displayed for the beginning and end of the interval shown.

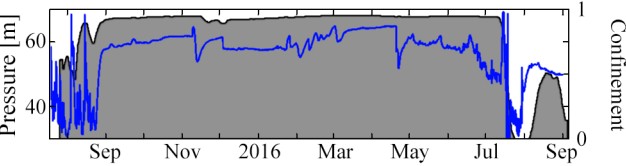

**Figure 15.** Confinement data (grey) and pressure (blue) for one of the digital sensors in Fig. 9c from July 2015 to September 2016.

subglacial drainage system as stage 1. Note that the "stages" identified here are not the same as the "phases" discussed in Schoof et al. (2014), who focused only on the later part of the melt season and the subsequent winter; for instance, phase 2 in Schoof et al. (2014) corresponds to the transition from stage 2 to 3 here.

Later in the season, the drainage system becomes more focused, in what we will call stage 2. During this stage, the

mean water pressure in the system drops, and the magnitude of diurnal pressure variations increases (see Fig. 12, also Harper et al. (2002)). Different parts of bed still exhibit diurnal oscillations but cease to be mutually well-connected, as also observed by Fudge et al. (2008). We will refer to the parts of the bed that remain internally well-connected as hydraulic subsystems (Fig. 5 panels f and g, Fig. 6 panels c and f, and Fig. 11 panels d–h, j and k are examples of this behaviour, with panels d–g in the latter sharing many features but appearing quite distinct from panel k). Subsystems progressively shrink, shutting down drainage over an increasing fraction of the bed. At most boreholes, the drainage shutdown is marked first by the sudden disappearance of diurnal cycles in a switching event, often followed by a sustained increase in pressure that takes approximately one to a few weeks to stabilise at a value close to overburden.

In high-melt years, the fragmentation of the drainage system can be extreme. Figure 5 shows only a handful of boreholes exhibiting diurnal oscillations towards the end of stage 2. Our data suggest these boreholes may align with down-glacier drainage axes. Had we sampled the glacier bed differently, we could have had no boreholes showing diurnal oscillations during this period. This widespread drainage shutdown around highly focused drainage subsystems would explain why the end of diurnal oscillations in most boreholes precedes the decline in inferred meltwater supply and proglacial river runoff as observed by Harper et al. (2002) and Fudge et al. (2005).

The distinct response of different subsystems to the same surface conditions must be the result of peculiarities of each subsystem. The amplitude of the diurnal pressure cycle typically varies over periods of several days, but the temporal pattern of amplitude variations differs between subsystems, and generally, does not reproduce corresponding variations in diurnal melt amplitude (Fig. 11 here and Fig. 3 of Schoof et al. (2014)).

The systematic increase throughout stages 1 and 2 of the relative amplitude of diurnal pressure and inferred melt oscillations (Fig. 12), and the correlation with positive temperatures (Figs. 5 and 11) is consistent with an increase in the drainage system efficiency.

A widespread termination of diurnal oscillations in the remaining connected holes is typically triggered by a marked drop in meltwater supply, usually coincident with a snowfall event. We label this as the start of stage 3 in Figs. 5 and 9, 11; in Fudge et al. (2008), this is referred to as the "fall event" (though their data makes connections with snowfall less easy to establish). The termination of diurnal oscillations is often followed by a rise in borehole pressures towards overburden, marking the beginning of the winter pressure regime, where pressure variations are no longer closely correlated, suggesting an absence of hydraulic connections.

The shrinking and fragmentation process during stage 2, and possibly the onset of stage 3, may, however, be partially reversed by brief episodes in which the reconnection of at least some boreholes is observed. These reconnection episodes are often associated with strong increases in meltwater supply, usually on hot days when temporary snow cover clears. During 2015, snow events during late July and early August led to several episodes in which most boreholes shown in Fig. 9 appeared to disconnect from each other, pressures in them not only ceasing to exhibit diurnal oscillations but also evolving independently. These episodes ended with surface snow cover disappearing and melt supply resuming, leading to widespread and often abrupt reconnection at high basal water pressures.

Similarly, we cannot exclude the possibility that highly focused drainage subsystems remain open during the early parts of stage 3: the borehole array cannot sample all conduits directly, and we are only certain of having intersected a main conduit in one instance. That conduit, the 2013 "fast-flow" borehole, remained at close to atmospheric pressure for nine days at the start of stage 3 (Fig. 5). Subsequently, water pressure started to rise, but even then, the disappearance of snow cover and continued melting led to a pressure drop and renewed diurnal pressure oscillations correlated with surface temperatures from September 1st to 6th (Fig. 13).

We have referred to snow cover on the glacier being a good indicator of a drop in water supply to the bed. Often this snow cover persists for a period of days in positive temperatures. With the data we have, we cannot state unequivocally whether the reduction in water supply is primarily due to the high albedo of snow suppressing melt, or due to water retention in the snowpack.

The spatial evolution of the drainage system is consistent with the drainage system becoming channelised during the melt season. By this, we mean the formation of individual Röthlisberger-type ("R") channels, incised into the base of the ice by dissipation-driven melting (Röthlisberger, 1972). Formation of channels should cause the mean water pressure to drop, as the focusing of water discharge causes larger channel wall melt rates that have to be offset by faster creep closure, driven by larger effective pressures (Nye, 1976; Spring and Hutter, 1982). It can also account for the increased sensitivity of the pressure response to the inferred melt input, and the reduction of dye tracer transit time observed in other glaciers (Nienow, 1993).

The clustering of boreholes into drainage subsystems indicates good hydraulic connections between them. However, as channels cannot coexist stably in close proximity (e.g. Schoof, 2010; Hewitt, 2011), it is unlikely that all boreholes that sample the same drainage subsystem are located in R-channels, or in an R-channel at all. A more obvious explanation is that in stage 2, each independent subsystem contains a channel surrounded by a distributed drainage system consisting of linked cavities or a similar conduit configuration (Kamb, 1987; Fowler, 1987; Hubbard et al., 1995; Schoof, 2010; Hewitt, 2011). Such a distributed system is consistent with the observation of slow-moving water in the 2014 slow-flow borehole. In addition, the existence of nar-

row R-channels within those systems is also consistent with the finding of the 2013 fast-flow borehole in stage 2.

Pressure records alone are insufficient to determine if there is water flow and whether a sensor is in a channel or a distributed system, even if the distributed system is hydraulically well-connected. The pressure record shown in Fig. 5 is the one record of which we know that it almost certainly reflects pressure variations in a channel. We know that highly turbulent flow occurred in the bottom 50 cm of the borehole, which we take to be the height of the channel, but its width is unknown. The first week of that time series resembles the smooth pressure variations observed in many other boreholes (albeit at fairly low water pressures), while it develops very distinct features later: water pressure drops to atmospheric at night, and there are unusually small time lags relative to and very high correlation with inferred surface melt rates.

The 2013 fast-flow borehole does not connect hydraulically to other nearby ones that lie along an across-glacier line (Fig. 5d), but appears to connect to a narrow set of boreholes extending 500 m downslope (Fig. 5 e and g). The pressure time series from those boreholes differ somewhat from that measured in the channel, so there is probably a narrow distributed system close to the channel, the width of that system being less than the 15 m borehole spacing.

These observations are consistent with a highly developed channel with higher water discharge that has become hydraulically isolated from the neighbouring bed: the high effective pressures in the channel would favour the closure of cavities or other connections in the surrounding bed. This closure may also be enhanced due to the effect of bridging stresses (Lappegard et al., 2006). Bridging stresses transfer part of the weight of the ice overlying the channel to its surrounding bed, effectively increasing the ice overburden in those regions above its mean value (Weertman, 1972).

The 2013 season was marked by high net inferred melt: the total PDD at the end of that season exceeded the PDD for 2014 and 2015 by 46% or more (Figs. 4 and 14). The high inferred melt rates are consistent with channelisation reaching an end-member state. The rapid flow of water in the borehole also made identification easier; it is unclear if a smaller channel would have been as easy to identify.

Using the channel end-member feature of diurnal oscillations with pressure dropping to atmospheric at night, we have identified seven other boreholes where the drainage system is likely to have evolved into a well-developed channel (Fig. 2, red symbols), in all cases during the second half of July or first days of August during years with relatively high cumulative PDD, which ought to favour channel formation. Their locations loosely match zones with high up-stream areas (Fig 2, dark shading), which correspond to portions of the bed likely to concentrate basal water flow due to the expected hydraulic gradients.

Late in the season, the shut-down of the now well-developed basal drainage system during a period of dwindling inferred melt supply is consistent with high effective pressures causing the closure of subglacial connections, especially as disconnection events often occur at low observed water pressures. Different boreholes appear to become hydraulically isolated from each other during this process. We interpret the subsequent evolution of pressure records after disconnection as reflecting the response of an isolated water pocket in the borehole, presumably containing a fixed (or nearly fixed) volume of water exposed to the ambient stress field. Initially, creep closure will reduce any volume still occupied by air in the borehole and pressure can rise gradually; once there is no air space left, changes in water pressure must reflect the pressure required to maintain the borehole volume constant (assuming no further freezing) while the borehole may still deform under anisotropic stress conditions (Meierbachtol et al., 2016). Intuitively, we would expect the borehole to become flattened perpendicular to the direction of greatest compressive stress, requiring a larger borehole pressure to maintain a constant volume, which could account for the slow rise observed in water pressure, and possible for slightly above-overburden values. Importantly, the pressure in an disconnected borehole should depend on its shape and can, therefore, differ from borehole to borehole; abrupt creation of new storage volume for instance due to crevasse propagation could also lead to abrupt changes in pressure in disconnected boreholes. Therefore, we have to caution against interpreting the pressure in individual disconnected boreholes as an indication of the conditions in the unconnected parts of the bed: instead the borehole pressure may be controlled predominantly by local stresses in the ice, and the orientation, volume and shape of the unfrozen portion of the borehole.

During winter, a handful of boreholes exhibited large-scale quasi-periodic pressure oscillations as detailed in Schoof et al. (2014) and shown in Fig. 13. We have previously hypothesised that these multi-day winter oscillations indicate ongoing drainage in a few locations, with the oscillations driven by the interaction between conduit growth and distributed water storage in smaller water pockets, basal crevasses and moulins; such oscillations could be triggered when water supply drops below a critical value in combination with a steady background water supply (Schoof et al., 2014). Winter oscillations are common in boreholes that showed end-member channel behaviour at the end of the summer, as is the case for our 2013 fast-flow borehole shown in Fig. 13, and borehole D in Schoof et al. (2014). However, similar winter oscillations can occur also in boreholes that were disconnected or belonged to a distributed drainage system during the previous summer.

## 4.1 Interannual variability

The timing of spring events and speed at which the evolution of the drainage system occurs appears to be linked systematically to the availability of meltwater. Cool, snowy summers are most obviously linked to a poorly developed drainage

system with weak diurnal cycles (2012) and poor correlations between boreholes, as well as the absence of a sharp spring event (Fig. 14).

The spatial structure of the drainage system also varies from year to year. The plateau area reliably has drainage activity, though upstream area pattern in Fig. 2 does not directly agree with the observed drainage structure (Fig. 11), but is merely suggestive. Channel formation is influenced by pressure gradients controlled by surface and bed topography. However, changes in water supply geometry and the instability inherent in channel growth and competition between emerging channels implies that channels need not form in the same location every year. This is consistent with our failure to find in 2014 and 2015 a channel at the 2013 fast-flow borehole location.

### 4.2 Challenges to current subglacial drainage models

Boreholes do not only disconnect from or reconnect to each other during the summer: a significant number of boreholes never connect at all. Others disconnect from the drainage system as it becomes more focused and fragmented into subsystems during stage 2. Some boreholes even disconnect and reconnect multiple times (Figs. 7, 9 and 11).

There is typically a very clear distinction between connected holes showing a similar response to the diurnal input, and disconnected ones that do not. Within a given drainage subsystem, there is typically no gradual phase shift or diminution of oscillation amplitudes from borehole to borehole, as would be expected if the drainage system were a diffusive system with a finite diffusivity (Hubbard et al., 1995): effectively, our data suggest that, if the distributed system is diffusive, its diffusivity is very high, or zero where the system has become disconnected.

This observation contrasts with the interpretation of data in Hubbard et al. (1995), who identify a gradual phase shift and decay in amplitude of diurnal pressure oscillations away from an inferred subglacial channel location. In our view, the phase lag in their Fig. 5, can however also be interpreted as showing diurnal switching of their borehole 40 from being well-connected with their boreholes 29 and 35 to being disconnected. The latter interpretation would be consistent with Murray and Clarke (1995), and with Fig. 7 here also showing an example of switching events with similar characteristics on South Glacier.

Hubbard et al. (1995), suggest that the bed substrate at their study site is composed of glacial till of varying grain size distributions, acknowledging that "a network of small channels" on a hard bed could also account for their observations. However, in terms of hydrology, till and a distributed drainage system at the ice-bed interface share many characteristics: we expect both to give rise to a diffusive model for water pressure if water storage in the distributed system is an increasing function of water pressure. The primary difference is in how the permeability of that system evolves.

In the 'hard-bed' view, the permeability evolves over time in response to changes in effective pressure, whereas for a granular till, porosity and therefore permeability are simply functions of effective pressure and therefore respond instantly to changes in it (Flowers, 2015). The main inconsistency of appealing to drainage through continuous till layer as the main pathway for water flow is that we would expect to see more standard diffusive behaviour, and certainly no sharp switches between connected and disconnected portions of the bed. In addition, till with a sufficient coarse-grained fraction of cobbles and boulders would probably be capable of supporting the formation of cavities in the lee of those larger grains. In short, if till is capable of creating cavities, or is interspersed with bedrock bumps or somehow capable of supporting switching events by other means, then our interpretation would not be affected by assuming a hard or granular bed.

Pressure measurements at South Glacier therefore suggest that the distributed parts of each drainage subsystem are hydraulically well-connected, with all connected boreholes showing almost identical pressure variations. The limited electrical conductivity and turbidity measurements however also indicate that relatively little water might actually flow in the distributed system (Oldenborger et al., 2002). Unlike in the data in Hubbard et al. (1995), there are no diurnal variations in electrical conductivity. With a hydraulically well-connected system, this has to correspond to a low water storage capacity, where substantial variations in water pressure do not require similarly large changes in stored water. Alternatively, storage capacity could be relatively localised, so that water does not need to flow everywhere. Oldenborger et al. (2002) shows how water pressure variations with no corresponding change in conductivity can be observed over an impermeable bed. However, the proposed mechanism requires the boreholes to be disconnected, and would not operate on hydraulically connected boreholes as in this case.

While there are typically insignificant cross-glacier differences in diurnal pressure response within well-defined drainage subsystems, the same is not true in the down-glacier direction, even where we believe a hydraulic connection can be identified. The pressure time series along the inferred channel system in Fig. 5 (panels c, e and g) are merely suggestive of a hydraulic connection, but hardly identical. The amplitude of pressure variations decreases markedly downstream from the fast-flow borehole, which would be consistent with a diffusive system, though it is unclear whether the change in amplitude occurs along the length of the channel, or within a putative distributed system flanking the channel, since the holes further down-glacier most likely did not sample the channel directly. Importantly, however, there is no systematic phase lag accompanying the decrease in amplitude, as would be predicted by a diffusion model (Hubbard et al., 1995). It is however conceivable that additional water input from surface sources along the flow path can have a significant effect on the phase of the pressure signal.

We have referred to boreholes that cease to exhibit diurnal pressure variations as having disconnected. Connection and disconnection typically manifest themselves very abruptly in time (Fig. 7, see also Fig. 5 of Murray and Clarke (1995)). This transition usually takes from few tens of minutes to a few hours. However, the initiation of the transition, often identified as a clear change in the rate of change of pressure with respect to time, can in many cases have the appearance of an instantaneous phenomenon, even at our shortest sampling interval of one minute. Therefore, it is unclear if these time scales can be associated with the connection or disconnection process, as they might only represent how fast the system responds to a perhaps instantaneous switch between connected and disconnected states.

Usually, disconnection occurs during a drop in water pressure in the subsystem, and reconnection during an increase (Figs. 7 and 9). This is consistent with connection or disconnection resulting from viscous creep closing connections between individual cavities within the distributed system (Kamb, 1987), or presumably with elastic gap opening or closing if sufficiently rapid. Disconnection could also be the result of cavities shrinking while remaining connected, if the borehole simply terminates on an ice-bed contact area between connected cavities and those contact areas are systematically larger than the $\sim 10$ cm diameter of our boreholes. This process has been observed previously by Meierbachtol et al. (2016). It however seems unlikely that this effect, which should be random, would lead to a recognisable spatial structure of narrow drainage regions flanked by increasing large disconnected regions. Instead, we would expect a random distribution of apparently connected and disconnected boreholes.

The anti-correlated signals we observe in our data (Fig. 9e) have previously been explained by a mechanical load transfer mechanism, where the ice around a pressurised conduit redistributes normal load, reducing the normal stress over neighbouring areas of the bed. Unconnected water pockets in those areas would therefore experience a drop in water pressure (Murray and Clarke, 1995; Gordon et al., 1998; Lefeuvre et al., 2015). A three-dimensional Stokes flow model (Lefeuvre et al., 2018) supports this interpretation, and suggest that the anti-correlation pattern depends on the bed slope, which can be one of the factors affecting the observed distribution of borehole displaying this behaviour. Boreholes exhibiting anti-correlated pressures must be effectively disconnected, so that a change in normal stress mainly causes changes in the pressurisation of the borehole rather than water exchange. The load transfer mechanism is consistent with our observations.

An alternative explanation suggests that such signals are associated to enhanced cavity opening due to basal sliding changes (Bartholomaus et al., 2011; Hoffman and Price, 2014; Iken and Truffer, 1997). However, it is unlikely that a variation in sliding would precisely mimic the local water pressure variations in the adjacent drainage subsystem, as

suggested by Fig. 9e: the force balance that determines sliding velocities should be affected by changes in basal shear stress across a larger portion of the bed.

Note that we observe anti-correlated signals in boreholes that are not immediately adjacent to boreholes showing a correlated signal (purple and blue markers in Fig. 9). It would be difficult to explain the anti-correlated signal in these boreholes by normal load transfer over larger distances, when other disconnected boreholes nearby show no such behaviour. This suggests that the connected drainage system can contain fine structure (either as channels or narrow regions of distributed drainage) with lateral extents smaller than the $\sim 15$ m borehole spacing. The same is indicated by the formation of disconnected "islands" in lines of otherwise connected boreholes at the same spacing as seen in Fig. 7 for the August observation period (see also Murray and Clarke (1995), for analogous observations).

We have referred to a borehole as disconnected when observations show that pressure variations on a diurnal time scale are not communicated to a borehole. However, the evolution of the mean water pressure in disconnected boreholes is consistent with a residual amount of water leakage into the connected drainage system: during the summer, that mean pressure gradually decreases. The end of the monotonic increase in water pressure of disconnected boreholes observed in Fig. 8, coincides with the spring event, followed by a slow decrease. The large sample obtained in summer 2016 supports this trend up to a 99% confidence despite the large variability of the observations.

As in Hoffman et al. (2016), such a slow evolution could be accounted for by flow through a relatively impermeable till aquifer underlying a much more effective but less pervasive interfacial drainage system, and the magnitude of that leakage could have a significant impact on basal sliding rates if disconnected areas act as sticky spots.

Widespread hydraulic isolation of the bed in winter is supported by high recorded water pressures and the marked pressure drop at the spring event observed in 20% of boreholes. In contrast, theories based on a remnant "distributed" system would ordinarily suggest relatively low water pressures in winter (Schoof, 2010; Hewitt, 2013). Although it is possible that some boreholes do not connect because they were not properly drilled to the bed, we believe that the existence of persistently disconnected areas is robust. Non-spatially biased samples suggest that up to 15% of the bed could remains unconnected year round. The existence of such unconnected holes, and the possibility of dynamic connection and disconnection, represents a challenge to existing drainage models, which typically assume pervasive connections at the bed.

In addition to conduits at the bed interface, englacial conduits are known to exist inside temperate glaciers (Fountain and Walder, 1998; Nienow et al., 1998b; Harper et al., 2010). However, it is unclear whether they allow mostly vertical water transport, or if horizontal water transport over significant distances is also possible through them. Frequent

drainage events during drilling (also observed by Iken and Bindschadler (1986)) suggest the existence of a large number of englacial conduits, and borehole re-drilling observations show that upward conduits can remain open through the winter season in a layer extending several metres above the bed. However, we have no evidence of significant along-glacier drainage in winter, while we know that englacial connections can remain. This suggests that the englacial connections remain isolated from each other during winter. It is unclear if they can connect in summer and establish an englacial drainage system capable of supporting significant down-glacier drainage. The persistence of conduits through winter is most likely related to the basal layer of temperate ice (Wilson et al., 2013), and hydraulic isolation preventing creep closure. The apparent ubiquity of englacial conduits suggests a need to assess their role in downstream water transport in future; if significant, this represents another area of improvement for drainage models.

### 4.3 Mechanically connected boreholes

Strong correlations over long distances were observed in boreholes displaying all the features of disconnected boreholes except a superimposed low-amplitude diurnal pressure variations with high-frequency variations (Fig. 10). From their wide spatial distribution, it appears impossible for them to be connected by hydraulic conduits. As such conduits would need an extremely high diffusivity to preserve the observed high-frequency features over large distances (>500 m as seen in Fig. 2). Moreover, a high diffusivity is at odds with the diverging evolution of temporally smoothed borehole water pressures in the same holes.

These signals do not seem to be instrumental artifacts, and in many cases were recorded by independent data loggers. We have also considered effects due to induction on non-twisted signal cable coils, temperature, or solar irradiation. However, in those cases, such signals should also be superimposed on records from distributed drainage systems, contrary to our observations. Possible explanations must be related to periodic large-scale stress changes in the ice compressing disconnected boreholes whose volume must remain constant, thereby eliciting an instant water pressure response. The most likely cause of such large-scale stress changes would appear to be the occurrence of periodic diurnal basal slip events as suggested by Andrews et al. (2014).

### 4.4 Data interpretation caveats

We generally assume that the sensors at the bottom of boreholes measure the water pressure at the bed. However, this may not always be the case if the sensor becomes encased in ice, is connected to an englacial conduit, or if the borehole did not reach the bed or has penetrated into the basal till.

It is likely that with time, some sensors can become encased in ice, as suggested by the fact that older sensors are less likely to show diurnal oscillations (for that reason, sensors in old disconnected boreholes were often decommissioned before they ceased to produce a signal), and the observations in doubly-instrumented boreholes (see section 1 of the supplementary material). Digital confinement data suggest that in some cases, as in Fig. 15, the termination/initiation of diurnal oscillations is associated with an increase/decrease in confinement. This observation would also be consistent with ice encapsulation of the sensor during winter.

Although the upper end of the boreholes typically freezes shut within few days, the abundance of englacial conduits opens the possibility that the sensors could connect to an englacial conduit through the lower portion of the borehole while it is still open. In such a case, the pressure record could at least partially reflect the evolution of englacial conduits instead of the basal drainage system.

Alternatively, in the absence of englacial connections, a sensor in a borehole that fell short of the bed would appear as disconnected, even if the underlying bed is not. However, we believe this is not a common situation due to the strict procedures followed to assess whether a borehole reached the bed or not (see section 2).

Observations by Hart et al. (2015) using wireless pressure sensors installed across the basal till layer in a glacier in Norway showed that, while a sensor at the ice-till interface shows clear diurnal variations, another one placed a short distance away inside the till layer can show a signal very similar to our disconnected boreholes. This could be a problem affecting some of our sensors, as borehole drilling could eventually penetrate the till. Nevertheless, the lifespan of a sensor buried in the till ought to be short if there is differential motion between ice and the sensor placement in the till (e.g. Engelhardt and Kamb, 1998), causing the signal cable to tear. Indeed, one sensor that was accidentally installed directly on the bed with limited (1 m) cable slack, rather than suspended just above the bed, survived for only just over a month, and showed uncharacteristic high-frequency noise superimposed on a smooth diurnal oscillation (see the lowest curve in Fig. 5g).

Calibration drifts may affect in-situ sensors over time, and differences in measured water pressure may not be reflective of an actual pressure gradient between two boreholes (supplementary material section 1); consequently, we have taken similarity in response to diurnal forcing as our indicator of connections, rather than looking directly at the evolution of pressure gradients.

## 5 Modelling

Our data show that the glacier bed not only contains regions that remain disconnected from the subglacial drainage system during the melt season, but that those regions can evolve in time, and that disconnection from or reconnection to the

drainage system can be quite abrupt. By itself, that insight is not new. Previous observational studies (Murray and Clarke, 1995; Gordon et al., 1998; Andrews et al., 2014; Meierbachtol et al., 2016) have pointed out the same set of phenomena. Most models in their present form (Schoof, 2010; Hewitt, 2011; Schoof et al., 2012; Hewitt et al., 2012; Hewitt, 2013; Werder et al., 2013; Bueler and van Pelt, 2015) however do not capture them: water can flow everywhere in the domain, although the permeability of the distributed system varies with position and over time. The expected signature of the distributed system in borehole records is then a progressive decrease in amplitude of diurnal oscillations away from subglacial channels, with a corresponding phase lag (see Fig. 8 of Werder et al. 2013): the sheet behaves as a diffusive system, in which the diffusivity varies smoothly in space and time and evolves as sheet thickness does (see also Hubbard et al., 1995). This contrasts with the possibility of abrupt disconnection from the drainage system that appears to be the main feature of our field data, rather than a slow, diffusive attenuation of pressure signals.

The only exception is the model of Hoffman et al. (2016), which contains a 'weakly connected' component that exchanges water with the active remainder of the drainage system through highly inefficient connections. Diurnal pressure variations in that weakly connected system are primarily due to the effect of ice motion rather than through the exchange of water, as we have also inferred for the groups of boreholes in our data that show common, mechanically transferred pressure variations (Fig. 10). The spatial extent of individual weakly connected parts of the bed is however left unresolved in Hoffman et al. (2016), and water exchange with the distributed system occurs locally, as is also the case in dual-porosity models (de Fleurian et al., 2014). Instead of prescribing the physics by which the connection between distributed and weakly connected systems evolves, a simple linear increase in the exchange coefficient is assumed to occur during the summer.

Here we take a different approach and try to construct a model that can resolve connected and unconnected (or weakly connected) regions explicitly, and track their evolution. Our basic premise is the following: models of distributed drainage (Hewitt, 2011; Schoof et al., 2012; Werder et al., 2013; Bueler and van Pelt, 2015) typically describe a system of cavities, and model the mean cavity size at any given location. Crucially, these cavities are assumed to connect whenever they have non-zero size. Here, we replace that assumption by a percolation limit: cavities only form a connected system once they have reached a critical size. We describe the implementation of such a limit in the context of a discrete network-based model for subglacial drainage, and discuss the relatively straightforward equivalent continuum formulation in section 3 of the supplementary material.

## 5.1   Model Formulation

We assume an arbitrary network of conduits connecting nodes labelled by a single index $i$; the edge connecting nodes $i$ to $j$ is identified by the double index $ij$. The basic set-up of the model, with a handful of alterations identified below, proceeds as in Schoof (2010).

Along each network edge $ij$, we assume there are $n_c$ conduits connecting node $i$ to node $j$: One 'R'-conduit that can behave either as a Röthlisberger (R) channel or a cavity, as in Schoof (2010), with average cross-section $S_{R,ij}$, and $n_c - 1$ 'Kamb' (K) conduits that behave only as cavities, and are not subject to enlargement by melting. This configuration mimics the sheet of Werder et al. (2013) and avoids the pitfall of having to resolve every basal conduit. We denote their average cross-sectional area by $S_{K,ij}$. The conduits evolve according to

$$\frac{\mathrm{d}S_{R,ij}}{\mathrm{d}t} = c_1 Q_{R,ij} \Psi_{R,ij} + u h_R (1 - S_{R,ij}/S_{R0}) - c_2 S_{R,ij} |P_{e,ij}|^{n-1} P_{e,ij} \tag{1}$$

$$\frac{\mathrm{d}S_{K,ij}}{\mathrm{d}t} = u h_K (1 - S_{K,ij}/S_{K0}) - c_2 S_{K,ij} |P_{e,ij}|^{n-1} P_{e,ij} \tag{2}$$

Here $Q_{R,ij}$ is discharge from node $i$ to $j$ in the R-conduit, and $\Psi_{R,ij}$ the hydraulic gradient along the R-conduit, $u$ is sliding velocity, $h_R$ the size of bed obstacles supporting cavity formation, and $S_{R0}$ is the cavity-size cut-off at which further conduit enlargement drowns out bed obstacles. $P_{e,ij}$ is the effective pressure driving conduit closure (related to $N_i$ as described by equation 7 below), and $c_1$, $c_2$ are the same constants as in Schoof et al. (2014). Subscripts $K$ refer to equivalent quantities for the K-conduits.

We associate a nominal effective pressure $N_i$ with each node, defined as overburden minus basal water pressure. Hydraulic potential $\Phi_i$ at each node and hydraulic gradient $\Psi$ along the network edges are given by

$$\Phi_i = \Phi_{0,i} - N_i, \quad \Psi_{R,ij} = \frac{\Phi_i - \Phi_j}{L_{ij}}, \quad \Psi_{K,ij} = \frac{\Phi_i - \Phi_j}{T L_{ij}} \tag{3}$$

where $\Phi_{0,i} = \rho_i g s_i + (\rho_w - \rho_i) g b_i$ is the geometrical contribution to hydraulic potential, $\rho_i$ and $\rho_w$ being the densities of ice and water, $g$ acceleration due to gravity, $s_i$ and $b_i$ ice surface and bed elevation at the node. $L_{ij}$ is the length of the network edge and $T \geq 1$ is the tortuosity of the K-conduits, relative to the R-conduits. In a departure from previous models, we assume a percolation cut-off for flow along the conduits and write

$$Q_{R,ij} = c_3 \max(S_{R,ij} - S_{PR}, 0)^\alpha |\Psi_{R,ij}|^{\beta-2} \Psi_{R,ij} \tag{4}$$

$$Q_{K,ij} = c_3 \max(S_{K,ij} - S_{PK}, 0)^\alpha |\Psi_{K,ij}|^{\beta-2} \Psi_{K,ij} \tag{5}$$

Here $c_3$ is the same constant as in Schoof et al. (2012), and $\alpha$ and $\beta$ are constant exponents as in Werder et al. (2013), while $S_{PR}$ and $S_{PK}$ are the constant thresholds that conduit sizes must reach before water can flow in the conduits. For linked cavities, such a threshold is easy to justify: while $S_K$ may be the average cross-sectional area of cavities along the network edge, the local cavity size will naturally vary as bed obstacles are uneven, and it is natural to expect that cavities with non-zero size may fail to connect. We apply the threshold equally to the R-conduit as it can act as either a channel or a cavity, and a cut-off must apply self-consistently in its cavity state. A node that is connected to others purely by conduits that are all below the percolation threshold is then hydraulically disconnected from the drainage system.

At each node, water can be stored in englacial void space connected to the node, with volume storage capacity per unit water pressure $V_p$ (see also Werder et al., 2013; Schoof et al., 2014; Brinkerhoff et al., 2016). Water can also be supplied externally to each node at a locally-defined rate $m_i$, and flows along network edges through R- or K-conduits, or possibly through a permeable porous substrate if the conduits are closed. To account for conservation of mass, we also associate half the volume of water stored in a conduit between two nodes with each node, and likewise, account for half the water created by wall melting in an R-conduit as water supply to each node. Consequently, we impose mass conservation in the form

$$V_p \frac{dN_i}{dt} + \sum_j \left[ Q_{R,ij} + (n_c - 1)Q_{K,ij} + k_{\text{leak}}\Psi_{ij} \right]$$
$$= \sum_j \frac{\rho_i}{2\rho_w} c_1 Q_{R,ij} \Psi_{R,ij} L_{ij} + m_i \qquad (6)$$

where $k_{\text{leak}}$ represents the possibility of a substrate (till) with non-zero permeability. Sums over $j$ are taken over nodes connected to node $i$.

To close the model, we need to relate the conduit effective pressure $P_{e,ij}$ to the nominal effective pressures $N_i$ at network nodes. We write this in the form

$$P_{e,ij} = \sum_k G_{ijk} N_k \qquad (7)$$

where the sum is over all node indices $k$ in the network, and $G_{ijk}$ is a suitable positive averaging kernel that satisfies $\sum_k G_{ijk} = 1$; in our network model below we put $G_{ijk} = 1/2$ if $k = i$ or $k = j$ and $G_{ijk} = 0$ otherwise for simplicity; this is however a surprisingly key assumption (see also section 3 of the supplementary material). Suppose we have $k_{\text{leak}} = 0$ and no hydraulic connection at all between adjacent nodes. This can lead to arbitrarily large effective pressure gradients. The usual assumption of cavity formation models (Fowler, 1987; Schoof, 2005; Gagliardini et al., 2007) breaks down, namely that adjacent cavities are subject to the same nominal effective pressure, defined as overburden pressure (or far field normal stress) minus a common

water pressure. The rate of opening or closing of a cavity is unlikely to be a function of its own nominal effective pressure alone, and is likely to be affected by stresses around other nearby cavities (and hence dependent on their nominal effective pressure: the observation of anti-correlated water pressure records in our dataset indicating a load transfer of overburden onto highly pressurised parts of the bed (Murray and Clarke, 1995) also supports this assumption. We try to capture this load transfer effect by the averaging kernel $G$ above.

More practically, if conduit opening and closing were driven by a local effective pressure variable alone, then the generalisation of our model to a distributed sheet (Werder et al., 2013) would result in disconnected parts of the bed potentially never reconnecting. In order for a disconnected region to reconnect, sheet thickness in the disconnected region needs to change. On the absence of leakage through the substrate, the only way that can happen in a way that is driven by the hydrology of the connected regions is through a non-local sheet closure term, or through the sliding velocity $u$. We expand on this in section 3 of the supplementary material, but note here that conduit closure must involve a non-locally defined effective pressure in order for our percolation model to function as intended, allowing for expansion as well as the contraction of a connected region at the bed.

A key component that the model above continues to miss is the ability to open conduits due to overpressurisation of the system (Schoof et al., 2012; Hewitt et al., 2012; Bueler and van Pelt, 2015; Dow et al., 2015). While not necessary to explain switching events during the main melt season, this is likely to be key in establishing a drainage system at the start of the melt season: unlike existing sheet models, in which a distributed system always exists and can simply be expanded through water supply in the spring, the percolation limit model above allows the system to shut down completely, and rapid re-establishment through a spring event is likely to require overpressurisation. We discuss the extension of the approach in Schoof et al. (2012) and Hewitt et al. (2012) further in section 3 of the supplementary material.

## 5.2 Model results

Network-based models for drainage channelisation (e.g. Schoof (2010); Hewitt (2013); Werder et al. (2013)) have been used previously to model the seasonal evolution of drainage systems. In particular, they have been used to model the evolution of a channelised system from a more spatially extensive one as in stages 1 and 2 identified in this paper, and the subsequent shut-down of the system as in stage 3. See figure 3 of Schoof (2010), figure 5 of Hewitt (2013), and figure 12 of Werder et al. (2013) for examples of seasonal drainage evolution. What these models are missing is the ability to capture the formation of disconnected regions at the glacier bed and the subsequent ability of parts of the bed to connect

**Table 1.** Parameters used in the simulations

| Symbol | Value |
|---|---|
| $c_1$ | $1.35 \times 10^{-9}$ m$^3$ J$^{-1}$ |
| $c_2$ | $3.44 \times 10^{-24}$ Pa$^{-3}$ s$^{-1}$ |
| $c_3$ | $4.05 \times 10^{-2}$ m$^{9/4}$ Pa$^{-1/2}$ s$^{-1}$ |
| $k_{\text{leak}}$ | 0 |
| $n$ | 3 |
| $\alpha$ | 5/4 |
| $\beta$ | 3/2 |
| $\rho_i$ | 910 kg m$^{-3}$ |
| $\rho_w$ | 1000 kg m$^{-3}$ |
| $g$ | 9.8 m s$^{-2}$ |
| $S_{PR}, S_{PK}$ | $2.65 \times 10^{-3}$ m$^2$ |
| $S_{R0}, S_{K0}$ | 3.32 m$^2$ |
| $uh_R, uh_K$ | $3.47 \times 10^{-6}$ m$^2$ s$^{-1}$ |
| $T$ | 1 |
| $n_c$ | 2 |
| $V_p$ | $2.53 \times 10^{-5}$ m$^3$ Pa$^{-1}$ |

and disconnect to the drainage system, which is what we focus on here.

Instead of attempting to model a full seasonal cycle, we focus here on the effect of time-dependent water input into a fully channelised drainage system, in order to test whether our modification of existing models can capture the qualitative behaviour of the switching events such as those in Fig. 11 (see in particular panel g). In other words, we focus on the behaviour of the drainage system in stage 2. Our simulation is an idealised run, not based on the specific geometry or properties of the South Glacier field site and does not claim to reproduce observations beyond their generic features. Work to use proxies for surface melt rates and likely surface water supply routing in an inverse model for the drainage system is currently underway and will be reported elsewhere. Our aim here is simply to study the qualitative features of our forward model, modified from existing ones found in the literature, and to compare them equally qualitatively with our borehole records.

The rectangular domain is 5 km long and 1 km wide. The ice and bed surfaces are used with contour lines in panels c and g of Fig. 16, where black lines are surface contours at 100 m intervals, and grey lines are bed contours at the same intervals. Zero inflow is prescribed at the sides and top of the domain, and zero effective pressure at the lower end of the domain $y = 0$. The network geometry is the same as indicated in Fig. 1 of the supplementary material to Hewitt (2013), with a total of 201×201 nodes.

We allow water to be supplied in 40 discrete locations (effectively, moulins). Each moulin undergoes a diurnal cycle whose amplitude varies over several days, with mean water supply rates also varying over several days; the dominant period of the cycle is the same for each moulin but the ratio of diurnal amplitude to mean water supply is chosen randomly (while maintaining positive water supply rates at all times),

and we have allowed for slight phase shifts between moulins. The time series of water supply to all moulins are shown in Fig. 16i.

We show two different simulations. Both use the parameter values shown in table 1, except that the percolation cut-offs $S_{PR}$ and $S_{PK}$ have been set to zero in one of the simulations (to identify the effect a percolation limit has on our results), and $uh_R$ and $uh_K$ are also set to one-tenth the value given in table 1 in the same simulation (without a percolation limit, cavities of the same size will permit larger discharge, so we reduce the cavity opening rates $uh_R$ and $uh_K$ in order to limit cavity size).

Figure 16 shows one set of panels for each simulation, identified by the numbers 1 (no percolation cut-off) and 2 (finite percolation cut-off) in the panel labels. When referring to a specific panel for both simulations at the same time, we will identify it by the letter in the panel label. Both simulations start from a fully channelised steady state computed with moulin water supply set to constant values. Diurnal oscillations are subsequently superimposed on those constant water supply values. The system is run for several days to account for transients before the detailed results shown in Fig. 16 are computed.

The channelised configuration of the system does not change during the simulation (compare panels a and e in Fig. 16), although it differs slightly between the two simulations. Pressure oscillations result from the time-dependent water supply. These pressure variations are confined to the connected drainage system (compare panels c and g of 16, and see also the supplementary movie #2).

In simulation 2 (with non-zero percolation cut-offs), the extent of connected drainage system also evolves, shown as blue areas in panels d2 and h2 of Fig. 16 and the supplementary movie #2. While pockets of water can move downglacier essentially without connection to the channelised system (see the left-hand side of the main drainage axis in supplementary movie #2), the main feature here is the expansion of the connected system at times of large water supply and low effective pressures. The larger connected system in panel h2 of Fig. 16 corresponds to peak water supply, the smaller system in panel d2 to the minimum water supply in the cycle shown. This is at least qualitatively consistent with our observation of switching events, that establish connectivity during periods of increasing water supply.

In Fig. 17, we focus on what an array of boreholes would observe. The borehole array is located in the black rectangle shown in panel d2 of Fig. 16. Panels a2-c2 in Fig. 17 show the evolution of the connected parts of the bed within the borehole array, while panels d2-g2 show pressure time series, again grouped subjectively.

The presence of at least two distinct drainage subsystems is immediately obvious (circles and diamonds in panels a2-c2, corresponding to the time series shown in panels d2 and e2 respectively). These two subsystems correspond to two different drainage channels. The grouping of boreholes in

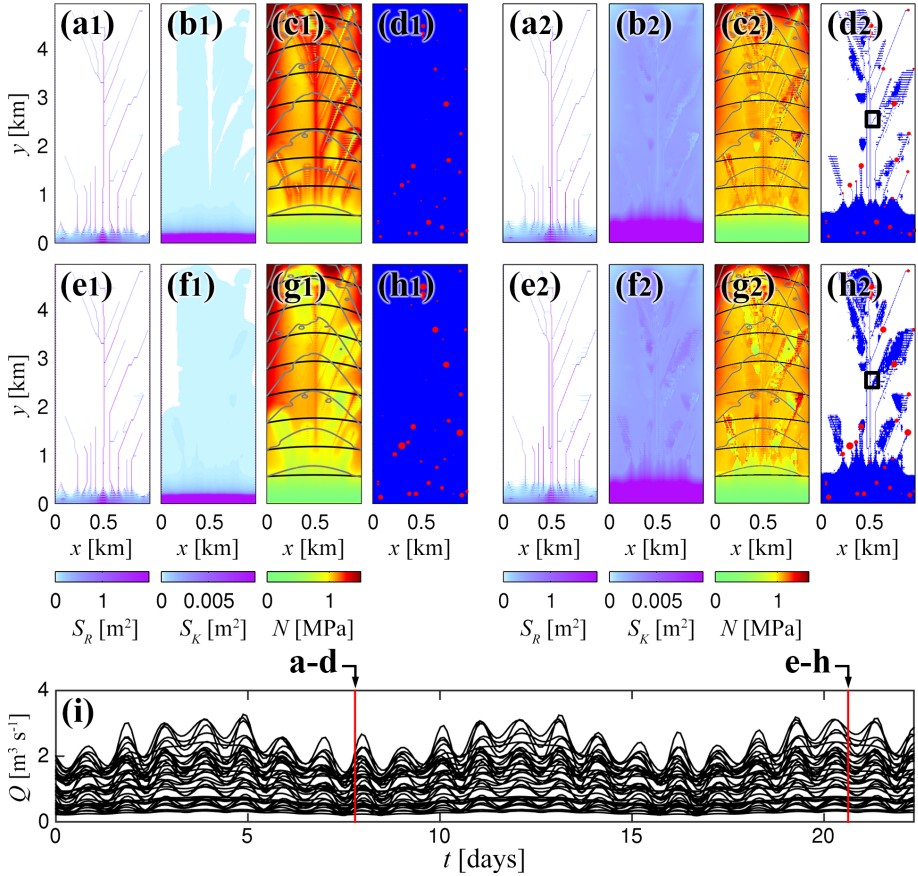

**Figure 16.** Snapshots of drainage system evolution for the model without a percolation cut-off (left), and with it (right). Panels a & e: $S_R$ conduit size. Panels b & f: $S_K$ conduit size. Panels c & g: effective pressure, black lines are 100 m surface contours, grey lines 100 m bed contours. Panels d & h: connectedness of conduits, indicated in blue if $S_{R,ij} > S_{PR}$ or $S_{K,ij} > S_{PK}$ along a given edge, in white otherwise. Red dotes indicate moulin locations, size of dot scaled with instantaneous water supply. Row a-d show solutions at $t = 7.8$ days and row e-h at $t = 20.6$ days. Panel i: water supply time series for all moulins in the domain.

panel f2 (triangles) is intermittently connected to the diagonal channel of panel e2 (diamonds), with different boreholes connecting and disconnecting at different times, through connection and disconnection are again typically favoured by low and high effective pressures, respectively. This is in qualitative agreement with our actual borehole data (see Fig. 11g). In addition, there is an additional grouping of persistently disconnected boreholes (Fig. 17 panel g2, crosses), although two of these become very poorly connected later in the cycle, permitting an excursion in effective pressure without obvious diurnal cycling.

One important aspect of the synthetic borehole records in panel f2 of Fig. 17, is the relatively minimal attenuation of amplitude and minimal phase lags observed within that distributed system relative to the channel (panel e2) to which the distributed system connects, and the abrupt switching to nearly constant effective pressures on disconnection. Compared with borehole data from South Glacier, we do not reproduce the tendency of disconnected boreholes to experience rising water pressure (i.e. falling effective pressure),

which we believe is related to the dynamics of disconnected boreholes incised upwards into the ice being squeezed by anisotropic stresses in the ice, an effect this drainage model is not designed to capture.

Removing the percolation cut-off for simulation 1 increases the ability of the distributed system to drain water (equations 4 and 5). To account for this we simultaneously lower conduit opening rates $uh_R$ and $uh_K$ to $3.47 \times 10^{-7}$ m$^2$ s$^{-1}$, keeping all other parameters the same. In order to create comparable drainage structures in both simulations, simulation 1 was started from the same initial state as simulation 2. The percolation cut-offs and conduit opening rates were then gradually changed with water supply rates held constant until a new steady state was achieved, before imposing the same diurnal oscillations as in simulation 2 again (panel i of Fig. 16).

A small change in channel configuration results from removing the percolation cutoffs: two of the main drainage channels along the centre of the glacier in Fig. 16 panels a2 and e2 collapse onto a single channel in panels a1 and e1, as

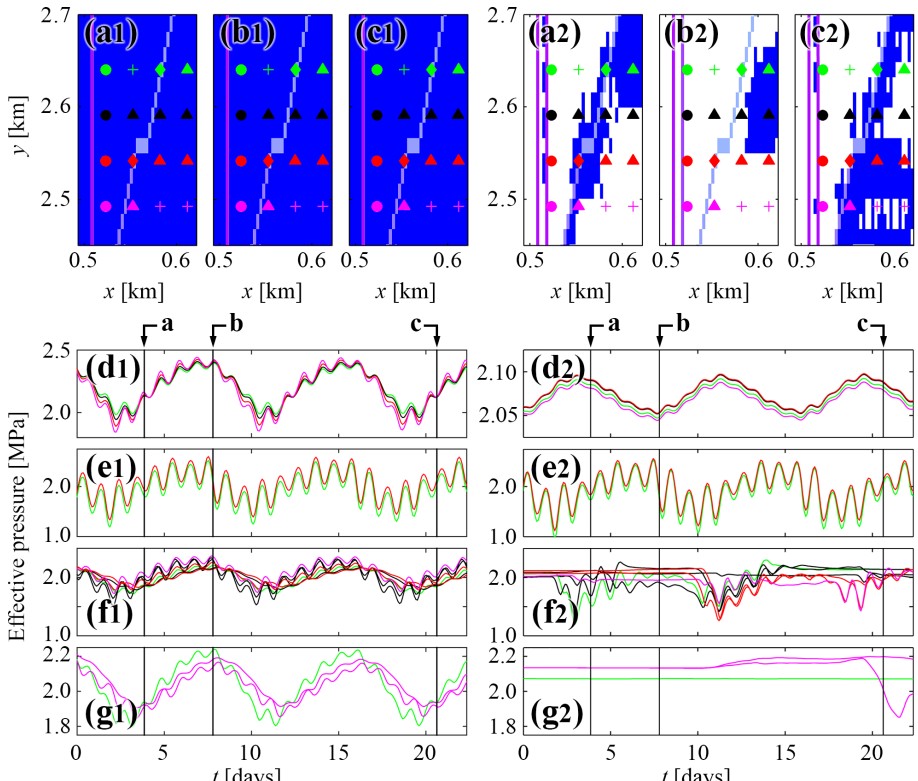

**Figure 17.** A synthetic borehole grid in the model with (right-hand column) and without a percolation cut-off (left-hand column), and with it (right). (a–c): enlargements of the inset box in Fig. 16(d2,h2) at times $t = 3.9$ (a), 7.8 (b) and 20.6 (c) days. Superimposed on the blue connectivity map is $S_R$ conduit size, plotted using the same colour scheme as indicated by the colour bar in panels a & e in Fig. 16 . Also shown are the locations of 16 synthetic boreholes, colour-coded by row, (d–g). Effective pressure time series from the boreholes, grouped according to borehole symbols: circles (d), diamonds (e), triangles (f), crosses (g). Each time series is colour-coded by row.

they are no longer isolated from each other by the percolation cut-off. The main difference in model results is however the much larger region over which the effect of oscillatory water input is felt away from the channels (compare Fig. 16 panels c1 and g1 with panels c2 and g2). This is a natural consequence of enforcing connectivity in the drainage system everywhere (see Fig. 16 panels d1 and h1).

On the left-hand side of Fig. 17 we use the same groupings of synthetic boreholes as for the simulation with a percolation cut-off (right-hand side of the same figure). The boreholes marked as diamonds produce an almost identical pressure time series as in the first simulation (compare panels e1 and e2 in Fig. 17), indicating that the behaviour of channelised drainage is not substantially affected by dispensing with the percolation cut-off. By contrast, the boreholes marked as circles in Fig. 17 experience higher mean effective pressures and bigger oscillations in panel d1 than d2. This is the result of the two channels on the left-hand edge of the domain in Fig. 17 panels a2-c2 having been merged into a single channel in panels a1-c1: the percolation cut-off allows subsystems to co-exist separately in closer proximity, in accordance with our observations at South Glacier.

The biggest difference is in the behaviour of the boreholes within the distributed system surrounding the channels (Fig. 17, panels f and g). Unlike in the case of the model with a percolation cut-off, the results in panels f1 and g1 no longer exhibit switching events, and there are no persistently disconnected boreholes. Instead, we see evidence of typically diffusive behaviour away from the channels: a reduction in the amplitude especially of the higher frequency (diurnal) forcing components with an attendant phase shift, and the absence of a sharp division between drainage subsystems. This behaviour mimics that in Fig. 8 of Werder et al. (2013), but contrasts with our field observations. Those observations indicate minimal variations in amplitude and phase shifts within drainage subsystems, with sharp boundaries separating different subsystems. The inability to explain those features of our field observations motivates the model modification we have proposed here.

That modification comes with one major drawback, which we do not attempt to resolve here. While the model is able to open drainage connections spontaneously, this is a slow process driven by viscous deformation, controlled by the non-local effective pressure term in equation 7. When the drainage system is subject to a rapid increase in water supply,

the physics by which drainage connections are established may involve either elastic hydrofracture driven by overpressurisation (e.g. Tsai and Rice (2012)) or the large-scale uplift of ice at flotation as described in Schoof et al. (2012). As we discuss in section 3 of the supplementary material, the latter is not straightforward to incorporate into our modified model, as is the former (which requires a blending of elastic and viscous effects). We identify this as an important area for future research.

# 6 Conclusions

While winter pressure record suggests that most boreholes remain disconnected during that period, a rapid springtime increase in melt overwhelms the water storage capacity of the snowpack, leading to the sudden supply of water to the bed and activation of an extensive and well-connected distributed drainage system. During this period, the majority of boreholes show similar diurnal pressure variations and experience modest water transport (see section 3.1).

Over time, water transport becomes concentrated in some areas, and probably becomes channelised: water flow ends up focused in R-channels surrounded by a distributed drainage system that carries relatively low water fluxes. Borehole water pressure data in most cases do not allow the direct identification of channels. In fact, in most cases, our borehole array probably fails to intersect the narrow R-channels. However, in one instance we were able to confirm the existence of a channel from direct observation in a borehole in which the lowermost 50 cm were occupied by turbulent water flow.

The increase in effective pressure associated with channelisation leads to the progressive shut-down of drainage activity in the surrounding distributed drainage system, possibly due to basal cavities becoming isolated from each other as they shrink under the effect of a larger effective pressure. During long and hot enough summers, most of the bed can become disconnected, concentrating drainage in narrow pathways.

The eventual complete shut-down of the entire drainage system at the end of the summer season is presumably the result of low water supply: high effective pressure and low dissipation rate in channels allow basal conduits to close. This appears to be strongly linked with the appearance of fresh snow cover, rather than the arrival of low temperatures alone (see section 4).

Most of our observations are consistent with borehole data from other sites. However, the density of boreholes at South Glacier has allowed us to identify, in particular, the prevalence of "switching events", through which the drainage system focuses, and the disconnected areas enlarge. Such disconnected areas always exist, even during the spring event. Disconnected parts of the bed are necessary to account for many aspects of our data, including anti-correlation between borehole pressure time series, above-overburden water pressures, and the occurrence of strongly correlated high-frequency pressure variations in sets of widely spaced boreholes (see section 4.3). As in Hoffman et al. (2016), our data suggest that disconnected areas need not be completely isolated, but can experience slow leakage into the active drainage system (see section 4.2).

In view of the above, perhaps the main shortcoming of most current drainage models is their inability to account for the evolution of an disconnected or weakly connected component (Hoffman et al., 2016). This ability can however be incorporated in the current modelling framework as a percolation threshold, assuming that cavities only form a connected system once they reach a critical size. We have implemented this approach in a simple model, allowing us to reproduce qualitatively some of the main features of our dataset: sharply-defined drainage subsystems with insignificant diffusive pressure signal attenuation and the existence of disconnected areas (See section 5.1).

However, the ability of the system to fully shut-down requires the incorporation of other physical process that could allow the reactivation of the drainage system during the spring event, something that is probably accomplished by overpressurisation. The model also requires a more careful treatment of normal stress redistribution, in particular in association with isolated and closely spaced cavities of very different water pressures. This is left for future work. In the future, we also hope that it will be possible to test models like the one presented here or more sophisticated versions of it, against detailed borehole datasets such as that from South Glacier.

*Data availability.* The presented dataset will be made publicly available in the future. Ongoing work is taking place to meet the format and create the ancillary data and documentation required for the release, that is expected to happen fully or partially by the end of 2018. In the meantime, it is available on request from the second author at cschoof@eoas.ubc.ca. The model code in Matlab and configuration parameters are included in the supplementary material.

*Competing interests.* The authors declare that they have no competing interests

*Acknowledgements.* We thank Manar Al Asad, Faron Anslow, Ashley Bellas, Kyla Burrill, Emilie Delaroche, Jennifer Fohring, Tom-Pierre Frappé-Sénéclauze, Johan Gilchrist, Marianne Haseloff, Ian Hewitt, Marc Jaffrey, Alex Jarosch, Conrad Koziol, Natalia Martinez, Arran Whiteford and Kevin Yeo for assistance in the field. Gwenn Flowers provided bed elevation, South Glacier AWS data as well as continuous help and advice without which this project would not have succeeded. Additional AWS data were made available by Christian Zdanowicz and Luke Copland. We are indebted to Parks Canada and Kluane First Nation for their support and permission to operate at the field site, to Doug Makkonen, Dion Parker and Ian

Pitchforth for expert flying, to Andy Williams, Sian Williams and Lance Goodwin for logistics support. This work was supported by the Natural Science and Engineering Research Council of Canada through Discovery Grants 357193-08 and 357193-13, Accelerator Supplement 446042-13, Northern Research Supplements 361960-06 and 361960-13 as well as Research Tools and Instruments Grant 376058-09; by the Polar Continental Shelf Project through grants 625-11, 638-12, 637-13, 663-14 and 667-15; and by the Canada Foundation for Innovation and British Columbia Knowledge Development Fund through Leaders Opportunity Fund project number 203786 and 227698.

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
