# Peer review of "Channelised, distributed, and disconnected: subglacial drainage under a valley glacier in the Yukon"

_The Cryosphere, 2017_

## Referee Comment (RC1) · Anonymous Referee #1 · 26 Feb 2018

The manuscript "Subglacial drainage characterization from eight years of continuous borehole data on a small glacier in the Yukon Territory, Canada" by Rada and Schoof, describes and interprets an eight year dataset of observations from dozens of boreholes on a small mountain glacier in the Yukon. The dataset is extremely rich and shows complexity at diurnal and seasonal temporal scales at both local and regional spatial scales. The boreholes show support for the standard model of subglacial hydrology of an inefficient distributed system punctuated by an efficient channelizing system of limited spatial extent. However, the data also provide strong evidence for disconnected regions of the bed that change in time. The authors present a novel extension to an existing subglacial drainage model that adds a physical representation of the evolution of these disconnected areas. The model is capable of qualitatively reproducing both the traditional distributed and channelized regions as well as the complexity observed in the disconnected regions of the bed.

This study adds South Glacier as a new benchmark location in the pantheon of subglacial hydrology borehole studies by providing an exhaustive description of the long record there beyond the description in Schoof et al. (2014). The study makes a convincing case for many of the "anomalous" features of borehole studies (rapid switches in behavior, out of phase boreholes, water pressure above overburden, etc.) by demonstrating features that are widespread both spatially and temporally and cannot be written off as a fluke of limited sampling. The modeling component of this paper is equally significant. It builds on a recent renewed interest in the disconnected or weakly connected parts of the bed by presenting the first model that allows these regions to evolve on their own. The model results are striking in their ability to reproduce diverse behavior over short length scales that qualitatively matches the complex borehole record. This significant achievement may pave the way for advanced subglacial drainage models that are finally able to reproduce the diversity of behavior seen in the field, as well as simulate the annual cycle.

While the paper has many strong points, that also leads to its limitations - it is a long, dense paper and some of the key points get lost a bit among the thorough observational descriptions. In particular, the important modeling section reads like a bit of an afterthought, which is unfortunate. At times it reads like a chapter from a dissertation, and it might more naturally form two shorter papers. The interpretation of the model results and comparison to existing literature feels a bit incomplete, as explained below.

General Comments ————

1. My primary concern with the paper is the somewhat abbreviated model interpretation and incomplete comparison with previous work. What is there is very interesting, but the rich model record could be compared a bit more thoroughly to the borehole record (here and elsewhere). There is also no consideration for which model parameters this enhanced model might be sensitive to (though I think a detailed sensitivity study is well beyond the scope of this paper). I did appreciate the model technical detail discussion in the SI, and it would be nice to see those topics mentioned in the main text to encourage readers to look into the SI. My biggest concern in this area is what feels like an incomplete comparison to previous work. The authors acknowledge previous investigations of the disconnected (or weakly connected) system, and their model is an elegant extension of previous, simpler attempts to model it (Hoffman et al., 2016). However, the interpretation in this paper is that the subglacial drainage system becomes increasingly fragmented and disconnected as the summer progresses, while previous studies suggest summer brings *increased* connectivity (Gordon et al., 1998; Hoffman et al., 2016; Iken and Truffer, 1997; Murray and Clarke, 1995). This difference in interpretation should be discussed, and, if possible, reconciled.

2. It would be nice to see a bit more acknowledgement of the possibility of borehole behavior being governed by the presence of subglacial till. This is discussed briefly in a few places, but the paper would benefit from additional consideration of it, or a stronger justification for a dominantly hard bed interpretation. For example, on p. 27 there is discussion about localized diffusive systems with limited flow, and it seems like till would fit the bill.

3. p18, lines 2-3: How much does snow cover or ice albedo change? It is a questionable assumption that the degree day factor remain does not change significantly as surface conditions change over the summer. The interpretation in Figure 11c seems rather tenuous.

4. The title is fine as it is, but it is worth considering the title somehow including something about the importance of the disconnected/weakly connected system in the interpretation, as this is a primary result.

5. As mentioned above, the borehole results section is quite long.

Specific Comments ————

Abstract seems a little short given the length of the paper, but it does hit the most significant highlights of the paper.

1, 22: basal "slip" may be considered the preferred term here (Cuffey and Paterson, 2010), to acknowledge the ice is not sliding differentially from the substrate at its sole.

2, 6: No comma here.

2, 6-11: Interactions between subglacial hydrology and ice motion could be mentioned here as well (e.g., Hoffman and Price, 2014). And Gordon.

2, 29: missing "a" -> "to provide a less efficient"

2, 33: "do" should be removed.

2, 32-34: (Creyts and Schoof, 2009) could be an appropriate additional reference for this topic.

3, 5: I believe you meant for the second "channels" on this line to be "conduits".

3, 9-15: This is a nice summary of the complexity in borehole observations. A few suggestions for additional references: "widespread areas of high water pressure during winter": (Ryser et al., 2014; Wright et al., 2016) "large pressure gradients": (Fudge et al., 2008) "sudden reorganizations": (Gordon et al., 1998) "anti-correlated temporal pressure variations": (Andrews et al., 2014; Ryser et al., 2014)

3, 31: "in-deep" -> "in-depth"

5, 33: Could mention that many authors refer to this as "hydraulic head".

Figure 2: I recognize that showing so many different symbols and colors is challenging, but it is difficult to differentiate some of them. Perhaps removing the 3d shading on the symbols would help. In particular, the red symbols are hard to make out. Maybe put a

circle around them or something to make them easier to see. Also, the black and blue lines are difficult to tell apart. Finally, the concept of "upstream area" from Schoof et al., (2014) should be briefly elaborated on (either in the caption or the text).

Figure 3: The figure is a bit small in my printout. In particular, the green dots are hard to see.

7, 8: Consider changing "Fast-Flow" to "Fast Water Flow". When I first read this I interpreted this to be a region where ice velocity is fast.

8, 31: bummer

Figure 4: The colors are a bit difficult to match to the map. Again, perhaps removing the 3d shading of the symbols on the map would help.

Figure 4 caption: in part c), it says two sensors were installed here, but I only see one line in the plot. Clarify if they are plotted on top of each other or if only one is plotted and, if so, which and why.

Figure 5: Is correlation to temperature calculated for panel c here as it was for Figure 4? If not, mention that in the caption.

11, 18: Should "ice" be "water" here?

13, lines 1, 2, 3, 5, 7, 9: Include figure number with each panel reference.

Figure 8: Caption for g) refers to six digital sensors included in panel b, but panel b is the temperature record.

15, 20: The second comma should be removed.

Figure 10: Panel c is pretty hard to make out details of. Perhaps this figure could be reorganized into two columns, or panel c could somehow be made a bit larger.

17, 3: Fig. 8b must be an incorrect reference - do you mean Fig. 10c?

Figure 13: Consider putting earlier on, perhaps with Figure 3.

Section 3.6: The data quality section would be more natural in section 2 (methods), than late in the results section.

22, 10: I think you mean "120% of overburden", not "above".

22, 11: Consider replacing "and" with "however" or "yet" to make it clear you are arguing against sensor drift being able to explain these observations.

23, 12-14: This text would flow better with this sentence in parentheses.

25, 5: (Hubbard et al., 1995) could be an additional appropriate reference here.

25, 14: An aside: water pressure in nearby moulins/crevasses would be useful here. Something to consider if this field campaign continues.

25, 21: Some discussion of bridging stresses leading to isolation of low pressure channels would be good here (Hewitt, 2011; Lappegard et al., 2006).

25, 23: Also, Figure 3.

25, 29: Mention that high up-stream areas means a likely water flow accumulation path (see comment above about introducing the significance of this upstream area).

26, 1-7: This discussion would benefit from inclusion of (Meierbachtol et al., 2016).

26, 28: "fragment into subsystem" -> "fragmented into subsystems"

27, 3-7: Interesting discussion. I think the quotes around "phase lag" should be removed.

27, 21: Is the distance long enough relative to channel flow speed for a phase lag to be expected? The other complication is there could be additional inputs of water from the surface that help to "lock" the channel phase to the surface phase even in the presence of diffusion within the subglacial system.

27, 27-29: Consider (Meierbachtol et al., 2016) again here.

27, 32: There is an alternative hypothesis as well of passive cavity opening due uniform basal sliding (Bartholomaus et al., 2011; Hoffman and Price, 2014; Iken and Truffer, 1997).

28, 8: These island sound like the system described by (Murray and Clarke, 1995).

28, 19: Wouldn't disconnected areas act as *slippery* spots since they maintain high water pressure?

28, 22-26: This is a significant result and well-stated here.

30, 3: "differential motion between ice and till": If basal slip is primarily due to till deformation, then there will not be differential motion between ice and till.

31, 11: Please define $n_c$.

31, 11: Is there a significance to the designation 'K' or is it just an arbitrary letter choice?

31, 11: Consider adding "along that edge" after "$n_c$-1 'K' conduits" to emphasize that this treatment is per edge.

Eq 1: It seems odd to use lettered sub-equations rather than a new number for each equation.

31, 21: Also define Psi here.

Eq. 1d/e: A minor quibble: It would seem more intuitive if the threshold size also contributed to flow once the threshold is reached (which is not the case in 1d/1e). However I doubt the choice of how to treat that affects the results in a qualitative way, so either approach is defensible.

Eqn. 1f/1g: Mention this is describing mass conservation to aid the reader. Also, this is a single equation so there should be a single label.

32, 17: This is a run-on sentence. How about ending it at "nodes" and starting a new

sentence with "We".

33, 7: Is (Dow et al., 2015) meant here?

5.2 It would be clearer to call this section "Model Results".

36, 1: "eventually" should be "eventual".

37, 11: The word "a" should be removed.

Data availability: What about model and model configuration and output? Mention it is included in the SI.

Supplemental Material ——————————

* paper_movie.mpg does not play for me.

* It would be more natural to switch the order of sections 1 and 2 to match the order these topics were presented in the main text.

* The SI material, particularly the modeling part, has some very useful information. I would like to see the main text refer to the SI in more places, with brief descriptions of what is found there.

References —————

Andrews, L. C., Catania, G. A., Hoffman, M. J., Gulley, J. D., Lüthi, M. P., Ryser, C., Hawley, R. L. and Neumann, T. A.: Direct observations of evolving subglacial drainage beneath the Greenland Ice Sheet, Nature, 514(7520), 80-83, doi:10.1038/nature13796, 2014.

Bartholomaus, T. C., Anderson, R. S. and Anderson, S. P.: Growth and collapse of the distributed subglacial hydrologic system of Kennicott Glacier, Alaska, USA, and its effects on basal motion, J. Glaciol., 57(206), 985-1002, doi:10.3189/002214311798843269, 2011.

Creyts, T. T. and Schoof, C. G.: Drainage through subglacial water sheets, J. Geophys.

Res., 114(F4), F04008, doi:10.1029/2008JF001215, 2009.

Cuffey, K. and Paterson: The Physics of Glaciers, 4th ed., Butterworth-Heinneman, Amsterdam., 2010.

Dow, C. F., Kulessa, B., Rutt, I. C., Tsai, V. C., Pimentel, S., Doyle, S. H., As, D., Lindbäck, K., Pettersson, R., Jones, G. A. and Hubbard, A.: Modeling of subglacial hydrological development following rapid supraglacial lake drainage, J. Geophys. Res. Earth Surf., 120, 1127-1147, doi:10.1002/2014JF003333, 2015.

Fudge, T. J., Humphrey, N. F., Harper, J. T. and Pfeffer, W. T.: Diurnal fluctuations in borehole water levels: configuration of the drainage system beneath Bench Glacier, Alaska, USA, J. Glaciol., 54(185), 297-306, doi:10.3189/002214308784886072, 2008.

Gordon, S., Sharp, M., Hubbard, Y. Ã. B., Smart, C., Ketterling, B. and Willis, I.: Seasonal reorganization of subglacial drainage inferred from measurements in boreholes, Hydrol. Process., 12, 105-133, 1998.

Hewitt, I. J.: Modelling distributed and channelized subglacial drainage: the spacing of channels, J. Glaciol., 57(202), 302-314, doi:10.3189/002214311796405951, 2011.

Hoffman, M. J., Andrews, L. C., Price, S. A., Catania, G. A., Neumann, T. A., Luethi, M. P., Gulley, J., Ryser, C., Hawley, R. L. and Morriss, B. F.: Greenland subglacial drainage evolution regulated by weakly-connected regions of the bed, Nat. Commun., 7, 13903, doi:10.1038/ncomms13903, 2016. Hoffman, M. and Price, S.: Feedbacks between coupled subglacial hydrology and glacier dynamics, J. Geophys. Res. Earth Surf., 119, 1-23, doi:10.1002/2013JF002943, 2014.

Hubbard, B., Sharp, M., Willis, I., Nielsen, M. and Smart, C.: Borehole water-level variations and the structure of the subglacial hydrological system of Haut Glacier d'Arolla, Valais, Switzerland, J. Glaciol., 41(139), 572-583, 1995.

Iken, A. and Truffer, M.: The relationship between subglacial water pressure and velocity of Findelengletscher, Switzerland, during its advance and retreat, J. Glaciol.,

43(144), 328-338, 1997.

Lappegard, G., Kohler, J., Jackson, M. and Hagen, J. O.: Characteristics of subglacial drainage systems deduced from load-cell measurements, J. Glaciol., 52(176), 137-148, doi:10.3189/172756506781828908, 2006.

Meierbachtol, T. W., Harper, J. T., Humphrey, N. F. and Wright, P.: Mechanical forcing on water pressure in a hydrologically isolated reach beneath Western Greenland's ablation zone, Ann. Glaciol., 57(72), 62-70, doi:10.1017/aog.2016.5, 2016.

Murray, T. and Clarke, G. K. C.: Black-box modeling of the subglacial water system, J. Geophys. Res., 100(B7), 10231-10245, 1995.

Ryser, C., Lüthi, M. P. P., Andrews, L. C. C., Catania, G. A. A., Funk, M., Hawley, R., Hoffman, M. and Neumann, T. A.: Caterpillar-like ice motion in the ablation zone of the Greenland ice sheet, J. Geophys. Res. Earth Surf., 119(10), 1-14, doi:10.1002/2013JF003067, 2014.

Schoof, C., Rada, C. A., Wilson, N. J., Flowers, G. E. and Haseloff, M.: Oscillatory subglacial drainage in the absence of surface melt, Cryosph., 8(3), 959-976, doi:10.5194/tc-8-959-2014, 2014.

Wright, P. J., Harper, J. T., Humphrey, N. F. and Meierbachtol, T. W.: Measured basal water pressure variability of the western Greenland Ice Sheet: Implications for hydraulic potential, J. Geophys. Res. Earth Surf., 121, 1-14, doi:10.1002/2016JF003819.Received, 2016.

---

## Referee Comment (RC2) · B.P. Lipovsky (Referee) · 26 Mar 2018

The authors present a new subglacial hydrological dataset. The volume of data is notable. The authors do a solid job of curating complexities of the observations for the reader (Section 2, 3, and 4). Among a wealth of other observations, the authors show that out of the 311 boreholes drilled, 71% showed slow-flow behavior, 3% showed fast flow behavior, and 26% appeared to be hydraulically disconnected. This observation motivates the authors to modify the previously published model by Schoof (2010) in order to account for hydraulically isolated parts of the bed. The modified model introduces a percolation description whereby cavities form a connected system if and only if they exceed a critical dimension. I have listed my questions and comments in the attached outline.

Brad Lipovsky

1. **Additional questions about the observations/interpretation (Sections 2, 3, 4)**
    a. Given the complexity of the spatial patterning, would it be possible to make a movie that plots all the data? I envision the map in Figure 2 with each symbol having a color that is associated with a pressure scale. This should be feasible given the low sampling rate. There's only so much that can be conveyed with words.
    b. How long does drainage of the borehole take upon connection to the bed? This timescale is mentioned only qualitatively in the manuscript. Early work by Kamb and Englehardt used this timescale to estimate properties of subglacial conduits.
    c. Relative amplitude of pressure and temperature. Interquartile ranges (instead of standard deviations) may be more useful given the orders of magnitude variability.
    d. Is it possible to quantify how fast switching events or connection/disconnection occur? For example, on page 27 line 26: "very abruptly in time". What does that mean, exactly? Do transitions ever occur faster than the sampling resolution?
    e. What does the pressure sensor response curve look like with and without the snubbers? Do the snubbers limit the ability of the sensor to measure high-frequency water pressure oscillations?
2. **Questions about the model (Section 5).**
    a. A broader question regarding this type of modeling (i.e., also applicable to Schoof, 2010; Werder et al., 2013): Are conduit models

convergent under grid refinement?  Werder et al. (2013) in their Appendix A discuss grid densification.  As those authors pointed out, this creates complexities associated with changing the domain geometry. But what refinement is undertaken in such a way that more grid nodes are added only at the midpoints between existing grid nodes.  Does the model converge under this narrower sense of grid refinement?

b. What are the smallest scales that must be resolved by the spatial discretization?  Do these length scales have practical significance for glacier modeling?

c. Is the model stable to perturbations of all wavelengths?  This question is motivated by the observed "very abrupt" pressure changes. Consider, for example, Equation 17 in the supplement to Schoof 2010.  The term $v\_m$ depends on the effective pressure gradient, which suggests that large effective pressure gradients may change the sign of the term in parentheses, and therefore destabilize flow.  Is this analysis correct?  If so, at what wavelengths does destabilization occur?  How are these related to the wavelengths in the previous point.

d. This line of questioning is based in part on my experience with subglacial hydrology modeling in the paper Lipovsky and Dunham (2015, JGR).  In that paper we showed that there is no flow destabilization (at least not at glaciological flow velocities) in a sheet configuration without melting when elastic effects are taken into account (and with other assumptions).

e. Some small points:  should the symbol S in Equation 1a be $S\_{R,ij}$?  Or is S another quantity?  Same with Equation 1b.  Also, $S\_{K0}$ is not defined in the text.

3. **Connections between observation and model**

a. I was disappointed by Section 5.2.  Up to this point, I was carried along in the narrative of the paper:  the reader learns about a dizzying array of new data, their broader interpretation, and then the formulation of a model improvement.  But then I'm not sure what I'm supposed to learn from these simulations.  Is the fit to data good?  Does it capture some of the aspects of the field observations and not others?  Given the ambitious scope of the paper, a much more extensive discussion of these topics is warranted.

b. I would strongly recommend the creation of a new "Section 5.3: Discussion of the Simulations".  There were so many observations in Section 3 that I had a difficult time keeping track of all of them (see later comment).  As written, there is no relationship drawn between Figures  16 and 17 and the main observational results/figures.

c. Near the last line of the paper it is stated, somewhat belatedly, that "However, the ability of the system to fully shut-down requires the incorporation of other physical process that could allow the reactivation of the drainage system during the spring event, something that is probably accomplished by over-pressurization." This should be included earlier, in a potential model discussion section.

d. Is the model capable of describing stage 1, 2, and 3 as defined in Section 4?

e. Does the observed spatial heterogeneity (Section 3) factor into the choice of smoothing length scale?

f. The bottom panels of Figure 17 would be better plotted in terms of water pressure (units equivalent water height) so that they can be easily compared to the rest of the figures in the paper…

g. …Which of the various observed time series should the reader associate with the four panels Figure 17d-g?

4. **Comments on the writing.**

a. There are so many important points in Section 3 that I had a difficult time sorting through all of them. I suggest adding a writing device to emphasize the most important ones. This is partially a stylistic choice. One option would be to enumerate the points at the start. Another option would be to align subsection headings with main points.

b. The manuscript, especially Section 3 and 4, would be improved by revision for brevity. There is a lot of repetition, particularly in Section 3.  The authors mention at least four times, for example, that clustering is subjective.

---

## Referee Comment (RC3) · Anonymous Referee #3 · 5 Apr 2018

General comment

The authors report a new set of observations of water pressure at the base of a glacier. The amount and quality of data acquired in this study are particularly impressive and unique. Based on this comprehensive dataset, a thorough analysis is conducted in order to distinguish typical behaviors of the subglacial hydrology network based on analyzing characteristic spatio-temporal patterns in the measurements. Observations are generally in agreement with expectations from theory, except the finding that many portions of the bed are observed to be hydraulically isolated, a feature that yet is not accounted for in subglacial hydrology models. To overcome this lack, the authors present a modelling framework (based on the adaptation of existing theory) that allows explicitly treating these hydrologically isolated parts of the bed.

Overall, I find the study particularly interesting and novel, since it provides new observational constraints on subglacial hydrology, as well as a unique and comprehensive dataset of interest by a large community. For these reasons I strongly recommend this paper for publication. However, before so, significant revision is needed in order to clarify text in places, better structure observations and clarify results. Below I provide specific comments that hopefully will help the authors to improve this. Moreover, the complexity and lengthiness of the paper is further reinforced by the inclusion of a modelling part at the end. Although I clearly appreciate the modelling effort, I am not convinced that this section really fits in this observational paper. As is I feel like lots of readers won't even notice the modelling part of the paper, especially given the strong imbalance between the long and extensive analysis of data and the short modelling analysis provided at the very end. For these reasons I strongly recommend the authors to consider publishing this modelling work separately, and my comments below are limited to the observational part.

Detailed comments

Section 2

Some context information about the glacier and its environment is missing. I think this information is needed for the reader to make best sense on what type of general glacier and hydrology regime.

What are the typical values for glacier surface speed (in winter versus in summer)? what are the expected sliding velocities (even rough estimates would be useful to know)? Can the authors give a qualitative sense on the potential effects of basal water pressure on glacier dynamics for this glacier and at this particular location where water pressure is monitored? What are typical outlet water discharge values and how much

do they typically vary from winter to summer? Since the the study is motivated by understanding the links between hydrology and sliding (see intro), I think it would be good to give a sense on these aspects to the reader, even if these statements are brief and qualitative.

There is also missing information about how the glacier evolved over the past 8 years during which basal water pressure has been monitored. In particular, did glacier thickness vary over the course of the 8 years of experiment? If yes please give an estimate about how much.

Section 3

Figure 4: I find it quite complicated to identify which hole goes with which measurement. Would there be a way to improve clarity in this figure? Maybe zoom in the map, or make two map subsets to make the color code easier to see.

Line 16 p 7 to line 6 p 8 : unclear text with long sentences.

P 7 to p 8: the whole discussion on what aspects borehole measurements have been grouped is quite vague, and repetitive. It would be good to have a single, short paragraph explaining how boreholes have been grouped, even if the criteria are qualitative (by eyes is a good enough justification), and then go on with the description without repeating how the selection has been done.

Label of Fig 6: amplitude offset? Or phase offset? Looks like it's amplitude.

I suggest to split section 3.1 into two sections. One would be something like "global overview of the dataset" with Fig 4 and 5 and the other would be something like "Diurnal and seasonal cycles in slow and fast flowing water" (Fig 3, 6 and 7). I think this would make it easier to read.

Line 5 to 15, p 12: unclear paragraph. Too long sentences.

Line 10 p 13: Comparing panel b with panel e in Fig 8 I do not see the "inverted" or

anti-correlated relationship. . . Wording and support from figures is confusing here.

Line 28 p 13: Fig. 9 is very lately introduced here. Actually figure 9 seems to help in the understanding of "inverted" or anticorrelated signals, but it comes too late. Perhaps to be place earlier?

P 17: I find the difference between the title of 3.4 (seasonal evolution) and title of 3.1 (annual cycle) to be too weak. . . As is I get lost trying to understand what's new in 3.4 that could not be observed or has not been said in 3.1.

Section 3.6: I suggest to put this section in supplementary material, and just have a single paragraph in the main text that states how and to which extent observations could be biased by changes in data quality. If kept in the main text, this paragraph could even be placed in a separate section before results are exposed.

Section 4

Would be good to have a section or a paragraph that summarizes all key observations, which would be placed outside the discussion section. Then the discussion section would only be based on the summarized, main observations. As is it is embedded and its makes it hard to read.

I don't see what is the difference between 4.4 data interpretation and what's discussed earlier. Isn't the earlier discussion also data interpretation?

Section 5

I suggest to remove that section from the paper, and write a separate paper on the modelling aspects.

---

## Author Comment (AC1) · 18 May 2018

We have found the comments of referees to be very pertinent and helpful, we do appreciate very much their efforts and want to express here our gratitude for their work. In the attached ZIP file we provide a PDF document with the answers to each comment and the modifications we have done to the paper to address them. The document includes the comments of the referees in bold text, and quotes of the paper to provide context to the answers in standard text. The actual answers are presented in blue. The answers to referee #1 start on page 1, for referee #2 (Brad Lipovsky) on page 18 and

the ones to referee #3 on page 27.

The ZIP file also contains the new version of section 5.2 (Model results) and a re-encoded version of the model animation (Movie #2), to make sure that it plays correctly for all referees.

Sincerely,

Camilo Rada

Please also note the supplement to this comment:
https://www.the-cryosphere-discuss.net/tc-2017-270/tc-2017-270-AC1-supplement.zip

---

## Author Response (AR1)

**Author's response**

The following document contains the answers to the referees' comments. All comments were very pertinent and helpful, we do appreciate their efforts very much and want to express here our gratitude for their work.

We start with the answers to each comment and the modifications that we have done to the manuscript to address them. The document includes in bold text the original comments, followed if necessary by a quote from the paper in regular text to provide context to the answers. The actual answers are presented in blue.

The answers to referee #1 start on page 1, to referee #2 (Brad Lipovsky) on page 16 and to referee #3 on page 25. After page 31 follows an annotated version of the manuscript, which includes highlighted text for all deletions and additions, as well as text boxes pointing to changes in the figures and the main ones in the text.

In addition to the specific changes described below, multiple minor modifications have been done with the following objectives:

- Consistency with British spellings
- Consistent use of the words disconnected and isolated
- Consistent use of the words subset and cluster
- Consistent use of hyphenation in compound words.

**Answers to referee #1**

**General Comments**

1. **My primary concern with the paper is the somewhat abbreviated model interpretation and incomplete comparison with previous work. What is there is very interesting, but the rich model record could be compared a bit more thoroughly to the borehole record (here and elsewhere). There is also no consideration for which model parameters this enhanced model might be sensitive to (though I think a detailed sensitivity study is well beyond the scope of this paper). I did appreciate the model technical detail discussion in the SI, and it would be nice to see those topics mentioned in the main text to encourage readers to look into the SI. My biggest concern in this area is what feels like an incomplete comparison to previous work. The authors acknowledge previous investigations of the disconnected (or weakly connected) system, and their model is an elegant extension of previous, simpler attempts to model it (Hoffman et al., 2016). However, the interpretation in this paper is that the subglacial drainage system becomes increasingly fragmented and disconnected as the summer progresses, while previous studies suggest summer brings \*increased\* connectivity (Gordon et al., 1998; Hoffman et al., 2016; Iken and Truffer, 1997; Murray and Clarke, 1995). This difference in interpretation should be discussed, and, if possible, reconciled.**

   R. We have now emphasized in the modeling section (the new version of section 5.2 is provided together with this document) that the model aims to put forward a possible modification of current models (and thereby spur model development by the wider glacier hydrology community), and does not pretend to be capable of reproducing the observations beyond their

generic features. We have added additional material that compares the effect of the model modifications we have made – specifically the addition of a percolation threshold – with results from model runs without those modifications. We show that pressure fields predicted by the model without a percolation threshold are much smoother and show typically diffusive behaviour such as diurnal pressure variations decreasing smoothly in amplitude away from channels (as described in Hubbard et al, 1995), acquiring significant phase lags in the process. We agree that models that try to capture switches in connectivity need to be developed further. That said, we feel that it is worthwhile to suggest at least a direction for development at the same time as making the point that existing models require a connection / disconnection switch. As above, we have amended the modelling section to make this point clearer by including a simulation in which the percolation cut-off is omitted from the model. This also serves both, to underline the point about why the model modification is important, and as a basic "sensitivity test" to illustrate what the most important (and only truly new) model parameter does.

Regarding the comment that "previous studies suggest summer brings **increased** connectivity", we believe our description is compatible with the views presented by those authors given their meaning of "increased connectivity". The subtleties of the discussion require us to carefully distinguish the difference between two possible meanings of "increased connectivity":

1. **Increase in efficiency of the connected system**: Due to its transition to an efficient channelized system (leading for example to shorter tracer transit times)
2. **Increase in the spatial extent of the connected system**: Enlarging the area affected by pressure variations in the connected system.

Firstly, connectivity increases dramatically at the spring event, and secondly, the development of the channelized system makes the system more efficient (while not necessarily occupying a larger part of the bed).

Some comments particular to each cited author:

- **Murray and Clarke, 1995**: Recognizes the existence and relevance of unconnected domains of the bed, and the heterogeneity and dynamism of such domains. However, with the limited timespan of their borehole records, they do not put forward any description regarding the evolution of such systems throughout the season. Therefore, there are no inconsistencies with our description.
- I**ken and Truffer, 1997**: They also highlight the importance of unconnected reaches of the bed and propose that an increase of their extent is responsible for the seasonal and multi-year slowdown of Findelengletscher glacier. Our interpretation is very consistent with theirs, as it can be illustrated by the following statement "We interpret the decrease in velocity and water pressure during the melt season as being caused by the formation of R channels at the expense of parts of the linked-cavity system.".
- **Gordon et al., 1998**: Their description of the evolution of the subglacial system is completely consistent with our interpretation., where a poorly connected set of boreholes undergo a transition as the season progress. This leads to some boreholes becoming part of an efficient drainage system, others to become completely isolated and others exhibiting a transitional behavior. The relative lack of boreholes that became isolated can be attributed to the fact that the study does not include the late-season shut-down of the drainage system (Stage 2 to 3 transition), and perhaps also because the study area

was focused in the region where a major channel was predicted to exist during the melt season.

- **Hoffman et al., 2016**: Our interpretation is also consistent with these authors. However, the interpretation of Hoffman et al. (2016) as showing "increased connectivity" over summer is not the same notion of connectivity we are using here. They attribute the late summer slow down to a pressure drop in a weakly connected system, something that is consistent with our data.

  In our interpretation, the alternative hypothesis they present for late summer slow down (discarded due to model results) seems equally likely, that is: attributing the slowdown to an increase in the size of the isolated domains of the bed. The model used by Hoffman et al (2016) does not dynamically change the portions of the bed that are connected or disconnected; instead, slow diffusion allows the low-frequency components of the pressure signal to be transmitted to the "weakly connected" (as opposed to completely disconnected) parts of the bed. This is in fact acknowledged in our study as well (Fig. 7), and a potential feature of the model we propose through the parameter k_leak.

2. **It would be nice to see a bit more acknowledgement of the possibility of borehole behavior being governed by the presence of subglacial till. This is discussed briefly in a few places, but the paper would benefit from additional consideration of it, or a stronger justification for a dominantly hard bed interpretation. For example, on p. 27 there is discussion about localized diffusive systems with limited flow, and it seems like till would fit the bill.**

   R. This is a good observation. We have now considered it by adding the the following:

   The paragraph in page 2, 5-8 has been changed to the following (the relevant text added is underlined ):
   "The extent to which water pressure is raised by increased water supply depends on the following three factors: the permeability of till underlying the glacier, the configuration of conduits, both at the bed and in the ice, and the storage capacity of the drainage system, which can act to buffer the effect of additional water supply. "

   In the discussion (section 4.2) after the line 7 of page 27 the following paragraph will be added:
   "Hubbard et al. (1995), suggest that the bed substrate at their study site is composed of glacial till of varying grain size distributions, acknowledging that "a network of small channels" on a hard bed could also account for their observations. However, in terms of hydrology, till and a distributed drainage system at the ice-bed interface share many characteristics: we expect both to give rise to a diffusive model for water pressure if water storage in the distributed system is an increasing function of water pressure. The primary difference is in how the permeability of that system evolves. In the 'hard-bed' view, the permeability evolves over time in response to changes in effective pressure, whereas for a granular till, porosity and therefore permeability are simply functions of effective pressure and therefore respond instantly to changes in it (Flowers, 2015). The main inconsistency of appealing to drainage through continuous till layer as the main pathway for water flow is that we would expect to see more standard diffusive behaviour, and certainly no sharp switches between connected and disconnected portions of the bed. In addition, till with a sufficient coarse-grained fraction of cobbles and boulders would probably be capable of supporting the formation of cavities in the lee of those larger grains. In short, if till

is capable of creating cavities, or is interspersed with bedrock bumps or somehow capable of supporting switching events by other means, then our interpretation would not be affected by assuming a hard or granular bed."

3. **p18, lines 2-3: How much does snow cover or ice albedo change? It is a questionable assumption that the degree day factor remain does not change significantly as surface conditions change over the summer. The interpretation in Figure 11c seems rather tenuous.**

   R. That is a very good point. And indeed the signal we base our interpretation on, could arise due to changes in the degree-day factor through the season. Therefore, we have done further research to asses the variability of degree-day factors and computed the relative amplitudes using an independent proxy of melt variability coming from surface elevation measurements by a sonic ranger at the AWS location. To address this question, we have added a short section (section 2) to the supplementary material. The new section study the variability of degree-day factors, and computes the relative amplitude of Fig. 11c using surface lowering as proxy for melt amplitude instead of the standard deviation of the positive part of temperature.

4. **The title is fine as it is, but it is worth considering the title somehow including some- thing about the importance of the disconnected/weakly connected system in the interpretation, as this is a primary result.**

   R. We have changed it to:
   "Channelized, distributed, and disconnected: subglacial drainage under a valley glacier in the Yukon"

5. **As mentioned above, the borehole results section is quite long.**

   R. We have reduced the results by moving section 3.6 to section 1 of the supplementary material as suggested by referee #3. References to it were added to the Methods and Discussion.

**Specific Comments**

- **Abstract seems a little short given the length of the paper, but it does hit the most significant highlights of the paper.**

  R. We agree, but given the consideration that it does hit the most significant highlights of the paper, we will keep it unchanged.

- **1, 22: basal "slip" may be considered the preferred term here (Cuffey and Paterson, 2010), to acknowledge the ice is not sliding differentially from the substrate at its sole.**
  "A similar effect is observed on glaciers resting on a till layer, where a lower N reduces the yield stress of the till, and therefore also enhances basal sliding"

  R. We have added in parentheses: "sliding is here intended to include motion at shallow depths within the till layer as well as at the ice-till interface". From the perspective of large-scale ice

motion, both processes appear as the same thing.

- **2, 6: No comma here.**

  R. Removed

- **2, 6-11: Interactions between subglacial hydrology and ice motion could be mentioned here as well (e.g., Hoffman and Price, 2014). And Gordon.**
  "The extent to which water pressure is raised by increased water supply depends on both, the configuration of conduits and the storage capacity of the drainage system, which can act to buffer the effect of additional water supply. In turn, the conduits that make up the drainage system can change in response to changes in water input, as the associated changes in effective pressure affect the rate at which viscous creep closes subglacial or englacial conduits. Changes in discharge also affect the rate at which wall melting enlarges conduits. Over time, the response of the drainage system to the same water input pattern can therefore change (Schoof, 2010)."

  R. That is something worth mentioning. And the new version of the paragraphs does it (relevant changes underlined):
  "The extent to which water pressure is raised by increased water supply depends on the following three factors: the permeability of till underlying the glacier, the configuration of conduits, both at the bed and in the ice, and the storage capacity of the drainage system, which can act to buffer the effect of additional water supply. In turn, the conduits that make up the drainage system can change in response to changes in water input, as the associated changes in effective pressure affect the rate at which viscous creep closes subglacial or englacial conduits. Changes in sliding (themselves due to changes in effective pressure) will also affect the opening of basal cavities (Hoffman and Price, 2014), and changes in discharge affect the rate of conduit enlargement by wall melting. Therefore, over time, the response of the drainage system to the same water input pattern can change (Schoof, 2010)"

- **2, 29: missing "a" -> "to provide a less efficient"**

  R. Added

- **2, 33: "do" should be removed.**
  R. Removed

- **2, 32-34: (Creyts and Schoof, 2009) could be an appropriate additional reference for this topic.**
  "Unlike channels, multiple cavities can co-exist in close proximity, because a larger cavity size facilities faster creep closure rates, while the opening rate is generally assumed to do not depend significantly on size. Therefore, larger cavities will tend to close faster and converge to equilibrium with small ones (Kamb et al., 1985; Fowler, 1987)."

  R. Citation added

- **3, 5: I believe you meant for the second "channels" on this line to be "conduits".**

"The formation of channels can be understood as an instability in drainage through a distributed network of channels,"

R. Yes, changed

- **3, 9-15: This is a nice summary of the complexity in borehole observations. A few suggestions for additional references: "widespread areas of high water pressure during winter": (Ryser et al., 2014; Wright et al., 2016) "large pressure gradients": (Fudge et al., 2008) "sudden reorganizations": (Gordon et al., 1998) "anti-correlated temporal pressure variations": (Andrews et al., 2014; Ryser et al., 2014)**

  R. Great suggestions. We have included all of these.

- **3, 31: "in-deep" -> "in-depth"**

  R. Corrected as suggested

- **5, 33: Could mention that many authors refer to this as "hydraulic head".**
  "In the present paper, water pressure values will be reported in metres of water (the height of the water column that would produce that pressure)."

  R. We would rather not change the text in this case, because "hydraulic head" includes the offset due to the elevation of the base of the glacier (base height + pressure/(rho_w*g)), so just writing pressure/(rho_w*g) = head would be incorrect. Therefore, to avoid confusion we decided to describe what we mean instead of referring to the concept of "hydraulic head".

- **Figure 2: I recognize that showing so many different symbols and colors is challenging, but it is difficult to differentiate some of them. Perhaps removing the 3d shading on the symbols would help. In particular, the red symbols are hard to make out. Maybe put a circle around them or something to make them easier to see. Also, the black and blue lines are difficult to tell apart. Finally, the concept of "upstream area" from Schoof et al., (2014) should be briefly elaborated on (either in the caption or the text).**

  R. We have made the symbols easier to differentiate, both by removing the shading effect and increasing the size. To explain better the "upstream area" concept we have changed the last sentence of the caption to "Grey shading indicates the upstream area, calculated as- suming an hydraulic gradient given by an effective pressure equal to half of the ice overburden pressure, and computed using the D∞ method described by Tarboton (1997)"

- **Figure 3: The figure is a bit small in my printout. In particular, the green dots are hard to see.**

  R. We have changed the points by bars to make the visualization clearer.

- **7, 8: Consider changing "Fast-Flow" to "Fast Water Flow". When I first read this I interpreted this to be a region where ice velocity is fast.**

"On July 28th, 2013, while installing a sensor in the hole marked "Fast-Flow" in Fig. 2, strong periodic pulls were felt through the sensor cable, revealing a conduit with turbulent, fast flow in the bottom 50 cm of the borehole. This borehole was also the only one in which there was an audible sound of flowing water. The fast-flow hole was drilled at the very end of the field operations, and no further detailed on-site investigation was conducted."

R. Due to the numerous references to the "Fast-flow" borehole we rather keeping that short name. However we agree that the way we present it in the first instance is confusing. Therefore, we have modified the above paragraph so that the context is better explained before introducing the "Fast-flow" short name. The final paragraph will read as follows:
"On July 28th, 2013, while installing a sensor at the bottom of a borehole, strong periodic pulls were felt through the sensor cable, revealing a conduit with turbulent, fast water flow in the bottom 50 cm of the borehole. This borehole was also the only one in which there was an audible sound of flowing water. The location of the hole is marked as "Fast-Flow" in Fig. 2, it was drilled at the very end of the field operations, and no further detailed on-site investigation was conducted."

- **8, 31: bummer**
  "After a data gap caused by a corrupted compact flash card, the records have become more dissimilar by August 2nd, but continue to exhibit common pressure variations."

  R. Indeed

- **Figure 4: The colors are a bit difficult to match to the map. Again, perhaps removing the 3d shading of the symbols on the map would help.**

  R. We have enlarged the symbols and removed the 3D shading.

- **Figure 4 caption: in part c), it says two sensors were installed here, but I only see one line in the plot. Clarify if they are plotted on top of each other or if only one is plotted and, if so, which and why.**
  "(c) Pressure in the fast-flow borehole (red) and its correlation with temperature in grey, computed for any given time over a 3-day running window. Note that two sensors were installed in the fast-flow borehole, offset vertically from each other by 70 cm."

  R. The two lines are indeed on top of each other most of the time, but they can be distinguished in the periods with no diurnal oscillations. We have changed the last sentence to: "...offset vertically from each other by 70 cm, making the two lines indistinguishable most of the time at the presented scale. Later in section 3.5, the complete record will be displayed, where the two curves are more distinguishable."

- **Figure 5: Is correlation to temperature calculated for panel c here as it was for Figure 4? If not, mention that in the caption.**
  "Panel c shows pressure in the slow-flow borehole (black) and three other boreholes in the same line."

R. No, it wasn't, because all the lines in the panel undergo disconnection events, so it does not make sense to produce a mean of all of them, or to pick one. However, as the correlation is relevant to the discussion. Therefore we have picked one timeseries that remains connected over the whole interval and computed the correlation of it with the temperature record. The correlation data has been added to the figure and the relevant lines of caption modified to:
"Panel c shows pressure in the slow-flow borehole (black) and three other boreholes in the same line. The correlation with temperature has been calculated using the only borehole that remains connected over the whole interval."

- **11, 18: Should "ice" be "water" here?**
"...it is therefore possible that more boreholes intersect conduits with fast-flowing ice,..."

    R. Indeed. Corrected.

- **13, lines 1, 2, 3, 5, 7, 9: Include figure number with each panel reference.**

    R. Good suggestion. The references to the panels have been changed to include figure numbers like "panel 8c". The same change have been applied in many other instances for consistency.

- **Figure 8: Caption for g) refers to six digital sensors included in panel b, but panel b is the temperature record.**

    R. Yes, it should have said "panel c". Corrected.

- **15, 20: The second comma should be removed.**

    R. Removed.

- **Figure 10: Panel c is pretty hard to make out details of. Perhaps this figure could be reorganized into two columns, or panel c could somehow be made a bit larger.**

    R. This figure was reduced to fit in one page using the discussion paper template. In the current version it was enlarged to full size.

- **17, 3: Fig. 8b must be an incorrect reference - do you mean Fig. 10C?**
"We have described the apparent spatial patterning of the drainage system above. This patterning is however not fixed but evolves over time. In Fig. 8b, it is clear that all 42 boreholes show very coherent temporal pressure variations at the start of the observation period."

    R. Yes it was incorrect. It should say figure 8c. It was corrected.

- **Figure 13: Consider putting earlier on, perhaps with Figure 3.**

    R. Good suggestion. We have moved it up, corresponding now to Fig. 3. And the following text was added to the end of the caption: "The interannual variability evident in the photo will be discussed in section 4."

- **Section 3.6: The data quality section would be more natural in section 2 (methods), than late in the results section.**
R. As mentioned above, section 3.6 have been moved to section 1 of the supplementary material. The corresponding references were added to sections Methods and Discussion.

- **22, 10: I think you mean "120% of overburden", not "above".**

  R. Indeed. Corrected.

- **22, 11: Consider replacing "and" with "however" or "yet" to make it clear you are arguing against sensor drift being able to explain these observations.**

  R. Good suggestion. It was replaced by "yet".

- **23, 12-14: This text would flow better with this sentence in parentheses.**
"We will refer to this initial state of the subglacial drainage system as stage 1. Note that the "stages" identified here are not the same as the "phases" discussed in Schoof et al. (2014), who focused only on the later part of the melt season and the subsequent winter; for instance, phase 2 in Schoof et al. (2014) corresponds to the transition from stage 2 to 3 here."

  R. Although we agree with the suggestion, we were previously advised by the editor to avoid sentences completely in brackets. So we will make no changes in consideration of the editorial guidelines.

- **25, 5: (Hubbard et al., 1995) could be an additional appropriate reference here.**

  R. Good suggestion. Added.

- **25, 14: An aside: water pressure in nearby moulins/crevasses would be useful here. Something to consider if this field campaign continues.**

  R. We agree. Unfortunately there are no moulins in the study area, and most of the crevasses in appear to be shallow (10-15 m deep) and generally do not having standing water in them. We have considered instrumenting the area below from a moulin in future work.

- **25, 21: Some discussion of bridging stresses leading to isolation of low pressure channels would be good here (Hewitt, 2011; Lappegard et al., 2006).**
"These observations are consistent with a highly developed channel with higher water discharge that has become hydraulically isolated from the neighbouring bed: the high effective pressures in the channel would favour the closure of cavities or other connections at the bed."

  R. It is indeed an important process to include. We have edited the above paragraph to the following:
"These observations are consistent with a highly developed channel with higher water discharge that has become hydraulically isolated from the neighbouring bed: the high effective pressures

in the channel would favour the closure of cavities or other connections in the surrounding bed. This closure may also be enhanced due to the effect of bridging stresses (Lappegard et al., 2006). Bridging stresses transfer part of the weight of the ice overlying the channel to its surrounding bed, effectively increasing the ice overburden in those regions above its mean value (Weertman, 1972)."

- **25, 23: Also, Figure 3.**

  R. Correct. Cross reference added.

- **25, 29: Mention that high up-stream areas means a likely water flow accumulation path (see comment above about introducing the significance of this upstream area).**
  "Using the channel end-member feature of diurnal oscillations with pressure dropping to atmospheric at night, we have identified seven other boreholes where the drainage system is likely to have evolved into a well-developed channel (Fig. 2, red symbols), in all cases during the second half of July or first days of August during years with relatively high cumulative PDD, which ought to favour channel formation. Their locations loosely match zones with high up-stream areas (Fig 2, dark shading)."

  R. In addition to the details already added to the caption of Fig. 2, at the end of the above paragraph we will add:
  "..., which correspond to portions of the bed likely to concentrate basal water flow due to the expected hydraulic gradients."

- **26, 1-7: This discussion would benefit from inclusion of (Meierbachtol et al., 2016).**
  "Initially, creep closure will reduce any volume still occupied by air in the borehole and pressure can rise gradually; once there is no air space left, changes in water pressure must reflect the pressure required to maintain the borehole volume constant (assuming no further freezing) while the borehole may still deform under anisotropic stress conditions. Intuitively, we would expect the borehole to become flattened perpendicular to the direction of greatest compressive stress, requiring a larger borehole pressure to maintain a constant volume, which could account for the slow rise observed in water pressure, and possible for slightly above-overburden values. Importantly, the pressure in an isolated borehole should depend on its shape and can, therefore, differ from borehole to borehole; abrupt creation of new storage volume for instance due to crevasse propagation could also lead to abrupt changes in pressure in isolated boreholes."

  R. Indeed a good reference. We have included it in "... may still deform under anisotropic stress conditions (see also Meierbachtol et al, 2016)"

- **26, 28: "fragment into subsystem" -> "fragmented into subsystems"**

  R. Corrected

- **27, 3-7: Interesting discussion. I think the quotes around "phase lag" should be removed.**

R. Quotes removed

- **27, 21: Is the distance long enough relative to channel flow speed for a phase lag to be expected? The other complication is there could be additional inputs of water from the surface that help to "lock" the channel phase to the surface phase even in the presence of diffusion within the subglacial system.**

  "The pressure time series along the inferred channel system in Fig. 4 (panels c, e and g) are merely suggestive of a hydraulic connection, but hardly identical. [...] Importantly, however, there is no systematic phase lag accompanying the decrease in amplitude, as would be predicted by a diffusion model (Hubbard et al., 1995)."

  R. In a diffusive system, the observed drop in amplitude should (with a single water input) correspond to a predictable phase lag. Therefore, if we observe a given drop in amplitude, we can predict the phase lag. That calculation involves a modeling exercise that we have decided not to include in the paper to avoid adding to its already considerable length. The point about additional water inputs is a fair one. To address it we have added the following at the end of the paragraph:
  "It is however conceivable that additional water input from surface sources along the flow path can have a significant effect on the phase of the pressure signal."

- **27, 27-29: Consider (Meierbachtol et al., 2016) again here.**

  "Usually, disconnection occurs during a drop in water pressure in the subsystem, and reconnection during an increase (figures 6 and 8). This is consistent with connection or disconnection resulting from viscous creep closing connections between individual cavities within the distributed system (Kamb, 1987). Disconnection could also be the result of cavities shrinking while remaining connected, if the borehole simply terminates on an ice-bed contact area between connected cavities and those contact areas are systematically larger than the $\sim 10$ cm diameter of our boreholes."

  R. We have added "This process has been observed previously by Meierbachtol et al., (2016) ."

- **27, 32: There is an alternative hypothesis as well of passive cavity opening due uniform basal sliding (Bartholomaus et al., 2011; Hoffman and Price, 2014; Iken and Truffer, 1997).**

  "The anti-correlated signals we observe in our data (Fig. 8e) have previously been explained by a mechanical load transfer mechanism, where the ice around a pressurized conduit redistributes normal load, reducing the normal stress over neighbouring areas of the bed. Therefore unconnected water pockets in those areas would experience a drop in water pressure (Murray and Clarke, 1995; Gordon et al., 1998; Lefeuvre et al., 2015). A 3D full Stokes model presented by Lefeuvre et al. (2018) supports this interpretation, and suggest that the anti-correlation pattern depends on the bed slope, which can be one of the factors affecting the observed distribution of borehole displaying this behaviour. Boreholes exhibiting those anti-correlated pressures must then be effectively isolated, so that a change in normal stress mainly causes changes in the pressurization of the borehole rather than water exchange. The load transfer mechanism is consistent with our observations."

R. Good point. We have considered the alternative explanation by adding:

"An alternative explanation suggests that such signals are associated to enhanced cavity opening due to basal sliding changes (Bartholomaus et al., 2011; Hoffman and Price, 2014; Iken and Truffer, 1997). However, it is unlikely that a variation in sliding would precisely mimic the local water pressure variations in the adjacent drainage subsystem, as suggested by Fig. 9e: the force balance that determines sliding velocities should be affected by changes in basal shear stress across a larger portion of the bed"

- **28, 8: These island sound like the system described by (Murray and Clarke, 1995).**
  "It would be difficult to explain the anti-correlated signal in these boreholes by normal load transfer over larger distances, when other isolated boreholes nearby show no such behaviour. This suggests that the connected drainage system can contain fine structure (either as channels or narrow regions of distributed drainage) with lateral extents smaller than the ∼ 15 m borehole spacing. The same is indicated by the formation of disconnected "islands" in lines of otherwise connected boreholes at the same spacing as seen in Fig. 6 for the August observation period."

  R. Indeed, we have added the following at the end of the paragraph: "(see also Murray and Clarke (1995), for analogous observations)."

- **28, 19: Wouldn't disconnected areas act as \*slippery\* spots since they maintain high water pressure?**
  "As in Hoffman et al. (2016), such a slow evolution could be accounted for by flow through a relatively impermeable till aquifer underlying a much more effective but less pervasive interfacial drainage system, and the magnitude of that leakage could have a significant impact on basal sliding rates if disconnected areas act as sticky spots."

  R. It is indeed very important to explain dynamic effects of isolated cavities. With this propose in the Introduction we have added the following paragraph (After line 4, page 3):

  "If a cavity becomes disconnected, its fixed volume will result in a water pressure drop if sliding accelerates. Conversely, decelerating basal sliding will lead to relatively high water pressure in order to prevent creep closure, reducing basal drag. In other words, isolated cavities can act either as sticky spots when basal sliding speeds up or as slippery spots when it slows down, working as a buffer for basal sliding variations (Iken and Truffer, 1997; Bartholomaus et al., 2011)."

  However, we want to avoid interpreting our isolated boreholes as representative of isolated cavities, the high water pressures we observe are probably not representative of "ambient" water pressures in the disconnected areas, at least not necessarily. If a borehole connects to nothing but an isolated section of the bed, its pressure would be a passive measure of how hard the borehole is being squeezed by the ice, including stresses acting on the vertical wall of the borehole, that tells us nothing about the bed. It's still quite conceivable that there is strong ice-bed coupling near such boreholes. The assumption that these boreholes connect to cavities at the bed is what we want to avoid.

  To clarify this in the paper, we have added the following at the end line 7 in page 26

(Discussion):

"Therefore, we have to caution against interpreting the pressure in individual disconnected boreholes as an indication of the conditions in the unconnected parts of the bed: instead the borehole pressure may be controlled predominantly by local stresses in the ice, and the orientation, volume and shape of the unfrozen portion of the borehole."

- **28, 22-26: This is a significant result and well-stated here**.
  "Although it is possible that some boreholes do not connect because they were not properly drilled to the bed, we believe that the existence of persistently disconnected areas is robust. Non-spatially biased samples suggest that up to 15% of the bed could remains unconnected year round. The existence of such unconnected holes, and the possibility of dynamic connection and disconnection, represents a challenge to existing drainage models, which typically assume pervasive connections at the bed."

  R. Thanks

- **30, 3: "differential motion between ice and till": If basal slip is primarily due to till deformation, then there will not be differential motion between ice and till.**
  "Nevertheless, the lifespan of a sensor buried in the till ought to be short if there is differential motion between ice and till, causing the signal cable to tear."

  R. To clarify we will rephrase it as "... if there is differential motion between ice and the sensor placement in the till (e.g. Engelhardt and Kamb, 1998), causing the signal cable to tear."

- **31, 11: Is there a significance to the designation 'K' or is it just an arbitrary letter choice?**

  R. "K" is for Kamb, "R" is for Rothlisberger. To clarify, in that first mention of 'K'-conduits we now refer to them as " 'Kamb' (K) conduits".

- **31, 11: Please define n_c.**

  R. See answer to next comment.

- **31, 11: Consider adding "along that edge" after "n_c-1 'K' conduits" to emphasize that this treatment is per edge.**
  "Along each network edge ij, we assume one 'R'-conduit that can behave either as a Röthlisberger (R) channel or a cavity, as in Schoof (2010), with average cross-section $S_{R,ij}$. To mimic the sheet of Werder et al. (2013) and avoid the pitfall of having to resolve every basal conduit, we also assume there are $n_c − 1$ 'K'-conduits that behave only as cavities, and are not subject to enlargement by melting."

  R. The paragraph have been edited to: "Along each network edge ij, we assume there are $n_c$ conduits connecting node i to node j: One 'R'-conduit that can behave either as a Röthlisberger (R) channel or a cavity, as in Schoof (2010), with average cross-section $S_{R,ij}$, and $n_c$-1 'K'-conduits that behave only as cavities, and are not subject to enlargement by melting. This

configuration mimics the sheet of Werder et al. (2013) and avoid the pitfall of having to resolve every basal conduit"

- **Equations: It seems odd to use lettered sub-equations rather than a new number for each equation.**

  R. Indeed, it is pointless in this case. We have changed the numbering to one number per equation.

- **31, 21: Also define Psi here.**
  "We associate a nominal effective pressure N_i with each node, defined as overburden minus basal water pressure. Hydraulic potential Φ_i at each node and hydraulic gradient along the conduits are then given by"

  R. The above paragraph have been modified to "We associate a nominal effective pressure $N_i$ with each node, defined as overburden minus basal water pressure. Hydraulic potential $\Phi_i$ at each node and hydraulic gradient Ψ along the network edges are given by..."

- **Eq. 1d/e: A minor quibble: It would seem more intuitive if the threshold size also contributed to flow once the threshold is reached (which is not the case in 1d/1e). However I doubt the choice of how to treat that affects the results in a qualitative way, so either approach is defensible.**

  R. We don't fully understand the meaning here. The threshold size does appear in the formula for flux even once the threshold is exceeded: in that case, discharge Q is proportional to (S-S_P)^alpha, where S is conduit size and S_P the relevant threshold (we have omitted the other subscripts for implicity).

- **Eqn. 1f/1g: Mention this is describing mass conservation to aid the reader. Also, this is a single equation so there should be a single label.**
  "To account for conservation of mass, we also associate half the volume of water stored in a conduit between two nodes with each node, and likewise account for half the water created by wall melting in an R-conduit as water supply to each node. Consequently we impose"

  R. To emphasize that just before presenting the equation, we have modified the last sentence to " Consequently we impose mass conservation in the form"

- **32, 17: This is a run-on sentence. How about ending it at "nodes" and starting a new sentence with "We".**
  "To close the model, we need to relate the conduit effective pressure P_{e,ij} to the nominal effective pressures N_i at network nodes, we write this in the form"

  R. Changed as suggested.

- **33, 7: Is (Dow et al., 2015) meant here?**
  "A key component that the model above continues to miss is the ability to open conduits due to

overpressurization of the system (Schoof et al., 2012; Hewitt et al., 2012; Bueler and van Pelt, 2015; Dow et al., 2016).”

R. Yes, corrected.

- **5.2 It would be clearer to call this section "Model Results".**
  “5.2 Results”

  R. Changed as suggested.

- **36, 1: "eventually" should be "eventual".**
  “The eventually complete shut-down of the entire drainage system at the end of the summer season is presumably the result of low water supply: high effective pressure and low dissipation rate in channels allow basal conduits to close.”

  R. Indeed, changed.

- **37, 11: The word "a" should be removed.**
  “We have implemented this approach in a simple model, allowing us to reproduce qualitatively some of the main features of our data set: a sharply-defined drainage subsystems with insignificant diffusive pressure signal attenuation and the existence of isolated areas (See section 5.1).”

  R. Removed

- **Data availability: What about model and model configuration and output? Mention it is included in the SI.**
  “Data availability. The presented data set will be made publicly available in the future. Ongoing work is taking place to meet the format and create the ancillary data and documentation required for the release, that is expected to happen fully or partially by the end of 2018. In the meantime, it can be accessed on request to the corresponding author.”

  R. We have added “The model code in Matlab and model configuration parameters are included in the supplementary material.”

**Supplemental Material**

- **paper_movie.mpg does not play for me.**

  R. We have re-encoded it to a widely compatible format, and it is attached to this submission.

- **It would be more natural to switch the order of sections 1 and 2 to match the order these topics were presented in the main text.**

  R. It does make sense. We have switched the order.

- **The SI material, particularly the modeling part, has some very useful information. I would like to see the main text refer to the SI in more places, with brief descriptions of what is found there.**

  R. We have added multiple references to the supplementary material where pertinent. In particular there are five references to the Continuum model formulation.

**Answers to referee #2 (Brad Lipovsky)**

**1. Additional questions about the observations/interpretation (Sections 2-4)**

**a. Given the complexity of the spatial patterning, would it be possible to make a movie that plots all the data? I envision the map in Figure 2 with each symbol having a color that is associated with a pressure scale. This should be feasible given the low sampling rate. There's only so much that can be conveyed with words.**

R. We have indeed considered and tried such visualization. However, we have decided to include it in a follow-up paper tackling the challenge of automatic clustering of time series for identification of boreholes subsystems. The reason for this is that the large variability of the unconnected sensors, the heterogeneity in the behaviours across short distances and the dynamism of the hydraulically connected subsystems, makes impossible to really distinguish any patterns in a visualization like the ones you propose unless we ignore/fade selected sensors and wisely color the subsystems of interest. Therefore, as the techniques we have developed for the identification and follow up of subsystems in time were beyond the scope of this paper, and are required to justify the decisions that have to be made to make such visualization useful, we have decided to do not include it here, but it will accompany our next paper.

**b. How long does drainage of the borehole take upon connection to the bed? This timescale is mentioned only qualitatively in the manuscript. Early work by Kamb and Englehardt used this timescale to estimate properties of subglacial conduits.**

R. Unfortunately, the drilling rod used does not have the capability to record water pressure at the tip and we did not deploy any instrumentation to record water level during the drilling process. Therefore, we were not able to measure the water level drop during drainage events. For that reason, drainage events were treated qualitatively, and we did record whether they happened or not and at what approximate depth was the drill tip at that moment. To clarify this, after the sentence starting on line 6 on page 15:
"Drainage events occurred during drilling at all depths, but more frequently at greater depths, with 60% (59%) happening in the lower half of the boreholes. This remains true for the 2012 drilling campaign, where the first sensors were installed before the spring event and observations are likely to reflect winter conditions."
We have added the following:
"Unfortunately, water level change and duration of drainage events were not recorded."

**c. Relative amplitude of pressure and temperature. Interquartile ranges (instead of standard deviations) may be more useful given the orders of magnitude variability.**

R. Indeed IQR would be more resilient to outliers. As per your suggestion we have tried IQR. The following two figures are a rough version of figure 11, panel c, the left one uses standard deviation, and the right one uses IQR

As can be seen, the differences are subtle and IQR does not provide a substantial improvement. On the other hand, it is a less common way to measure data dispersion, that might not be known

to many readers. For that reason, we have decided to keep the current definition of relative amplitudes using standard deviations.

[Figure]

**d. Is it possible to quantify how fast switching events or connection/disconnection occur? For example, on page 27 line 26 [24]: "very abruptly in time". What does that mean, exactly? Do transitions ever occur faster than the sampling resolution?**
"We have referred to boreholes that cease to exhibit diurnal pressure variations as having disconnected. Connection and disconnection typically manifest themselves very abruptly in time (Fig. 6, see also Fig. 5 of Murray and Clarke (1995))."

R. We mention the rough timescale when introducing the concept of switching event on page 19, line 3:
"In most cases, however, the transition is abrupt, and the same is true of boreholes connecting with each other: a rapid change in water pressure can occur over the course of a few hours or less as a connection is established. We term such abrupt transitions "switching events", following Kavanaugh and Clarke (2000) ."

We do not want to go much deeper into the details of switching events, but we do agree that it is important to mention the lower limit of the timescale and relate it to our sampling frequency. For that reason we have added the following after the paragraph on page 27 lines 23-24 cited above:
"This transition usually takes from few tens of minutes to a few hours. However, the initiation of the transition, often identified as a clear change in the rate of change of pressure with respect to time, can in many cases have the appearance of an instantaneous phenomenon, even at our shortest sampling interval of one minute. Therefore, it is unclear if these time scales can be associated with the connection or disconnection process, as they might only represent how fast the system responds to a perhaps instantaneous switch between connected and disconnected states."

**e. What does the pressure sensor response curve look like with and without the snubbers? Do the snubbers limit the ability of the sensor to measure high-frequency water pressure oscillations?**

R. That is indeed an important consideration. We have now addressed it by changing line 29 on page 5 to:
"Most transducers installed from summer 2013 onwards were equipped with a Ray 010B ¼"

brass piston snubber that act as protection against transient high-pressure spikes, without altering the signal at the sensor sampling frequencies as verified by doubly-instrumented boreholes (see Supplementary Material Section 1)."

In the supplementary material, we have added the following text at the end of Section 1 (previously Section2: "Doubly instrumented boreholes: a test of pressure measurement reliability"):

"Doubly instrumented boreholes also allow us to assess the effect of pressure snubbers on pressure records. In figures 3, 4, and 7 to 11, sensor P1 (blue) was equipped with a snubber and P2 (orange) was not. In figures 1 and 2, both sensors had snubbers, and in figures 5 and 6 neither of them did. In the cases where only one sensor had a snubber, it can be seen that no smoothing of the pressure signal is observed. Close examination of the pressure time series shows that even spikes lasting a few minutes are well-reproduced by both records. Therefore, at the sampling frequencies of our sensors (1 to 20 minutes), the effects of the snubber are negligible. By contrast, among the sensors not equipped with a snubber, one out of 48 suffered large, "instantaneous" pressure offsets, contrasting with only one in 174 experiencing the same among sensors equipped with a snubber. Therefore, pressure snubbers seem to be effective at filtering out the transient high-pressure spikes ("fluid hammer") that are thought to be responsible for those offsets through damaging the sensor diaphragm, but without affecting the accuracy of the instruments for measuring slower pressure variations."

**2. Questions about the model (Section 5)**

**a. A broader question regarding this type of modeling (i.e., also applicable to Schoof, 2010; Werder et al., 2013): Are conduit models convergent under grid refinement? Werder et al. (2013) in their Appendix A discuss grid densification. As those authors pointed out, this creates complexities associated with changing the domain geometry. But what refinement is undertaken in such a way that more grid nodes are added only at the midpoints between existing grid nodes. Does the model converge under this narrower sense of grid refinement?**

R. Though the question is perfectly legitimate, the answer is really beyond the scope of the paper (and pertains as much to the Schoof 2010 and Werder et al 2013 papers as anything else). In short, network-based conduit models are not convergent under refinement, and are not intended to be. Effectively, they are intended to capture all the conduits at the bed individually. The reason for this should become apparent shortly.

Recognizing that capturing a large number of conduits may not be computationally feasible is what motivates the use of a continuum sheet overlain with a network of "potential channels" in Werder et al, and the use of the "K"-conduits of cross-sectional area S_K in the present case. The hope would be that, if we doubled the number of nodes and hence of network edges through any give line drawn through the domain, then halving the number n_c of total conduits per network edge would give some sort of effective convergence, although in practice the R and K conduits still behave sufficiently differently for that not to be entirely the case. The dependence on network orientation will of course remain, and is one of the bigger obstacles that remain in drainage modelling (in fact, it would be great to be able to evolve the geometry of channels and let them meander etc, but the real challenge in doing that would probably arise

when we try to couple them with other drainage conduits.

Based on what we have just said, the reason for not expecting convergence under "refinement" alone (meaning, just adding network edges) is therefore fairly straightforward – adding extra network edges then (without changing n_c) just corresponds to physically adding extra conduits.

The reason for using a non-conitnuum method (in the sense that we do not have convergence under refinement in the usual sense) is that channels that actually behave as R-channels (whose size is dictated by a balance of dissipation-driven thermal erosion and creep closure) cannot co-exist in close proximity. Consequently, channelization is intrinsically a process that does not lend itself to standard continuum description, where intensive quantities (like a channel density) would need to be used. The problem is that only a single channel will ultimately survive locally, while a density-based description would allocate a number n = density*(grid cell size) of such channels to a given grid cell, and that has no hope of convergence, as the flux going through those channels, and hence the rate of wall erosion, then depends intrinsically on grid cell size.

**b. What are the smallest scales that must be resolved by the spatial discretization? Do these length scales have practical significance for glacier modeling?**

R. As per the above, a conduit model with a single conduit per network edge would have to resolve the scale of individual conduits. That is, the scale between adjacent conduit junctions. While that may be very small, it is the price that has to be paid to capture channelization. In our approach, as in that of Werder et al, we try to sidestep that slightly by lumping n_c conduits onto the same network edge, to allow realistically a coarser "resolution" (i.e. spacing between nodes). See above re: the meaning of resolution, however.

**c. Is the model stable to perturbations of all wavelengths? This question is motivated by the observed "very abrupt" pressure changes. Consider, for example, Equation 17 in the supplement to Schoof 2010. The term v_m depends on the effective pressure gradient, which suggests that large effective pressure gradients may change the sign of the term in parentheses, and therefore destabilize flow. Is this analysis correct? If so, at what wavelengths does destabilization occur? How are these related to the wavelengths in the previous point.**

R. As before, the model is not a continuum model, so the notion of continuous wavelength may be mistaken. Suffice it to say the following: If we took a one-dimensional "network" (effectively, "nodes on a string"), the model as formulated would be a legitimate finite-volume-type discretization of a pde model for a single channel, provided we make the parameter V_p proportional to the mean distance from the node in question to its neighbours (effectively, it has to represent storage capacity per unit length of the channel in that case, integrated over a single cell in the finite volume discretization). The corresponding one-dimensional pde problem is well-posed, in the sense that it is stable to short-wavelength perturbations (and that includes the action of the v_m term mentioned above). For longer wavelengths, we can get an instability (whose onset depends on the storage capacity per unit length) that can be explained physically – in fact, this instability is related to how jokulhlaups work and has been explored partly in Schoof et al 2014; a more theoretical take on this has been sitting on one (CS) of our desks for the last three years. That instability, which occurs for sufficiently long domains and sufficiently large storage capacities, is however not the point of the question, we suspect.

Instead, we assume that "wavelength" refers to the wavelength of perturbations in the cross-flow direction, and here the point about the model not being a continuum model in that sense becomes important. A more refined version of our model would solve the one-dimensional single conduit pde model referred to above on each network edge, treated as a line segment, so there would have to be many "grid points" on each network edge, resolving the spatial variations in N and S_R, S_K between the nodes of the network. At the nodes in our network, the individual conduits would join and be coupled through mass conservation and continuity of N. (As an aside, note that this approach can actually be fitted into our numericla framework, by putting additional network nodes connected to only 2 network edges between the nodes we already have, which generally have more than two edges connected to them. That procedure is generally overkill, in the sense that it yields a more accurate solution, but not one that is substantially different from the simpler network we are using in practice.)

What the more refined version of the model cannot do is solve a generalization of our single-conduit pde model to two dimensions (Note that this is a very different idea from a network of connected one-dimensional conduits; the change in dimensionality of the domain is what matters). A two-dimensional continuum model would not be well-posed, in the sense that it would be unstable to arbitrarily short wavelength perturbations. (The appendix to Schoof et al 2012 may be more instructive in that regard.)

Once more, the network model tries in effect to resolve individual conduits, not an actual continuum sheet. In that network, the channelizing instability will in fact typically involve some channels growing at the expense of their immediate neighbours, which in a sense is the equivalent of a short wavelength instability in a continuum model, except there is a "shortest scale" in our network problem (as opposed to the continuum counterpart), set by the spacing of individual channels. Instability at that discrete scale is not evidence of ill-posedness, while short-wavelength instability in its continuum counterpart would be.

In the long run, the instability in the network model leads to a coarsening of the channel structure: initially, very closely spaced channels grow, for instance as a pattern of alternating growing and shrinking channels. However, competition between nearby growing channels eventually leads to one channel winning out over its nearby competitors, so that locally only one channel survives. This is evident for instance in figure 3 of Schoof 2010, or in the supplementary movies #1 and #7 for that paper (where initially quite a dense channel structure emerges, which then coarsens). An attempt to understand the length scale for the coarsening (that is, of the length scale over which a single channel will no longer emerge victorious but at which different channels can co-exist) can be found in section 4.2 of the supplementary material to that paper. Obviously, the coarsening is a nonlinear effect that is not covered by any standard linearization techniques.

As we have pointed out, this really pertains to some of the existing literature on which our model builds, rather than the model development in the paper itself, so we have not incorporated any of the material above into the paper.

**d. This line of questioning is based in part on my experience with subglacial hydrology modeling in the paper Lipovsky and Dunham (2015, JGR). In that paper we showed that there is no flow destabilization (at least not at glaciological flow velocities) in a sheet configuration without melting when elastic effects are taken into account (and with other assumptions).**

R. We suspect there is a difference in scales being assumed here. We are interested in pre-existing "conduits" separated by areas of ice-bed contact, rather than an actual sheet, and the conduits are large enough to allow sufficient drainage to be potentially subject to enlargement by dissipation-driven melting, in the same way an R-Channel works.

**e. Some small points: should the symbol S in Equation 1a be S_{R,ij}? Or is S another quantity? Same with Equation 1b. Also, S_{K0} is not defined in the text.**

R. Yes, S should have been $S_{R,ij}$ and $S_{K,ij}$ in (1a) and (1b), respectively. This has been corrected. We have also amended the text below the equations to say "and $S_{R0}$ as well as $S_{K0}$ are cavity-size cut-offs at which further conduit enlargement drowns out bed obstacles"

**3. Connections between observation and model**

**a. I was disappointed by Section 5.2. Up to this point, I was carried along in the narrative of the paper: the reader learns about a dizzying array of new data, their broader interpretation, and then the formulation of a model improvement. But then I'm not sure what I'm supposed to learn from these simulations. Is the fit to data good? Does it capture some of the aspects of the field observations and not others? Given the ambitious scope of the paper, a much more extensive discussion of these topics is warranted.**

R. We have fully rewritten section 5.2, adding a second model run without the size cut-off as requested, and making sure to state clearly the insights provided by the model.

**b. I would strongly recommend the creation of a new "Section 5.3: Discussion of the Simulations". There were so many observations in Section 3 that I had a difficult time keeping track of all of them (see later comment). As written, there is no relationship drawn between Figures 16 and 17 and the main observational results/figures.**

R. We have expanded significantly on the description of model results (see above). Because the simulations we report on are motivated by trying to address specific features of the drainage observations, we have not created a separate section.

**c. Near the last line of the paper it is stated, somewhat belatedly, that "However, the ability of the system to fully shut-down requires the incorporation of other physical process that could allow the reactivation of the drainage system during the spring event, something that is probably accomplished by over-pressurization." This should be included earlier, in a potential model discussion section.**

R. We have included this in the description of the drainage model results (section 5.2).

**d. Is the model capable of describing stage 1, 2, and 3 as defined in Section 4?**

R. The model is not capable of describing the beginning of stage 1, where a widespread set of new connections needs to be generated quickly, see the point identified immediately above:

"However, the ability of the system to fully shut-down requires the incorporation of other physical process that could allow the reactivation of the drainage system during the spring event, something that is probably accomplished by overpressurization".

The model also requires a more careful treatment of normal stress redistribution, in particular in as- sociation with isolated and closely spaced cavities of very different water pressures. This is left for future work. In the future, we also hope that it will be possible to test models like the one presented here or more sophisticated versions of it, against detailed borehole datasets such as that from South Glacier.

The remaining stages pertain to channelization and focusing of flow, which pre-existing network models are known to be able to do (see Schoof 2010 for instance, as well as the Werder et al 2013 model). The only novelty (compared to these pre-existing network models) we require here is the ability to create fully disconnected regions that can, however, still evolve. That is why we have focused on the ability of the model with a percolation cut-off to cause switching events and evolving disconnected regions.
We also explicitly state that rapid, large-scale connection at near zero effective pressure cannot be captured by the modified model, see the new Model Results section.

**e. Does the observed spatial heterogeneity (Section 3) factor into the choice of smoothing length scale?**

R. Not directly. There are three smoothing length scale that one could justify physically: the linear transverse dimensions of a conduit, the spacing between conduits, and the ice thickness. In typical continuum models of cavity formation (e.g. Schoof 2005, Gagliardini et al 2007), the first two (conduit size and conduit spacing) are assumed to be comparable, and the effect of one cavity is felt roughly within that distance, so it makes sense to use conduit spacing as the averaging length scale, which is effectively what we do. Without more detailed knowledge of the detailed conduit configuration at the bed, it is difficult to be sure that this is what the observed spatial heterogeneity actually shows – our suspicion is that, if anything, we undersample that heterogeneity in the field. We are currently trying to test some of this using a record of switching events in the data (specifically, whether we can predict switching events using only pressure data in the vicinity of boreholes that switch on and off repeatedly), but that work will be reported elsewhere.

**f. The bottom panels of Figure 17 would be better plotted in terms of water pressure (units equivalent water height) so that they can be easily compared to the rest of the figures in the paper...**

R. As we state in the revised results section, our intention for the model is to investigate the qualitative behaviour of the model with a percolation cut-off. We do not have sufficient data on surface melt production or, more importantly, surface melt routing, in order for our calculations

to have anything more than a qualitative relation with the observed field data. Moreover, the actual magnitude of pressures depends not only on the water supply rates but also on the parameters in the model. As discussed in the appendix to Schoof et al (2014), the closure rate parameter c_2 in particular is not well constrained, but ultimately dictates the pressure scale. We have chosen Pa as units in order not to give too great a weight to the absolute pressure values calculated here, in addition, they are values of effective pressure in contrast of water pressure as in the rest of the paper. That said, 100 m of water are equivalent to 1 MPa."

**g. ...Which of the various observed time series should the reader associate with the four panels Figure 17d-g?**

R. As before, the runs are idealized and so a direct comparison is not appropriate. However, we now make clear at the start of the results section that our motivation is in the switching events of stage 2 – in particular, the abundant switching events evident in Figure 10g (see also Figure 8c) during the part of the drainage season that we have associated with a potentially channelized drainage system.

**4. Comments on the writing**

**a. There are so many important points in Section 3 that I had a difficult time sorting through all of them. I suggest adding a writing device to emphasize the most important ones. This is partially a stylistic choice. One option would be to enumerate the points at the start. Another option would be to align subsection headings with main points.**

R. It is a very reasonable concern and it have been points by other referee too. After considering multiple possibilities we have included an extended overview at the end of the introduction. The following text has been added after line 32 of page 3:

"To help the reader to navigate through the numerous observations presented in this paper, we provide below an extended overview of its contents, highlighting the most important points to be considered:

- The observed drainage system consists of three main components (section 3.1)
    1. Channelized: efficient, turbulent drainage at low water pressure
    2. Distributed: slow water velocities, damped response to diurnal meltwater input, high water pressure
    3. Disconnected: near-overburden mean water pressure with no diurnal variations
- The "disconnected" areas display a small but statistically significant and sustained drop in mean pressure during the melt season, suggesting weak connections potentially through porewater diffusion in till (section 3.1 & 4.2).
- The connected drainage system consists of spatially distinct parts (subsystems) that appear to act independently. Each is characterized by a common diurnal pressure variation pattern that differs markedly from other subsystems (section 3.2 & 4).
- Pressure variations in boreholes in disconnected areas can also occur due to bridging effects and potentially due to ice motion, the latter giving rise to low-amplitude, high frequency pressure variations shared by distant boreholes (section 3.2, 4.2 & 4.3).

- Observations suggest the existence of a dense network of englacial conduits, but it is unclear if these can transport water over extended distances horizontally (section 3.3 & 4.2).
- During a spring event, a large distributed drainage system quickly develops over a large fraction of the bed. This splits into an increasing number of subsystems over the summer season, each potentially focusing around a channelized drainage axis. The extent of disconnected areas of the bed grows as a result (section 3.4 & 4).
- The transition from connected to disconnected is abrupt, with the connected parts of the bed having a high hydraulic diffusivity ([new] section 3.5 & 4.2). Disconnection and reconnection "events" typically occur as water pressure is falling and rising, respectively. These observations motivate the modification of existing drainage models presented in section 5.
- The timing and degree of channelization reached by the subglacial drainage system varies widely depending on weather and surface conditions during summer, and the spatial pattern of drainage can change from year to year ([new] section 3.6 & 4.1).
- Abrupt growth of the distributed drainage system, analogous to that observed during the spring event, can be observed during the summer in response to a sudden, abundant meltwater input following an extended hiatus, the latter usually cased by a mid-summer snowfall event (section 4)."

**b. The manuscript, especially Section 3 and 4, would be improved by revision for brevity. There is a lot of repetition, particularly in Section 3. The authors mention at least four times, for example, that clustering is subjective.**

R. Indeed, we have moved section 3.6 to section 1 of the  Supplementary material for brevity. And the clustering criteria is now addressed only by the paragraph starting on line 7 of page 8. To which the following text was added:
"All borehole groupings presented in the following figures were manually selected using the same criteria as described for Fig. 5."

**Answers to referee #3**

**General comment**

**The authors report a new set of observations of water pressure at the base of a glacier. The amount and quality of data acquired in this study are particularly impressive and unique. Based on this comprehensive dataset, a thorough analysis is conducted in order to distinguish typical behaviors of the subglacial hydrology network based on analyzing characteristic spatio-temporal patterns in the measurements. Observations are generally in agreement with expectations from theory, except the finding that many portions of the bed are observed to be hydraulically isolated, a feature that yet is not accounted for in subglacial hydrology models. To overcome this lack, the authors present a modelling framework (based on the adaptation of existing theory) that allows explicitly treating these hydrologically isolated parts of the bed.**

**Overall, I find the study particularly interesting and novel, since it provides new observational constraints on subglacial hydrology, as well as a unique and comprehensive dataset of interest by a large community. For these reasons I strongly recommend this paper for publication. However, before so, significant revision is needed in order to clarify text in places, better structure observations and clarify results. Below I provide specific comments that hopefully will help the authors to improve this. Moreover, the complexity and lengthiness of the paper is further reinforced by the inclusion of a modelling part at the end. Although I clearly appreciate the modelling effort, I am not convinced that this section really fits in this observational paper. As is I feel like lots of readers won't even notice the modelling part of the paper, especially given the strong imbalance between the long and extensive analysis of data and the short modelling analysis provided at the very end. For these reasons I strongly recommend the authors to consider publishing this modelling work separately, and my comments below are limited to the observational part.**

**Detailed comments**

Section 2

- **Some context information about the glacier and its environment is missing. I think this information is needed for the reader to make best sense on what type of general glacier and hydrology regime.**
  - **What are the typical values for glacier surface speed (in winter versus in summer)?**
  - **What are the expected sliding velocities (even rough estimates would be useful to know)**

    R. After line 14 on page 5 the following has been added:
    "Surface velocities were measured with a GPS array (Flowers et al., 2014), and display a strong seasonal contrast. The velocity at the GPS tower at the centre of the array (see Fig. 2) varied from 30.6 to 17.9 m/year between summer 2010 and early spring 2011. Modelled basal motion in our study area accounts for 75–100% of the total surface motion (see Fig. 6b in Flowers et al. (2011), where our study area is located between 1600 and 2500 metres)".

- **Can the authors give a qualitative sense on the potential effects of basal water pressure on glacier dynamics for this glacier and at this particular location where water pressure is monitored?**

  R. The most plausible cause of the variations mentioned in the previous point, is the effect of varying basal water pressure, as mentioned generically in the introduction. The lack of direct evidence among the observations presented in the paper to support anything more specific makes us hesitant to postulate more detailed effects than those already described in the introduction.

- **What are typical outlet water discharge values and how much do they typically vary from winter to summer?**

  R. At the end of Section 2 "Field site and methods" we have added the following paragraph:
  "The limited available stream gauging data suggests typical summer flow around 1-2 m$^3$/s, with maximum values around 5 m$^3$/s and minima below the measuring capacity of the gauging station (Crompton et al., 2015). However, the outlet stream was never observed to run dry (J. Crompton, personal communication).".

  **Since the the study is motivated by understanding the links between hydrology and sliding (see intro), I think it would be good to give a sense on these aspects to the reader, even if these statements are brief and qualitative.**

- **There is also missing information about how the glacier evolved over the past 8 years during which basal water pressure has been monitored. In particular, did glacier thickness vary over the course of the 8 years of experiment? If yes please give an estimate about how much.**

  R. After line 9 on page 5 we have added the the following:
  "The average net mass balance over the whole glacier during the period 2008-2012 was estimated to be between -0.33 and -0.45 m/year water equivalent (Wheler et al., 2014), corresponding to 37-51 cm/year of average glacier thinning. Elevation changes in the study area derived from differential Global Positioning System (GPS) measurements of borehole locations (taken after drilling) suggest a thinning of 59 cm/year over the same period, and 37 cm/year in the period 2008-2015."

**Section 3**

- **Figure 4: I find it quite complicated to identify which hole goes with which measurement. Would there be a way to improve clarity in this figure? Maybe zoom in the map, or make two map subsets to make the color code easier to see.**

  R. As mentioned to referee #1, we have removed the 3D shading of the dots in the map and

increased the size of the markers to make identification easier.

- **Line 16 p 7 to line 6 p 8 : unclear text with long sentences.**
"Panel c of Fig. 4 shows the pressure recorded in the fast-flow borehole for the first 33 days after installation. Panel d shows the pressure records in three boreholes along the same line across the glacier at 15 m spacing. The lack of similarity between the fast-flow hole pressure record and those from other nearby boreholes differs from the behaviour of most boreholes that display diurnal pressure oscillations: typically, such boreholes show a signal similar to one or more neighbouring boreholes, forming a cluster that extends some distance laterally across the glacier (see section 3.2). In the case of the fast-flow borehole, somewhat similar temporal pressure patterns were observed at a much larger distance downstream, as shown in panels e and g of Fig. 4, and less so in panel h, while a set of boreholes exhibiting very different variations close to those in panel d is shown in panels f. For reference, panel i shows the remaining pressure time series recorded in the same area, highlighting the diversity of pressure patterns observed. No systematic time lags were found between peaks on the fast-flow borehole and pressure peaks of boreholes displayed in panels e and g."

R. The text has been changed to:
"Panel c of Fig. 5 shows the pressure recorded in the fast-flow borehole for the first 33 days after installation, and panel 5d shows the pressure records in three boreholes along the same line across the glacier at 15 m spacing. Note the lack of similarity between the fast-flow hole pressure record and those from other nearby boreholes. This lack of similarity contrasts with the typical behaviour of boreholes exhibiting diurnal pressure oscillations. Such boreholes usually share a similar pattern of pressure oscillations with one or more neighbouring boreholes, forming a cluster that extends some distance laterally across the glacier (see section 3.2).
However, in the case of the fast-flow borehole, somewhat similar temporal pressure patterns were observed downglacier and at much larger distances than the 15 m lateral borehole spacing, as shown in panels 5e and 5g, and less so in panel 5h. By contrast, a set of boreholes exhibiting very different variations close to those in panel 5d is shown in panels 5f. For reference, panel 5i shows the remaining pressure time series recorded in the same area, highlighting the diversity of pressure patterns observed. No systematic time lags were found between peaks in the fast-flow borehole and pressure peaks of boreholes displayed in panels 5e and 5g."

- **P 7 to p 8: the whole discussion on what aspects borehole measurements have been grouped is quite vague, and repetitive. It would be good to have a single, short paragraph explaining how boreholes have been grouped, even if the criteria are qualitative (by eyes is a good enough justification), and then go on with the description without repeating how the selection has been done.**

R. We have gather the information about the grouping criteria in one clear paragraph (starting on line 52 of page 6 of the new manuscript) as suggested.

- **Label of Fig 6: amplitude offset? Or phase offset? Looks like it's amplitude.**
"We have applied offsets to make the agreement between the records clearer. These are, in order, 27, 26, 24, and 29 meters in (a), and 27, 20, 22, and 27 in (b)."

R. They are pressure offsets, with no changes in amplitude (just adding a constant value but no multiplier). We will make that clear by changing the above paragraph to: "We have applied a constant value offset in pressure to each time series (meaning, added a constant to the directly measured pressure) to make the agreement between the records clearer. The offset values are, in order, 27, 26, 24, and 29 metres in (a), and 27, 20, 22, and 27 in (b)."

- **I suggest to split section 3.1 into two sections. One would be something like "global overview of the dataset" with Fig 4 and 5 and the other would be something like "Diurnal and seasonal cycles in slow and fast flowing water" (Fig 3, 6 and 7). I think this would make it easier to read.**

  R. We considered ways of doing this but were consistently stumped by the fact that the diurnal and seasonal cycles (especially the diurnal ones) were the strongest indicators of drainage activity, and could not conceive of a satisfactory way of splitting the section.

- **Line 5 to 15, p 12: unclear paragraph. Too long sentences.**
  "When the whole data set is viewed over a given time window during summer, it is often possible to identify multiple subsets of boreholes showing very similar temporal pressure variations within that subset (often recognizable by the way in which the amplitude of diurnal oscillations changes over time), but these temporal variations are different from other boreholes. One example of this phenomenon comes from the boreholes in Fig. 4f, where we can see a group of boreholes that display a very coherent signal but with a distinctive two-day period. The boreholes in panel f are directly adjacent to those in panel e, we have associated with the fast-flow borehole and which show a very different, diurnal pattern of pressure variations (see also panels c and g). Less clear-cut though indicative of the same phenomenon is Fig. 5, where we see boreholes in panels d–f that exhibit quite different diurnal pressure variations to those observed in the group associated with the slow-flow borehole in panel c. Figure 3 of Schoof et al. (2014) also shows an example of the same phenomenon during July and August 2011: borehole B in that figure is, in fact, one of a group of 5 that exhibit almost identical diurnal water pressure oscillations that are quite distinct from those in boreholes A1–A6 in the same figure."

  R. The paragraph has been edited as follows:
  "When the whole dataset is viewed over a given time window during summer, it is often possible to identify multiple clusters of boreholes, each exhibiting a specific pattern of temporal pressure variations. Often, these patterns are defined by the way in which the amplitude of diurnal oscillations changes over time. While boreholes in a given clusters will share the pattern of temporal variability, this will differ significantly from the pattern of temporal variability in the other clusters. One example of this phenomenon comes from the boreholes in Fig. 5f, where we can see a group of boreholes that display a very coherent signal but with a distinctive two-day period. However, those boreholes in figure 5f are directly adjacent to those in 5e. The latter by contrast show a very different pattern of diurnal pressure variations (that we have associated with the fast-flow borehole, along with panels 5c and 5g). Less clear-cut, though indicative of the same phenomenon is Fig. 6, where we see boreholes in panels d-f that exhibit quite different diurnal pressure variations from those observed in panel c (the group associated with the slowflow borehole). Figure 3 of Schoof et al. (2014) also shows an example of the same phenomenon during July and August 2011: borehole B in that figure is, in fact, one of a group of 5 that exhibit almost identical diurnal water pressure oscillations that are quite distinct from those in boreholes A1–A6 in the same figure."

- **Line 10 p 13: Comparing panel b with panel e in Fig 8 I do not see the "inverted" or anti-correlated relationship. . . Wording and support from figures is confusing here.**

  R. We agree with the difficulty  of distinguish the features we mention. Therefore, we have highlighted in black one line that clearly displays the anti-correlated feature within the group. Without that highlight, the details are lost due to the overlap between lines. The paragraph has been changed to:
  "The group of 14 boreholes in panel e of Fig. 8 also shares common diurnal pressure variation patterns, though this is not immediately clear as the mean pressures and amplitude of pressure variations varies significantly. For that reason we have highlighted in black one line that shows these variations clearly. Notably, these variations are "inverted" versions of the pressure variations seen in panel b, with peaks becoming troughs and vice versa."

- **Line 28 p 13: Fig. 9 is very lately introduced here. Actually figure 9 seems to help in the understanding of "inverted" or anticorrelated signals, but it comes too late. Perhaps to be place earlier?**

  R. Figure 9 shows a different kind of anticorrelated signal – the particular features being highly correlated or anticorrelated high-frequency pressure variations, occurring over large spatial distances, which are distinct from those we were trying to highlight in Fig 8. As a result, in order to avoid misinterpretation, we don't want to present the figure  earlier. However, we hope that the above changes in figure 8 and the clarification of the text will help in the understanding of anti-correlated signals.

- **P 17: I find the difference between the title of 3.4 (seasonal evolution) and title of 3.1 (annual cycle) to be too weak. . . As is I get lost trying to understand what's new in 3.4 that could not be observed or has not been said in 3.1.**

  R. We attribute this confusion to badly chosen section titles. To set the right expectations regarding the content, and help the reader we have change the title of Section 3.1 (Annual cycle and water flow) to "Modes of water flow: fast, slow and unconnected".
  In addition, we have split Section 3.4 (Seasonal evolution) in two, that correspond now to sections 3.4 and 3.5.  The new section 3.4 is titled "Seasonal development of the subglacial drainage system". The new section 3.5 starts after line 6 of page 18, and it will be titled "Basal hydrology transitions and 'Switching events'".

- **Section 3.6: I suggest to put this section in supplementary material, and just have a single paragraph in the main text that states how and to which extent observations could be biased by changes in data quality. If kept in the main text, this paragraph could even be placed in a separate section before results are exposed.**

R. As mentioned above, section 3.6 was moved to section 1 of the Supplementary Material. References to it were added to the Methods section. Figure 15 was kept in the main text to be referenced by section 4.4, where we have also added part of the text of section 3.6 that was relevant to that figure. The paragraph starting on line 21 of page 29, now reads as follows:

"It is likely that with time, some sensors can become encased in ice, as suggested by the fact that older sensors are less likely to show diurnal oscillations (for that reason, old isolated sensors were often decommissioned before they ceased to produce a signal), and the observations in doubly-instrumented boreholes (see section 1 of the supplementary material). Digital confinement data suggest that in some cases, as in Fig. 15, the termination/initiation of diurnal oscillations is associated with an increase/decrease in confinement. This observation would also be consistent with ice encapsulation of the sensor during winter."

**Section 4**

- **Would be good to have a section or a paragraph that summarizes all key observations, which would be placed outside the discussion section. Then the discussion section would only be based on the summarized, main observations. As is it is embedded and its makes it hard to read.**

  R. This is a very similar concern than the one expressed by referee #2 on the first of his "Comments on the writing". Please refer to the answer to that comment.

- **I don't see what is the difference between 4.4 data interpretation and what's discussed earlier. Isn't the earlier discussion also data interpretation?**

  R. Indeed the title is not appropriate. We have changed it from "Data interpretation" to "Data interpretation caveats".

**Section 5**

- **I suggest to remove that section from the paper, and write a separate paper on the modelling aspects.**

  R. Further work on models for dynamic connection and disconnection in drainage models is clearly desirable. As in Hoffman et al (2016), our stated goal here is to link our observations directly to the shortcomings of existing models. This motivates the introduction and structure of the paper. As a result, we believe that it is sensible to point out avenues by which they can be fixed. The model alteration we propose is not an enormous one, and we feel it's preferable to propose it in context than to write an overly short paper simply modifying a couple of equations in an existing model. We are also interested in putting the idea "out there" so that other groups might feel encourage to work on updating drainage models. We view this as analogous to the work in Hoffman et al (2016), in the sense that we do not aim to have the last word on the subject.

[revised manuscript text omitted]

New version of the figure comparing results from model runs with and without cavity-size cut-off

[Figure]

**Figure 16.** Snapshots of drainage system evolution for the model without a percolation cut-off (left), and with it (right). Panels a & e: $S_R$ conduit size. Panels b & f: $S_K$ conduit size. Panels c & g: effective pressure, black lines are 100 m surface contours, grey lines 100 m bed contours. Panels d & h: connectedness of conduits, indicated in blue if $S_{R,ij} > S_{PR}$ or $S_{K,ij} > S_{PK}$ along a given edge, in white otherwise. Red dotes indicate moulin locations, size of dot scaled with instantaneous water supply. Row a-d show solutions at $t = 7.8$ days and row e-h at $t = 20.6$ days. Panel i: water supply time series for all moulins in the domain.

Over time, water transport becomes concentrated in some areas, and probably becomes channelised: water flow ends up focused in R-channels surrounded by a distributed drainage system that carries relatively low water fluxes. Borehole water pressure data in most cases do not allow the direct identification of channels. In fact, in most cases, our borehole array probably fails to intersect the narrow R-channels. However, in one instance we were able to confirm the existence of a channel from direct observation in a borehole in which the lowermost 50 cm were occupied by turbulent water flow.

The increase in effective pressure associated with  channelisation leads to the progressive shut-down of drainage activity  in the surrounding distributed drainage system, possibly due to basal cavities becoming isolated from each other as they shrink under the effect of a larger effective pressure. During long and hot enough summers, most of the bed can become disconnected, concentrating drainage in narrow pathways.

The  eventual complete shut-down of the entire drainage system at the end of the summer season is presumably the result of low water supply: high effective pressure and low dissipation rate in channels allow basal conduits to close. This appears to be strongly linked with the appearance of fresh snow cover, rather than the arrival of low temperatures alone (see section 4).

Most of  our observations are consistent with borehole data from other sites. However, the density of boreholes at South Glacier has allowed us to identify, in particular, the prevalence of "switching events", through which the drainage system focuses, and the  disconnected areas enlarge. Such  disconnected areas always exist, even during the spring event. Disconnected parts of the bed are necessary to account for many aspects of our data, including anti-correlation between  borehole pressure time series, above-overburden water pressures, and the occurrence of strongly correlated high-frequency

[Figure]

New version of the figure comparing results from model runs with and without cavity-size cut-off

[revised manuscript text omitted]

---

## Author Response (AR2)

**Author's response**

The following document contains the answers to the editor and referees' comments after the second review step. As before, all comments were very pertinent and helpful, and want to express here our gratitude to the editor and the two referees that took part in the second stage of the review process.

The document includes in bold text the original comments, followed if necessary by a quote from the paper in regular text to provide context to the answers. The actual answers are presented in blue.

Responses are followed by an annotated version of the manuscript, which includes highlighted text for all deletions and additions.

**Answers to editor comments**

**- page 11, line 34: some of the pressure ?**

> R. Replaced by "the pressure variations"

**- page 15, line 20: Figs. 5 and 11**

> R. Replaced "Figures" by "Figs." as suggested

**- page 17, line 1159: Figs. 7, 9 and 11**

> R. Replaced "figures" by "Figs." as suggested

**- page 17, lines 1218-1220: "need not be accompanied" and "need not flow" look strange. Typos?**
"With a hydraulically well-connected system, this has to correspond to a low water storage capacity, so that substantial variations in water pressure need not be accompanied by the similarly large changes in stored water. Alternatively, that storage capacity could be relatively localised, so that water need not flow everywhere."

> R. It has been rephrased as:
> "With a hydraulically well-connected system, this has to correspond to a low water storage capacity, where substantial variations in water pressure do not require similarly large changes in stored water. Alternatively, storage capacity could be relatively localised, so that water does not need to flow everywhere."

**- page 18, line 1262: Figs. 7 and 9**

> R. Replaced "figures" by "Figs." as suggested

**- page 18, line 1285: Stokes flow model (Lefeuvre et al., 2018) supports**

> R. "Lefeuvre et al. (2018)" replaced by "(Lefeuvre et al., 2018)" as suggested

**- page 20, line 1533: and and -> and; also "as well as Sk0" should be removed to be consistent with the whole description of the K subscript.**

> R. Repeated words and unnecessary term descriptions removed. It now reads:
> "..., and $S_{R0}$ is the cavity-size cut-off at which further conduit enlargement drowns out bed obstacles. $P_{e,ij}$ is the effective pressure driving conduit closure (related to $N_i$ as described by equation 7 below), and $c_1$, $c_2$ are the same constants as in Schoof et al. (2014). Subscripts K refer to equivalent quantities for the K-conduits."

**- page 21, line 1607: ( and -> (and**
> R. Extra space removed

**- Table 1: value -> Value**
> R. Table heading capitalized

**Answer to referee #1:**

**- Discussion of disconnected areas of the bed forms a key aspect of the paper, and I appreciate the additional short paragraph in the introduction explaining the effects of this system on ice dynamics. However I think the paper would benefit from a brief discussion of the distinction between cavities that are fully disconnected and those that are poorly or weakly connected. In particular, in a fully disconnected cavity some assumption about compressibility of the water and/or ice is required, but is not necessarily required in the weakly connected case.**

> R. We agree that a more detailed treatment of this issue (as well as others) would benefit the paper. However, we have decided to do not add more on this topic for two reasons: First, because we are preparing a follow-up paper that uses tools that allow a better quantification of the extent and pressure of weekly connected areas, therefore, we plan to discuss that topic in more detail there. Second, to avoid extending an already lengthy paper.

> Regarding the concerns expressed, we do not think that a fully disconnected cavity conflicts with incompressibility: the pressure adjusts to keep a constant cavity volume, something that we already mentioned in the paper. Weakly connected just means that the volume changes at a rate proportional to the pressure difference between the cavity and drainage system, where the constant of proportionality measures the connectedness. The limit of a small constant gives near-zero rate of change, with the constant volume constraint again setting the pressure. We think that our simple model captures these phenomena, including the resulting difference in the long-term behaviour.

[revised manuscript text omitted]